# DATA UNIFORMITY IMPROVES TRAINING EFFICIENCY AND MORE, WITH A CONVERGENCE FRAMEWORK BEYOND THE NTK REGIME

## ABSTRACT

Data selection plays a crucial role in data-driven decision-making, including in large language models (LLMs), and is typically task-dependent. Properties such as data quality and diversity have been extensively studied and are known to enhance model performance. However, it remains unclear whether there exist other quantitative and general principles of data selection that can consistently improve performance, especially for complicated tasks. In this paper, we demonstrate that selecting more uniformly distributed data can improve training efficiency while enhancing performance. Specifically, we establish that more uniform (less biased) distribution leads to a larger minimum pairwise distance between data points, denoted by $h_{\min}$, and prove that a smaller $h_{\min}$ can slow down the training dynamics of gradient descent (GD). Moreover, we theoretically show that the approximation error of neural networks decreases as $h_{\min}$ increases. Our analysis introduces a convergence framework for GD beyond the Neural Tangent Kernel (NTK) regime, applicable to a broad class of architectures, including transformers, without requiring Lipschitz smoothness. This framework further provides theoretical justification for the use of residual connection and function composition in deep neural architectures. In the end, we conduct comprehensive experiments for supervised fine-tuning across various settings, including different optimization strategies, model sizes, and training datasets. The results consistently demonstrate that selecting data by maximizing pairwise distance significantly accelerates training and achieves comparable or better performance in LLMs across diverse datasets. Code and Datasets are available at the link: `https://anonymous.4open.science/r/data-uniformity-1A5C`.

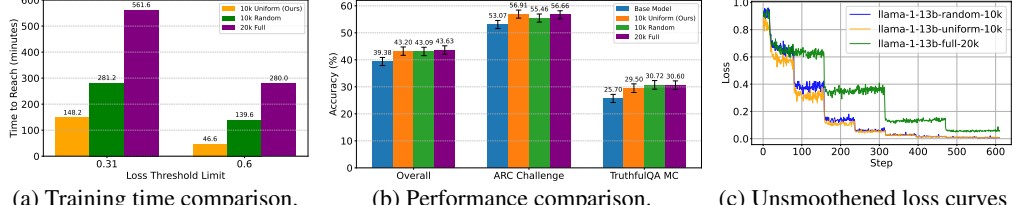

| (a) Training time comparison. | (b) Performance comparison. | (c) Unsmoothened loss curves |
| --- | --- | --- |

Figure 1: Comparison of LLaMA-1-13B training efficiency and evaluation performance across different data selection strategies. (a) **Time to reach loss thresholds** (0.31 and 0.6), reported based on wall-clock time using a single A100 GPU. The y-axis shows the time (in minutes) required to reach each target loss for the first time. Uniform selection consistently leads to faster convergence, indicating improved training efficiency. (b) **Evaluation performance on ARC Challenge and TruthfulQA MC** at matched loss checkpoints, showing that the 10k Uniform subset outperforms the 10k Random subset and matches the full 20k dataset. (c) **Training loss curves**, 10k uniform subset is faster than 10k random subset and full dataset.

## 1 INTRODUCTION

Data selection is fundamental to a lot of applications including large language models (LLMs) (Albalak et al., 2024; Zhao et al., 2023; Chang et al., 2024), such as TeaMs-RL (Gu et al., 2024) and

WizardLM (Xu et al., 2024). One main difficulty of data selection is that it is often tailored to specific tasks, which highlights the need for general and widely applicable rules to guide the data selection process. Existing rules include improving data quality and data diversity (Muennighoff et al., 2025; Albalak et al., 2024; Gu et al., 2024), which have been widely studied and are recognized for their positive impact on model performance (Gu et al., 2024; Ba et al., 2024; Gong et al., 2019, etc.). However, data selection is still a challenging problem that calls for more quantitative and broadly applicable principles.

Regarding the role of data, existing results from approximation theory suggest that uniform sampling exhibits near-optimal behavior under many circumstances. Krieg et al. (2022) showed that i.i.d. random sampling achieves optimal convergence rates, and Krieg & Sonnleitner (2024) proved that uniform sampling is asymptotically as effective as optimal deterministic schemes for Sobolev functions. In the context of learning, Dong et al. (2024) further demonstrated that uniform sampling yields near-optimal generalization in low-dimensional linear probing. These results collectively emphasize the efficiency of uniform sampling, despite its simplicity. Motivated by these results, we wonder whether selecting data to be more uniformly distributed can benefit training and lead to improved generalization performance for complicated tasks.

In this paper, we demonstrate that increased data uniformity accelerates training and achieves performance comparable to using a larger dataset. More precisely, we characterize the effect of data uniformity through the minimum distance between input data points, denoted by $h_{\min}$, both theoretically and experimentally:

- Theoretically, we first establish the relationship between data uniformity and the minimum distance $h_{\min}$ (Theorem 1): a more uniform (less biased) distribution results in larger $h_{\min}$.

- On the optimization side, we first develop a general convergence framework for gradient descent (GD) beyond the neural tangent kernel (NTK) regime (Theorem 2; Remark 1). This framework applies to a broad family of complicated architectures with residual connections, including deep transformers with feedforward and attention layers using softmax, tanh, GELU, SiLU, and other analytic activations (Definition 1). It builds on tools from measure theory and differential topology, combined with polynomial generalized smoothness (Definition 4) and local relaxed dissipativity (Definition 5), and is also applicable to general nonconvex optimization without Lipschitz smoothness. Based on this convergence analysis, we show that smaller $h_{\min}$ can slow down training (Corollary 3), and provide theoretical insights into the benefits of residual connection and function composition in neural networks (Remark 2).

- On the approximation side, we prove that larger $h_{\min}$ and smaller local maximum distance $h_{\max_{d+1}}$ lead to smaller approximation error between the neural network and ground truth (Theorem 3). This is achieved by establishing a data-dependent Bramble-Hilbert lemma, combined with ideas in computational geometry.

- Experimentally, we demonstrate the benefits of selecting more uniformly distributed data by maximizing pairwise distances between data points in supervised fine-tuning. Specifically, we employ our approach into state-of-the-art (SOTA) data distillation and data selection baselines, including TeaMs-RL (Gu et al., 2024), WizardLM (Xu et al., 2024), Zcore (Griffin et al., 2024), and LESS (Xia et al., 2024). Comparative experiments show that this uniformity-driven sampling strategy consistently leads to faster convergence and comparable or improved performance across both $\ell^2$-SGD and cross-entropy loss with Adam optimization (e.g., as shown in Figure 1). Our results generalize across multiple data sources, optimization strategies, dataset sizes, and model scales, including LLaMA-1 7B and 13B models.

## 2 RELATED WORK

**Convergence of Neural Networks.** In the overparameterized regime, the neural tangent kernel (NTK) framework, introduced by Jacot et al. (2018), has led to a series of convergence results showing that when the network width is sufficiently large—typically a high-order polynomial in the number of samples $N$, network depth $L$, and inverse of angle between data—(S)GD with infinitesimal learning rates can drive the training loss to zero exponentially (Allen-Zhu et al., 2019; Du et al., 2019; Lee et al., 2019; Zou et al., 2020; Zou & Gu, 2019; Ji & Telgarsky, 2019; Chen et al., 2020b; Song & Yang, 2019; Oymak & Soltanolkotabi, 2020). However, in the NTK regime, networks essentially behave as linear models and do not exhibit meaningful feature learning. To address this limitation, recent work has moved towards analyzing feature learning, using either gradient flow or

finite number of large steps Yang & Hu (2020); Chen et al. (2022; 2025); Ba et al. (2022); Allen-Zhu & Li (2020; 2022); Cao et al. (2022); Shi et al. (2021); Telgarsky (2022). There is also a series of works that analyze SGD convergence under various regularity assumptions, including the PL condition, quadratic growth, the aiming condition, and Lipschitz smoothness (Liu et al., 2023; Ma et al., 2018, etc.). Another line of work adopts a mean-field perspective (Song et al., 2018; Chizat & Bach, 2018; Rotskoff & Vanden-Eijnden, 2018; Wei et al., 2019; Chen et al., 2020a; Sirignano & Spiliopoulos, 2020; Fang et al., 2021), dynamical mean-field theory perspective (Bordelon & Pehlevan, 2022; 2023), and distributional perspective Han & Imaizumi (2025). In contrast to these approaches, which typically assume fixed architectures and/or infinitesimal learning rates, our work introduces tools applicable to a broad class of networks and remains effective under larger learning rates, offering a theoretical foundation for convergence in more practical and complex settings.

See more related work in Appendix A.

## 3 PRELIMINARY

We use $\|\cdot\|$ to denote $\ell^2$ norm for vectors and matrices, use $\|\cdot\|_p$ to denote $\ell^p$ norm for vectors and $L^p(\Omega)$ norm for functions, use $\|\cdot\|_{r,p,\Omega}$ to denote the Sobolev norm of $W^{r,p}(\Omega)$. For the function $f : \mathbb{R}^m \to \mathbb{R}^n$, We use $\nabla f \in \mathbb{R}^{n \times m}$ to denote the Jacobian of the map $f$ w.r.t. all variables, and use $\nabla_x f$ to represent the Jacobian w.r.t. $x$. For a vector $\alpha$, we denote $|\alpha| = |(\alpha_1, \cdots, \alpha_d)| = \sum_{i=1}^d \alpha_i$; for a set $\Omega$, we use $|\Omega|$ to denote the area of the set. The closure and interior of a set $\Omega$ are denoted as $\bar{\Omega}$ and $\Omega^o$. For two measures $\nu$ and $\mu$, we use $\nu \ll \mu$ to denote the absolute continuity of $\nu$ w.r.t. $\mu$. We denote $\text{Leb}_d(\cdot)$ to be the Lebesgue measure of a set in $\mathbb{R}^d$. We use $\text{conv}\{v_1, \cdots, v_n\}$ to denote the convex hull over $v_1, \cdots, v_n$. We denote $B_x(r)$ to be the open ball of radius $r$ centered at $x$ under Euclidean distance. We use $\mathbb{R}_{\geq 0}$ to denote all the non-negative real values.

We introduce the Sobolev space $W^{m,p}(\Omega)$, which is defined as $W^{m,p}(\Omega) = \{\phi \in L^p(\Omega) : D^\alpha \phi \in L^p(\Omega), \forall |\alpha| \leq k\}$, with Sobolev norm $\|\phi\|_{m,p} = \left(\sum_{|\alpha| \leq m} \|D^\alpha \phi\|_p^p\right)^{1/p}$ for $1 \leq p < \infty$, and $\max_{|\alpha| \leq m} \|D^\alpha \phi\|_\infty$ for $p = \infty$ where $D^\alpha \phi = \frac{\partial^{|\alpha|} \phi}{\partial^{\alpha_1} x_1 \cdots \partial^{\alpha_d} x_d}$.

Next, we introduce analyticity, polynomial generalized continuity and smoothness, and local relaxed dissipativity as key properties for proving the convergence of neural networks.

### 3.1 ANALYTIC FUNCTION

Analyticity is a useful property in both measure theory and differential topology. For a compact domain $\mathcal{D}$, real-analytic functions are dense in the space of continuous function $C(\mathcal{D})$ under Whitney $C^0$-topology (Grauert, 1958), i.e., with respect to uniform convergence on compact sets. Therefore, assuming analyticity is reasonable in many settings. It is defined as follows.

**Definition 1** (Analytic function). *A function $f(x) = (f_1(x), \cdots, f_m(x))^\top : U \to \mathbb{R}^m$ with an open subset $U \subseteq \mathbb{R}^m$, is called real-analytic on U, if for each $x_0 \in U$, the function $f_i(x)$ can be represented by a convergent power series in some neighborhood of $x_0$, for all $i = 1, \cdots, m$.*

Many activation functions and components of neural networks are real-analytic. For example:

**Corollary 1.** *Any product, sum, and composition of softmax, tanh, sigmoid, GELU, SiLU, polynomial, normalization $\tau_\epsilon(x) = \frac{x}{\sqrt{\|x\|^2 + \epsilon^2}}$ for some $\epsilon > 0$, and exponential functions, is real-analytic.*

The above corollary implies that feedforward layers using tanh, GELU, or SiLU, as well as attention layers with softmax or linear activation, are real-analytic. Consequently, architectures such as transformers, residual networks, and feedforward networks built with real-analytic activations and normalizations are themselves real-analytic.

### 3.2 POLYNOMIAL GENERALIZED CONTINUITY AND SMOOTHNESS

Classical non-convex optimization theory typically assumes Lipschitz smoothness of the objective function, a condition that is rarely satisfied in neural network training. One direction of the generalization of Lipschitz smoothness stems from the concept $(L_0, L_1)$-smoothness (Zhang et al., 2019). Building on this, Li et al. (2023) introduced the generalized smoothness for a broader class of non-Lipschitz smooth functions. Motivated by these developments, we propose a new formulation in nonconvex optimization tailored to neural network architectures, based on the following concepts.

**Definition 2** (Poly-boundedness). *A function $f(x) : \Omega \subseteq \mathbb{R}^n \to \mathbb{R}^d$ is polynomially bounded if*

$$\|f(x)\| \leq S(\|x_1\|, \cdots, \|x_{n_x}\|), \ \forall x \in \Omega,$$

where $x_i \in \mathbb{R}^{n_{i,1} \times n_{i,2}}$, and $\dim x = n = \sum_{i=1}^{n_x} n_{i,1} n_{i,2}$; $S(\cdot)$ is some polynomial whose coefficients are all positive.

**Definition 3** (Poly-continuity). *A function $f(x) : \Omega \subseteq \mathbb{R}^n \to \mathbb{R}^d$ satisfies polynomial generalized continuity if*

$$\|f(x) - f(x')\| \le S(\|x_{\max,1}\|, \cdots, \|x_{\max,n_x}\|)\|x - x'\|, \; , \; \forall x, x' \in \Omega,$$

*where $\|x_{\max,i}\| = \max\{\|x_i\|, \|x_i'\|\}$, and $S(\cdot)$ is some polynomial whose coefficients are all positive.*

**Definition 4** (Poly-smoothness). *A function $f(x) : \Omega \subseteq \mathbb{R}^n \to \mathbb{R}^d$ in $C^1$ satisfies polynomial generalized smoothness if $\nabla f$ satisfies polynomial generalized continuity.*

See Appendix C.2.1 for comparisons between poly-smoothness and generalized smoothness (Li et al., 2023). The three definitions are easily satisfied in typical neural network settings. Specifically:

**Corollary 2.** *Any product, sum, and composition of polynomial, softmax, tanh, sigmoid, normalization $\tau_\epsilon$, GELU, SiLU, satisfies polynomial boundedness for both the functions and their gradients, polynomial generalized continuity, and polynomial generalized smoothness.*

The corollary above implies that many neural networks, including transformers, residual networks, and feedforward networks employing the aforementioned functions, satisfy the three properties.

### 3.3 Local relaxed dissipative condition

Besides poly-boundedness, continuity, and smoothness, additional conditions are needed to guarantee convergence. Inspired by some weak dissipative conditions from operator theory (for example, hypomonotonicity (Moudafi, 2004)), we define the following local relaxed dissipative condition:

**Definition 5** ($(x, x^*, r, \rho, \epsilon)$-dissipativity). *A function $f(x) \in C^1$ satisfies the $(x, x^*, r, \rho, \epsilon)$-dissipative condition near $x$ if there exists some stationary point $x^*$ and some constant $r > \|x - x^*\|$, s.t., $\forall y \in B_{x^*}(r) \backslash \{x : \|\nabla f(x)\| \le \epsilon\}$, we have*

$$\nabla f(y)^\top (y - x^*) \ge -\rho \|\nabla f(y)\|^2,$$

*where $\epsilon \ge 0$, and $\rho \in \mathbb{R}$ is a constant depending on $x, x^*, \epsilon$, but independent of $y$.*

The above condition characterizes the quality of the local landscape near a stationary point by comparing the gradient direction and the stationary point direction. It holds trivially with $\rho = \frac{1}{\epsilon} \max \|y - x^*\| = \frac{r}{\epsilon}$ by Cauchy-Schwartz inequality. Note that $\rho$ can be negative, and a function $f$ with more favorable dissipative properties leads to a smaller $\rho$. For example, if $f$ is locally convex near $x^*$ with some $r > 0$, then $\rho = 0$.

## 4 Theory

In this section, we consider the data $\{(x_i, y_i)\}_{i=1}^N$ where $x_i, y_i \in \mathbb{R}^d$, and define $h_{ij} = \|x_i - x_j\|$, and $h_{\min} = \min_{i,j} h_{ij}$. We mainly use $h_{\min}$ to characterize the effect of data distributions. More precisely, we show that an increase in $h_{\min}$ corresponds to a more uniform (less biased) distributions (Section 4.1). We then prove that a larger $h_{\min}$ leads to faster training (Section 4.2) and smaller approximation error (Section 4.3). Additionally, we present a convergence framework for a family of neural networks under GD beyond the NTK regime (Remark 1). This framework provides theoretical support for the practical effectiveness of structures such as residual connection and function composition (Remark 2).

We assume that the data satisfy the following properties:

**Assumption 1** (Data). *Assume $x_1, \cdots, x_N \overset{i.i.d.}{\sim} \pi(x)dx$, where $\pi(x) \in L^\infty$ is supported on an open set $\Omega \subseteq \mathbb{R}^d$ with $|\Omega| < \infty$, and $\pi \ll \text{Leb}$ on $\Omega$. Assume the ground truth function $g(x) \in W^{r,p}(\Omega)$ for $1 \le p < \infty$ and $r \ge 1$, and $y_i = g(x_i) \in \mathbb{R}^d$.*

In the above assumption, the input data are sampled i.i.d. from some density, which is absolutely continuous w.r.t. Lebesgue measure on its support, and lies in $L^\infty$ to exclude degenerate cases like point masses. For the ground truth function $g$, we assume that it is relatively smooth.

Unlike analyses in the NTK regime (Allen-Zhu et al., 2019; Zou et al., 2020; Zou & Gu, 2019; Chen et al., 2020b; Oymak & Soltanolkotabi, 2020, etc.), which typically assume normalized input data so that $h_{\min}$ reflects only angular separation between points, we do not assume normalized input, and therefore, $h_{\min}$ encodes both the angular separation and norm difference between input data.

## 4.1 MINIMUM DISTANCE $h_{\min}$ VS MORE UNIFORM DISTRIBUTION

In this section, we discuss the relationship between $h_{\min}$ and the sampling density function of input data $\pi(\cdot)$: less biased density results in larger $h_{\min}$, and consequently admits faster convergence (Theorem 2) and smaller approximation error (Theorem 3).

Before stating the result, we introduce the following notations. Let $\pi_{\max} = \|\pi(x)\|_\infty$. For some $0 < \bar{\pi}_{\max} \le \pi_{\max}$, we define $\Omega_{\max} = \{x \in \Omega \,|\, \pi(x) \ge \bar{\pi}_{\max}\}$. Below is the main theorem showing the relationship between $h_{\min}$ and $\pi_{\max}, \bar{\pi}_{\max}$.

**Theorem 1.** *Suppose Assumption 1 holds. For any biased distribution such that there exists a ball* $B\left(C\left(\frac{-\log\delta}{\bar{\pi}_{\max}(N-1)V_d}\right)^{1/d}\right) \subseteq \Omega_{\max}$, *and* $\frac{\pi_{\max}}{\bar{\pi}_{\max}} = \mathcal{O}(1)$, *we have with probability at least* $1 - 2\delta$,

$$\left(\frac{2\delta}{\pi_{\max}N(N-1)V_d}\right)^{1/d} \le h_{\min} \le C\left(\frac{-\log\delta}{\bar{\pi}_{\max}(N-1)V_d}\right)^{1/d},$$

*where* $V_d = \frac{\pi^{d/2}}{\Gamma(d/2+1)}$ *is the volumn of $d$-dimensional unit ball, and $C > 0$ is some universal constant.*

The above theorem provides both the lower and upper bounds on $h_{\min}$ in terms of $\pi_{\max}$ and $\bar{\pi}_{\max}$. The specific assumption on the biased distribution requires the density to remain high over a sufficiently large region, avoiding narrow peaks with near-zero support. When $\pi(\cdot)$ is biased, both $\pi_{\max}$ and $\bar{\pi}_{\max}$ can be large, leading to smaller upper and lower bounds for $h_{\min}$. Conversely, if $\pi(\cdot)$ becomes more "flattened", then $\pi_{\max}$ decreases, and $\bar{\pi}_{\max}$ also tends to decrease when the flattening is significant. Thus, a more uniform (i.e., less biased) density yields a larger $h_{\min}$.

## 4.2 OPTIMIZATION: CONVERGENCE OF NEURAL NETWORKS BEYOND NTK REGIME

In this section, we establish the convergence of a broad class of neural networks under GD beyond NTK regime (Theorem 2). Moreover, smaller $h_{\min}$ tends to slow down convergence (Corollary 3).

**Neural networks, Loss, and Algorithm.** We consider a family of neural networks as follows:

$$\begin{aligned} u_{0,i} &= x_i; \\ u_{\ell+1,i} &= u_{\ell,i} + \varphi_\ell(\theta_\ell; u_{\ell,i}), \forall \ell = 0, \cdots, L-1; \\ f(\theta; x_i) &= \varphi_L(\theta_L; u_{L,i}) \end{aligned} \tag{1}$$

where $\theta = (\theta_0, \cdots, \theta_L)$ is the collection of weights with the $\theta_i$ corresponding to the $i$th layer. For the rest of the paper, we sometimes omit the subscript of $\theta$ for simplicity. We use the following $\ell^2$ loss

$$\mathcal{L}(\theta; \{x_i, y_i\}_{i=1}^N) = \frac{1}{N}\sum_{i=1}^N l(f(x_i), y_i),$$

where $l(\theta; x_i, y_i) = \frac{1}{2}\|y_i - f(\theta; x_i)\|^2$. We consider minimizing $\mathcal{L}(\theta)$ by gradient descent

$$\theta^{k+1} = \theta^k - \eta\nabla_\theta\mathcal{L}(\theta^k),$$

where $\eta > 0$, and $\theta^k$ is the vectorization of all parameters $\theta$ at the $k$th iteration.

We then introduce the following assumptions before stating the main theorem.

**Assumption 2** (Real-analyticity). *Assume $\varphi_\ell(\theta; u)$ are real-analytic (Definition 1) for all $\theta$, $u$, and $\ell = 1, \cdots, L$, and $\varphi_0(\theta; u)$ is real-analytic for all $\theta$, $u$ in some bounded open neighborhood of $\bar{\Omega}$.*

**Assumption 3** (Poly-boundedness, continuity, smoothness). *Assume for all $\ell = 0, \cdots, L$, $\varphi_\ell(\theta; u)$ satisfies polynomial generalized continuity (Definition 3) and smoothness (Definition 4) for both $\theta$ and $u$. In addition, $\varphi_\ell(\theta; u), \nabla_\theta\varphi_\ell(\theta; u), \nabla_u\varphi_\ell(\theta; u)$ are polynomially bounded (Definition 2).*

Assumption 2 and 3 hold true for many practical structures, including transformers, feedforward networks, and residual networks. See more discussions in Section 3.1 and 3.2.

**Assumption 4** (Nonlinear architecture). *Let $\tilde{f}_\ell(\theta; u_{\ell,i}) = f(\theta; x_i)$. Assume there exists some $\bar{\ell} \in \{0, \cdots, L-1\}$, s.t., for any $\theta_{\bar{\ell}}, \cdots, \theta_L$ except for a measure-zero set, there exists one tuple $(u_1, \cdots, u_N)$, s.t.,* $\begin{pmatrix} \nabla_{\theta_{\bar{\ell}}}\tilde{f}_{\bar{\ell}}(\theta; u_1) \\ \vdots \\ \nabla_{\theta_{\bar{\ell}}}\tilde{f}_{\bar{\ell}}(\theta; u_N) \end{pmatrix}$ *has full row rank $Nd$, and $\dim\theta_{\bar{\ell}} > Nd$.*

The above assumption is mild, and guarantees nonlinearity of the architecture through Theorem 7 (Mityagin, 2015). The $\tilde{f}_\ell(\theta; u_{\ell,i})$ is the composition of layers from the $(\ell+1)$th to the $L$th layer. It requires the row spans of the Jacobian of $\tilde{f}_{\bar\ell}(\theta; u_i)$'s do not intersect. The following is one example.

**Lemma 1.** *The following structure satisfies Assumption 4 with $\ell = L - 1$: $\varphi_L(\theta; u) = \theta_L u$ and $\varphi_{L-1}(\theta; u) = \theta_{L-1,2}\sigma(\theta_{L-1,1}u)$, where $\sigma(\cdot)$ is GELU, $\theta_L \in \mathbb{R}^{d\times d}$ $\theta_{L-1,1} \in \mathbb{R}^{m_{L-1}\times d}$, $\theta_{L-1,2} \in \mathbb{R}^{d\times m_{L-1}}$, and $m_{L-1} > Nd$.*

The above lemma considers an architecture in which the $(L-1)$th layer is a two-layer feedforward network, and the final layer is linear. It further shows that the width $m_{L-1} > Nd$ is sufficient to satisfy Assumption 4. Indeed, this lower bound of $m_{L-1}$ can potentially be improved in the proof of Lemma 1, since Assumption 4 only requires $m_{L-1} > N/2$. See Appendix C.3 for detailed proofs.

Below is the main theorem showing the convergence of neural networks (1) under GD.

**Theorem 2.** *Suppose Assumptions 1 to 4 hold. Consider the initialization $\theta^0$ in $\mathbb{R}^{\dim\theta}$ except for a measure-zero set. Let $R = \sqrt{\|\theta^0 - \theta^*\|^2 + \frac{4\rho+2}{\delta}\mathcal{L}(\theta^0)}$ for some stationary point $\theta^*$. Let $\eta$ be in some dense set of $\left(0, \min\left\{\frac{2-\delta}{\mathsf{L}}, 1\right\}\right)$ for some small constant $0 < \delta < 2$, and $\mathsf{L} = S\left(\cdots, \|\theta_i^*\| + R + \max_{\theta\in B_{\theta^*}(R)} \|\nabla_{\theta_i}\mathcal{L}(\theta)\|, \cdots\right)$, where $S(\cdot)$ is some polynomial whose coefficients are all positive. Assume $\mathcal{L}(\theta)$ satisfies the $(\theta^0, \theta^*, R, \rho, \epsilon_\mathcal{L})$-dissipative condition in Definition 5. Then, under some arbitrarily small adjustment on the scale of $\varphi_\ell$ for $\ell = 0, \cdots, L - 1$, with probability 1 over the joint distribution of the input data $x_1, \cdots, x_N$, GD converges to $\|\nabla\mathcal{L}(\theta)\| \le \epsilon_\mathcal{L}$, and*

$$\mathcal{L}(\theta^k) \le \prod_{s=0}^{k-1}\left(1 - \eta\left(1 - \frac{\eta\mathsf{L}}{2}\right)\frac{\mu_{\text{low},s,X}}{N}\right)\mathcal{L}(\theta^0), \ \forall k \ge 1, \ \text{and} \ \|\nabla\mathcal{L}(\theta^k)\| > \epsilon_\mathcal{L},$$

*where $\mu_{\text{low},s,X} > 0$ depends on $\theta^s$ and $x_1, \cdots, x_N$.*

In the above theorem, the loss of GD is proved to decay until it converges to the neighborhood of a stationary point, under practical assumptions. A complete version of the above theorem can be found in Theorem 5. Please see Appendix E for detailed proofs.

The $\mathsf{L}$ is the Lipschitz smoothness constant along the GD trajectory, and is proved to be bounded instead of assumed. It depends on the poly-smoothness function $S(\cdot)$ of $\mathcal{L}(\theta)$ (Lemma 7), the initial condition $\theta^0$, initial loss $\mathcal{L}(\theta^0)$, maximum $\nabla\mathcal{L}(\theta)$ in some bounded region, and some stationary point $\theta^*$ and its landscape nearby determined solely by $\theta^0$. The $\theta^*$ can be any stationary point that satisfies the dissipativity condition (Definition 5), and is not necessarily the limit of GD iterations. The radius $R$ of the bounded region for local relaxed dissipativity is determined by the initialization $\theta^0$ and does not assume any information about the subsequent GD iterations. The constant $\mu_{\text{low},s,X}$ is the lower frame bound of the derivatives of $(f(x_1)^\top, \cdots, f(x_N)^\top)$ which is proven to form a frame with probability 1 for almost all $\theta$, and consequently $\mu_{\text{low},s,X} > 0$ (see more details in Appendix E.1).

The unusual density requirement on the learning rate $\eta$ arises from the parametric transversality theorem (Theorem 6 (Hirsch, 2012), Lemma 11). The excluded "bad" values of $\eta$ may cause convergence to a degenerate map with non-full-rank Jacobian, reducing expressivity by restricting the image to a lower-dimensional subspace. Fortunately, such cases can be avoided by a small perturbation of $\eta$, which suffices to yield a converging $\eta^1$. The small adjustment to the scale of $\varphi_\ell$ follows the same reasoning to prevent degeneracy, and only needs to be done once before GD iterations.

Based on the convergence of the neural network, we derive the following corollary, which highlights the relationship between data pairwise distance, particularly $h_{\min}$, and the convergence rate.

**Corollary 3.** *Under the same assumptions as Theorem 2, fix some $x_i$. Let $X = (x_1, \cdots, x_N)$, and $\mathcal{D}_{i,H} = \{j \mid h_{ij} = \|x_i - x_j\| \le H, \ \forall j \ne i\}$. Then for any $k \ge 1$, there exists $r_{k,i,H} > 0$ and $L_{k,i,H} > 0$, s.t., when $\sqrt{\sum_{j\in\mathcal{D}_{i,H}} h_{ij}^2} \le r_{k,i,H}$, we have*

$$\mu_{\text{low},\text{s},\text{X}} \le L_{k,i,H}\sqrt{\sum_{j\in\mathcal{D}_{i,H}} h_{ij}^2}, \ \forall s \le k.$$

---

[1]The set of $\eta$ in Theorem 2 is residual, i.e., roughly a "large" set in $\left(0, \min\left\{\frac{2-\delta}{\mathsf{L}}, 1\right\}\right)$, compared to its complement (see Appendix E.3).

Figure 2: Proof sketch of the convergence framework beyond the NTK regime. Assumptions are highlighted in blue, key lemmas in brown, and the statements or implication of lemmas in grey.

*Specifically, if $H = h_{\min}$ and $\mathcal{D}_{i,H} \neq \varnothing$, there exists $L_{k,i}, r_{k,i} > 0$, s.t., when $h_{\min} \leq r_{k,i}$, we have $\mu_{\mathrm{low},s,X} \leq L_{k,i} h_{\min}, \ \forall s \leq k$.*

The above corollary shows that, for a fixed data point $x_i$, when nearby points become even closer to $x_i$, the training process potentially slows down, which is verified experimentally in Section 5. More precisely, there exists a subsequence of $\{\mu_{\mathrm{low},s,X_t}\}_t$ depending on the data $X_t$, s.t., as the nearby distance $\sqrt{\sum_{j \in \mathcal{D}_{i,H}} h_{ij}^2}$ decreases for some $X_t$, the value of $\mu_{\mathrm{low},s,X_t}$ also monotonically decreases. Consequently, the value of $1 - \eta\left(1 - \frac{\eta L}{2}\right)\frac{\mu_{\mathrm{low},s,X_t}}{N}$ increases, and thus leads to slower convergence. The dependence on $h_{\min}$ is a special case of this phenomenon and follows the same reasoning.

**Remark 1** (Convergence beyond the NTK regime). *Theorem 2 provides a general framework of convergence for a family of neural networks beyond the NTK regime in the following two senses, both of which concern the local relaxed dissipativity and the learning rate. (1) When the loss landscape is unfavorable around the initial condition $\theta^0$ (e.g. $\rho = \frac{R}{\epsilon_{\mathcal{L}}}$ which holds for any landscape; see Section 3.3), which may indicate insufficient overparameterization, this setting naturally lies outside the NTK regime. (2) When the loss landscape is favorable near $\theta^0$, i.e., $\rho$ is small, this may imply a large width of the neural network. However, the learning rate $\eta$ in Theorem 2 can be much larger than the infinitesimal (gradient flow) regime assumed in the NTK analysis, due to a small $\rho$ and the independence of width if $\theta^0$ is properly scaled. This means that the NTK regime can be covered by our analysis, i.e., when the neural network is in the NTK regime, our analysis then guarantees convergence to a neighborhood of a global minimum. See Appendix B for more detailed discussions.*

**Remark 2** (Theoretical insights on architectures: residual connection, and function composition). *When constructing a neural network, it is important to ensure that the model is expressive enough. Theoretically, this corresponds to requiring non-degeneracy of the neural network $f$, i.e., its Jacobian $\nabla f$ should be full rank, avoiding loss of full expressivity. In fact, non-degeneracy is a generic property among real-analytic functions. If a function is degenerate, we can typically restore non-degeneracy by adding a small identity component, i.e., by incorporating a residual connection (see Lemma 11). This helps explain the empirical success of residual architectures in practice. Moreover, since neural networks are compositions of layers, if each layer has a Jacobian with full rank (e.g., via residual connections), then by chain rule the Jacobian of the overall composition remains full rank, preserving non-degeneracy. See Appendix B for more detailed discussions.*

The proof sketch of Theorem 2 is shown in Figure 2. The proof of Corollary 3 mainly relies on the local Lipschitz continuity of eigenvalues for real-analytic matrix functions (see Theorem 4.1 in Kurdyka & Paunescu (2008)). Detailed proofs are in Appendix E.

### 4.3 APPROXIMATION: DATA-DEPENDENT ERROR IN THE INTERPOLATION REGIME

In this section, we show that a larger $h_{\min}$ and smaller local maximal distances lead to smaller approximation error in the interpolation regime $\mathcal{I} = \mathrm{conv}\{x_1, \cdots, x_N\}$.

The following theorem characterizes the data-dependent approximation error.

**Theorem 3.** *Consider a set of data $\{(x_i, y_i)\}_{i=1}^N$ under Assumption 1. Let $\mathcal{I}_i = \mathrm{conv}\{x_{i_1}, \cdots, x_{i_{d+1}}\}$ with $x_j \notin \mathcal{I}_i, \ \forall j \neq i_1, \cdots, i_{d+1}$, and define $h_{\max_{d+1}} = \max_i \max_{x_q, x_j \in \mathcal{I}_i} h_{qj}$. Assume $h_{\min} \leq 1 \leq h_{\max_{d+1}}$. Suppose there is a family of neural networks that can approximate polynomials of degree $n_d = n_d(N, d)$, i.e., for any $\epsilon_2 > 0$ and $\psi \in P_{n_d}$, there exists some $f$ s.t. $\|f - \psi\|_p \leq \epsilon_2$. Then, for some $C_1 > 0$ and $m > \max\{r, n_d\}$, there exists some $\phi \in W^{m,p}(\Omega)$ satisfying $\|g - \phi\|_{r,p} \leq C_2 h_{\max_{d+1}}^m h_{\min}^{-r} \|\phi\|_{m,p,\mathcal{I}^\circ}$, s.t., with probability 1 over the joint distribution of $x_1, \cdots, x_N$, we have*

$$\|f - g\|_{p,\mathcal{I}^\circ} \leq C_1 h_{\max_{d+1}}^m h_{\min}^{-r} \|\phi\|_{m,p,\mathcal{I}^\circ}.$$

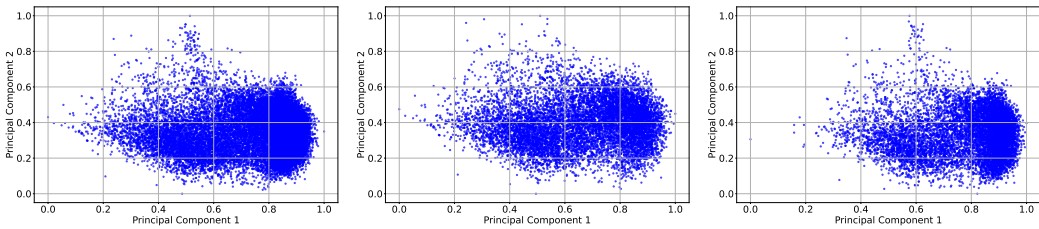

(a) 20k Full data points.  (b) 10k Uniform data points (Ours).  (c) 10k Random data points.

Figure 3: Visualization of data point distributions for datasets selected using different methods from WizardLM (Xu et al., 2024).

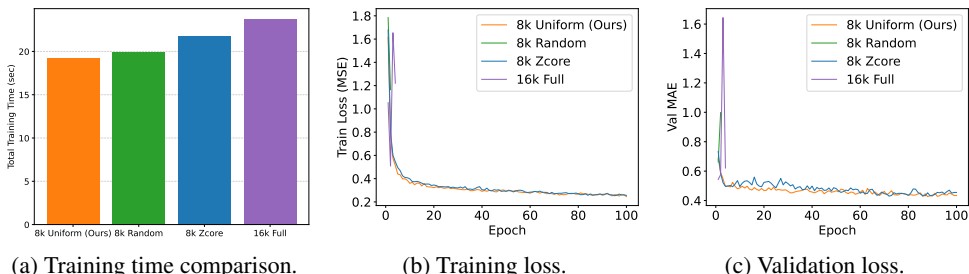

(a) Training time comparison.  (b) Training loss.  (c) Validation loss.

Figure 4: L2-SGD training on the California Housing dataset (Pace & Barry, 1997). Training time is measured as the total time required to complete the same number of epochs (e.g., 100). Uniform subsets (ours) achieve faster, smoother, and more stable convergence with fewer samples, matching or surpassing Zcore (Griffin et al., 2024) and mitigating gradient instabilities seen in random and full datasets.

The above theorem differs from classical results in approximation theory. It fixes a specific approximation function and examines how the distances between data points affect its error. More precisely, the error bound is determined mainly by two quantities: a local maximum distance $h_{\max_{d+1}}$, and a minimum distance $h_{\min}$. It then indicates that using fewer but more uniformly distributed data points, i.e., increasing $h_{\min}^2$, can achieve similar approximation error as increasing the number of data points, i.e., decreasing $h_{\max_{d+1}}$. This is also validated in our experiments (see Section 5).

The proof mainly involves establishing a data-dependent Bramble-Hilbert lemma (Bramble & Hilbert, 1970) for each $d-$simplex $\mathcal{I}_i$ in the Delaunay triangulation. See a formal version of Theorem 3 and detailed proofs in Appendix F.

## 5  EXPERIMENTS

In our experiments, we introduce a data uniformity strategy that iteratively selects points maximizing the minimum pairwise distance $h_{\min}$ (Algorithm 1), and visualize differences between uniform subsets, random subsets of equal size, and the full datasets. Comparison to random subsets isolates the effect of $h_{\min}$ while holding dataset size fixed. We then perform supervised fine-tuning with two circumstances: $\ell^2$ loss with SGD for theoretical validation and cross-entropy loss with Adam for practical evaluation. This design enables systematic assessment of performance and training efficiency across datasets, scales, and model sizes. We further deploy our method and compare with SOTA baselines, including TeaMs-RL (Gu et al., 2024), WizardLM

> **Algorithm 1:** Data uniformity strategy
> **Input:** Full dataset $\{x_i\}_{i=1}^N$, target size $\tilde{N}$
> **Output:** Subset $S$ with $|S| = \tilde{N}$
> 1: // initialize selected set
> 2: $i \leftarrow 1$; randomly pick $x^*$ from $\{x_i\}_{i=1}^N$
> 3: $S \leftarrow \{x^*\}$
> 4: **while** $i < \tilde{N}$ **do**
> 5:    // maximize min distance
> 6:    $\tilde{x}_i \leftarrow \arg \max_{x \in \{x_j\}_{j=1}^N \setminus S} \min_{y \in S} \mathrm{dist}(x, y)$
> 7:    $S \leftarrow S \cup \{\tilde{x}_i\}$
> 8:    $i \leftarrow i + 1$
> 9: **end while**

(Xu et al., 2024), Zcore (Griffin et al., 2024), and LESS (Xia et al., 2024). Results show that uniform subsets accelerate convergence (Corollary 3) and maintain comparable or superior performance relative to both full datasets and strong baselines (Theorem 3). Full experimental settings appear in Appendix J.

---

[2]Larger $h_{\min}$ also contributes to smoother approximation as $r$ specifies the order of smoothness in $W^{r,p}(\Omega)$.

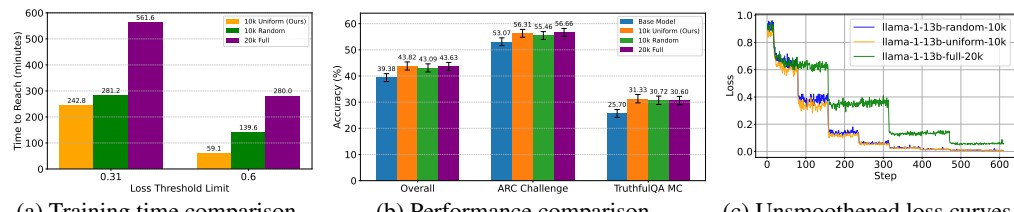

(a) Training time comparison.     (b) Performance comparison.     (c) Unsmoothened loss curves

Figure 5: Cross-entropy training with AdamW on the WizardLM dataset (Xu et al., 2024) using LLaMA-1-13B (Touvron et al., 2023). Uniform subsets (ours, via $\ell^2$ distance) (a) reach target loss thresholds in shorter run time, (b) achieve comparable or superior ARC Challenge and TruthfulQA MC accuracies, and (c) have faster loss decay, showing efficiency of uniformity-aware selection.

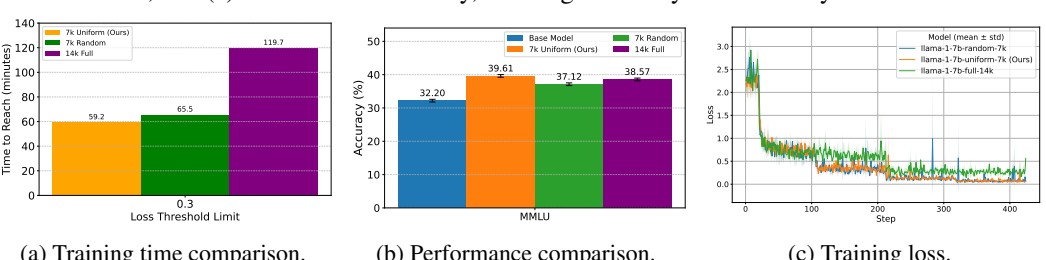

(a) Training time comparison.     (b) Performance comparison.     (c) Training loss.

Figure 6: Cross-entropy training with AdamW on the LESS dataset (Xia et al., 2024) using LLaMA-1-7B (Touvron et al., 2023). Uniform subsets (ours, via cosine distance) (a) reach target loss thresholds faster, (b) achieve comparable or superior MMLU accuracy, and (c) show smoother, more stable convergence than random or full datasets, demonstrating efficiency of uniformity-aware selection.

## 5.1 DATA SELECTION AND DATA VISUALIZATION

We propose a greedy distance-maximization strategy to build compact yet representative instruction-tuning datasets (Algorithm 1; see also Tan et al. (2023) for a related use of uniformity not using $h_{\min}$, and Ethayarajh (2019) for discussions on uniformity). Each data point is encoded via Word2Vec embeddings, and points are iteratively chosen to maximize the minimum Euclidean or cosine distance, ensuring broad coverage. Applied to TeaMs-RL (Gu et al., 2024), WizardLM (Xu et al., 2024), Zcore (Griffin et al., 2024), and LESS (Xia et al., 2024), we construct balanced subsets (e.g., 4.5k from 9k TeaMs-RL, 10k from 20k WizardLM) for fair comparison with random baselines. PCA visualizations (Maćkiewicz & Ratajczak, 1993) reveal that uniform subsets achieve broader, less redundant coverage than random or full datasets, highlighting their effectiveness in preserving semantic diversity with fewer samples (see Appendix H.1).

## 5.2 $\ell^2$-SGD TRAINING EXPERIMENTS

We validate our theoretical insights under $\ell^2$ loss with SGD on the California Housing dataset (Pace & Barry, 1997), comparing our uniform subsets with random, Zcore (Griffin et al., 2024), and full datasets. Zcore is an SOTA coreset selection method that leverages foundation model embeddings to compute importance scores, producing compact and representative subsets that have been shown to outperform several strong baselines (Griffin et al., 2024). As shown in Figure 4, random and full datasets exhibit divergence, uniform subsets achieve comparable or better performance than Zcore with shorter training time, measured as the total time to complete the same number of epochs (e.g., 100), while avoiding gradient instabilities. Consistent results across other datasets, model sizes, and baselines such as TeaMs-RL and WizardLM (Appendix H.2) confirm the efficiency of uniformity-based selection, supporting theory (Section 4) and demonstrating improved efficiency and stability.

## 5.3 CROSS ENTROPY LOSS ADAM TRAINING EXPERIMENTS

Beyond $\ell^2$-SGD, we also conduct experiments using cross-entropy loss with Adam, the standard optimization setup for LLMs (Gunel et al., 2021; Mao et al., 2023), and observe consistent improvements in the two settings: LLaMA-13B on the WizardLM dataset (Figure 1 where data is selected using cosine distance, and 5 with Euclidean distance) and LLaMA-1-7B on the LESS dataset (Xia et al., 2024) (Figure 6). The LESS dataset was originally constructed via gradient-based selection for MMLU (Xia et al., 2024), and we further reapply our method on this preselected data. Results show that uniformity-aware selection can further enhance the performance of LESS. Figure 1(a), 5(a),

and 6(a) demonstrates that the 10k and 7k uniform subsets reach the loss threshold more than 2× faster than the corresponding full datasets, and also faster than the corresponding random subsets of the same sizes, improving training efficiency. Figure 1(b), 5(b), and 6(b) show that, at matched loss checkpoints (ensuring fair evaluation across methods), the uniform subset achieves accuracy comparable to or exceeding random and full datasets while requiring less training time. Additionally, Figure 6 illustrates smoother and more stable convergence (less spikes) across seeds. Consistent findings are also observed on TeaMs-RL and WizardLM across different dataset scales, data selection distances (e.g., cosine distance), model sizes, and sources (see Appendix H.3).

## 6 DISCUSSION AND CONCLUSION

We remark that using different distances in Algorithm 1 offer distinct benefits on WizardLM. The training speed of using cosine distance is faster than using Euclidean distance (Figure 1(a) vs Figure 5(a)), while the generalization of using Euclidean distance is better than cosine distance (Figure 5(b) vs (Figure 1(b)). We will leave it for future exploration. In sum, we demonstrate the importance of data uniformity, selecting more uniformly distributed data, through both theory and experiments. Theoretically, we show that higher uniformity, i.e., larger minimum pairwise distance $h_{\min}$, can accelerate training and reduce approximation error. We also develop a general convergence framework for complicated neural networks beyond the NTK regime, and it also highlights how residual connection and function composition preserve expressivity. Empirically, across optimizers ($\ell^2$-SGD, Adam), model sizes (LLaMA-1 7B, 13B), and datasets (TeaMs-RL, WizardLM, ZCore, LESS), uniform selection consistently accelerates convergence and achieves accuracy comparable to or better than full datasets with fewer training time, improving both training loss and downstream tasks such as ARC and TruthfulQA. These findings emphasize data uniformity as an effective strategy for efficient and scalable LLM training.

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

APPENDIX

**Contents**

## A    MORE RELATED WORK

**Data Diversity.** Data diversity has been explored across a wide range of domains for enhancing generalization, efficiency, and robustness. For example, Muennighoff et al. (2025) investigated the role of data diversity in improving test-time scaling generalization. In WizardLM (Xu et al., 2024), diverse queries are obtained by sampling inputs randomly for distillation from larger LLMs. TeaMs-RL (Gu et al., 2024) leveraged reinforcement learning (RL) to teach LLMs for diverse instructional data generalization, improving downstream model generalization. Beyond language modeling, data diversity has shown benefits for generalization in supervised learning (Yu et al., 2022), software security (Nguyen-Tuong et al., 2008), fault tolerance in software systems (Ammann & Knight, 2002), and medical image segmentation (Hofmanninger et al., 2020). Particularly, in RL, diverse data supports more efficient offline training (Nguyen-Tang & Arora, 2023) and improves exploration and policy learning (Eysenbach et al., 2019). Other domains, such as mobile infostation networks (Yuen et al., 2003) and privacy protection (Machanavajjhala et al., 2007), also leverage diversity for improved robustness. More broadly, Gong et al. (2019) demonstrated that data diversity enhances machine learning performance across various stages, such as model diversification and inference diversification.

**Uniformity on the hypersphere.** Different from our Euclidean distance $h_{\min}$, there is a line of empirical work studying the uniformity on the hypersphere. (Wang & Isola, 2020) illustrated the importance of uniformity on the sphere in representation learning, which captures the distances between angles. Wang et al. (2022a) later on showed that uniformity on hypersphere contributes to higher recommendation performance. Wang et al. (2023a) tested the effect of uniformity on inference tasks. Additional work related to uniformity on the hypersphere includes Tan et al. (2023); Wang et al. (2024); Ouyang et al. (2024); Sang et al. (2025); Cai et al. (2024); Son et al. (2024), etc.

**Residual Connections.** Residual connections have been widely studied for their theoretical and practical benefits in deep learning. Hardt & Ma (2016) showed that deep linear residual networks exhibit no spurious local optima and maintain strong expressivity, a result further supported by Liu et al. (2019), who demonstrated the absence of spurious minima in residual architectures. Huang et al. (2020) highlighted the role of residual connections in preserving learnability across depth, while Scholkemper et al. (2024) found that they help mitigate oversmoothing in graph neural networks. In the context of transformers, Qin et al. (2025) showed that residual connections improve the conditioning of output matrices in single-layer architectures with feedforward network and attention. In contrast to these works, our study analyzes general and more complex architectures and provides a theoretical explanation based on measure theory and differential topology for the intrinsic non-degeneracy of the layerwise mapping induced by residual connections throughout training.

**Approximation of Neural Networks.** A lot of literature has investigated the universal approximation properties of neural networks through the lens of Sobolev spaces. Early work by Andoni et al. (2014) considered the learnability of a two-layer neural network with analytic activation functions. Building on classical approximation theory, Lu et al. (2021) applied the Bramble–Hilbert theorem to characterize the expressivity of deep ReLU networks for smooth functions, deriving error bounds in the $L^\infty$ norm. Extending the analysis to Sobolev norms, Gühring et al. (2020) established approximation rates for ReLU networks in $W^{s,p}$. Similarly, De Ryck et al. (2021) derived approximation error bounds in $W^{k,\infty}$ norm for tanh networks, while Shen et al. (2022) focused on convolutional architectures and their ability to approximate functions in Sobolev spaces. More recently, Jiang & Li (2024) analyzed the approximation capabilities of transformer architectures in the $L^\infty$ norm. Regarding the general $L^p$ norm for $1 \leq p \leq \infty$, there are works studying, for example, transfromers Yun et al. (2019); Kajitsuka & Sato (2023); Kratsios et al. (2021); Edelman et al. (2022); Luo et al. (2022), residual networks (ResNets) Lin & Jegelka (2018); Tabuada & Gharesifard (2022), feedforward networks (Hanin & Sellke, 2017; Kidger & Lyons, 2020; Park et al., 2020; Cai, 2022, etc.).

## B    MORE DISCUSSIONS ON THE CONVERGENCE RESULTS

Below are more detailed versions of Remark 1 and 2.

**Remark 3** (Convergence beyond the NTK regime). *Theorem 2 provides a general framework of convergence for a family of neural networks beyond the NTK regime in the following two senses, both of which concern the local relaxed dissipativity and the learning rate:*

*(1) When the loss landscape is unfavorable around the initial condition (for example, in the worst case $\rho = \frac{R}{\epsilon}$ for some $\epsilon > 0$, which holds for any landscape; see Section 3.3), it indicates insufficient overparameterization, meaning the width of the neural network is too small, and therefore lies outside the NTK regime. Moreover, a large $\rho$ leads to large $\mathsf{L}$, which may imply small learning rate $\eta$. However, different from the NTK regime, $\eta$ does not necessarily decrease as the width increases, since it depends mainly on the initialization, and with proper scaling, it can be independent of width.*

*(2) When the loss landscape is favorable near the initial condition, i.e., $\rho$ is small, this may imply a large width of the neural network. However, the learning rate $\eta$ in Theorem 2 can be much larger than that required by the infinitesimal (gradient flow) regime assumed in the NTK analysis, due to a small $\rho$ and the independence of width as discussed in (1).*

*Apart from the local relaxed dissipativity condition, the only width requirement appears in Assumption 4. As an example of the Assumption 4, Lemma 1 shows that it suffices for the width of the $(L-1)$th feedforward layer to be greater than $Nd$; no constraints are imposed on the widths of other layers. This lower bound $Nd$ on a single layer, which can potentially be improved (see more discussions below Lemma 1), is much milder than the NTK-type convergence conditions, which typically require the widths of all layers to scale as high-degree polynomials in $d, N, L$, etc. (see Section 2). We also remark that the NTK regime is covered by our analysis. More precisely, if the width is large enough, i.e., the neural network is in the NTK regime, our analysis then guarantees convergence to a neighborhood of a global minimum.*

**Remark 4** (Theoretical insights on architectures: residual connection, composition, and normalization). *When constructing a neural network, it is important to ensure that the model is expressive enough. Theoretically, this corresponds to requiring the neural network mapping $f$ to be non-degenerate, i.e., its Jacobian $\nabla f$ should be full rank in some region. Otherwise, the mapping may collapse onto a lower-dimensional subspace, and thus lose full expressivity.*

*In fact, non-degeneracy is a generic property among real-analytic functions. If a function is degenerate, we can typically restore non-degeneracy by adding a small identity component, i.e., by incorporating a residual connection, and then the modified mapping is non-degenerate (see Lemma 11). This helps explain the empirical success of residual architectures in practice.*

*Moreover, neural networks are constructed as compositions of layer-wise transformations. If each layer has a Jacobian with full rank (e.g., ensured via residual connections), then the Jacobian of the overall composition remains full rank due to the chain rule: $\nabla(f \circ g)(x) = \nabla f(g(x)) \nabla g(x)$. This shows that the compositional (iterative) structure of neural networks preserves non-degeneracy. In contrast, other combinations, such as summations of functions, do not necessarily preserve this property, even if each individual function is non-degenerate. For example, consider $f_1(x) = x^\top x$ and $f_2(x) = -x^\top x$; while each is non-degenerate, their sum $f_1 + f_2 = 0$ is degenerate.*

*Regarding the normalization used in (1), it differs from practical schemes such as batch normalization and layer normalization. Nonetheless, they share a common goal, which is to reduce the magnitude of neurons and restore desirable regularity properties in the network. Such normalization helps flatten the optimization landscape, thereby facilitating more stable and efficient training. See more discussions in Wang et al. (2023b).*

## C PRELIMINARY PROPERTIES AND ASSUMPTION JUSTIFICATION

### C.1 ANALYTIC FUNCTION

Real-analyticity is preserved under many operations. Specifically:

**Proposition 1.** *Real-analytic functions have the following properties:*

*1. The sums, products, divisions where the denominators are not zero, and compositions of real-analytic functions are real-analytic.*

*2. The derivative and integral of real-analytic functions are real-analytic.*

*3. Real-analytic function is $C^\infty$.*

We then prove that a lot of neural network architectures are real-analytic.

*Proof of Corollary 1.* Since exponential function admits Taylor series in some neighbourhood of any point in $\mathbb{R}^d$, and the denominators of softmax and sigmoid functions are strictly positive at any point, by Proposition 1, we have that softmax, sigmoid, GELU, SiLU are analytic functions on $\mathbb{R}^d$.

Similarly, polynomial functions admit Taylor series in some neighbourhood of any point in $\mathbb{R}^d$, and the denominator of $\tau_\epsilon$ is strictly positive at any point, we have $\tau_\epsilon$ and polynomial functions are analytic.

Also, tanh can be Taylor expanded at any point and thus analytic.

By Proposition 1, any product, sum, compositions of the above functions are still analytic. $\qquad\square$

### C.2 Polynomial boundedness, polynomial generalized continuity and smoothness

Polynomial boundedness, polynomial generalized continuity and smoothness are preserved under the following operations:

**Proposition 2.** *Assume $f$ and $g$ satisfies polynomial boundedness for both the functions and their gradients, polynomial generalized continuity, and polynomial generalized smoothness. Then*

*1. $f(x) + g(x)$, where $f, g : \mathbb{R}^d \to \mathbb{R}^m$,*

*2. $f(x)g(x)$, where $f \in \mathbb{R}^{m \times d}$, and $g \in \mathbb{R}^{d \times n}$,*

*3. $(f \circ g)(x)$, where $f : \mathbb{R}^d \to \mathbb{R}^m$, and $g : \mathbb{R}^n \to \mathbb{R}^d$,*

*satisfy the same properties under different polynomials.*

*Proof.* Let $S_{f,0}$, $S_{g,0}$ be the corresponding polynomials in polynomial generalized continuity of $f$ and $g$, $S_{f,1}$, $S_{g,1}$ be the ones in polynomial generalized smoothness, $S_f$, $S_g$ be the corresponding polynomials in polynomial bound, and $S_{\nabla f}$, $S_{\nabla g}$ be the corresponding polynomials in polynomial bound of their gradients. For notational simplicity, we use $S(x)$ to denote $S(\|x_1\|, \cdots, \|x_d\|)$, and $S(x_{\max})$ to denote $S(\|x_{\max,1}\|, \cdots, \|x_{\max,d}\|)$.

Then

$$\|f(x) + g(x) - f(x') - g(x')\| \le \|f(x) - f(x')\| + \|g(x) - g(x')\| \le (S_{f,0} + S_{g,0})(x_{\max})\|x - x'\|.$$

Similarly, $\nabla f + \nabla g$ follows the same idea with $(S_{f,1} + S_{g,1})(x_{\max})$. Moreover, $f + g$ is upper bounded by $S_f(x_{\max}) + S_g(x_{\max})$, and $\nabla f + \nabla g$ is upper bounded by $S_{\nabla f}(x_{\max}) + S_{\nabla g}(x_{\max})$.

Also,

$$\begin{aligned}
\|f(x)g(x) - f(x')g(x')\| &= \|f(x)g(x) - f(x)g(x') + f(x)g(x') - f(x')g(x')\| \\
&\le \|f(x)\|\|g(x) - g(x')\| + \|f(x) - f(x')\|\|g(x')\| \\
&\le (S_f(x)S_{g,0}(x_{\max}) + S_{f,0}(x_{\max})S_g(x'))\|x - x'\| \\
&\le (S_f(x_{\max})S_{g,0}(x_{\max}) + S_{f,0}(x_{\max})S_g(x_{\max}))\|x - x'\|.
\end{aligned}$$

Obviously, $(S_f(x_{\max})S_{g,0}(x_{\max}) + S_{f,0}(x_{\max})S_g(x_{\max}))$ is a polynomial with positive coefficients, and therefore $fg$ satisfies polynomial generalized continuity.

Based on the above derivation, $\nabla(fg) = \nabla f \cdot g + f\nabla g$ is a sum of two product, and thus satisfies polynomial continuity with $(S_{\nabla f}S_{g,0} + S_{f,1}S_g + S_{\nabla g}S_{f,0} + S_{g,1}S_f)$.

Also, $fg$ is upper bounded by $S_f S_g$, and $\nabla(fg) = \nabla f \cdot g + f\nabla g$ is upper bounded by $S_{\nabla f}S_g + S_{\nabla g}S_f$.

Next, since all the $S$ have positive coefficients, we have

$$\begin{aligned}
\|f(g(x)) - f(g(x'))\| &\le S_{f,0}(\cdots, \|g(x)_{\max,i}\|, \cdots)\|g(x) - g(x')\| \\
&\le S_{f,0}(S_g(x_{\max}), \cdots, S_g(x_{\max}))S_{g,0}(x_{\max})\|x - x'\|.
\end{aligned}$$

Also, $\nabla_x f(g(x)) = \nabla_g f(g(x)) \nabla g(x)$. Similar to the above derivations, we have $\nabla_x f(g(x))$ is polynomial continuous with $S_{f,1}(S_g(x_{\max}), \cdots, S_g(x_{\max})) S_{g,0}(x_{\max}) S_{\nabla g}(x_{\max}) + S_{\nabla f}(S_f(x_{\max}), \cdots, S_f(x_{\max})) S_{g,1}(x_{\max})$.

In the end, it can be shown that $f(g(x))$ is upper bounded by $S_f(S_g(x_{\max}), \cdots, S_g(x_{\max}))$, and $\nabla f(g(x))$ is upper bounded by $S_{\nabla g}(x_{\max}) S_{\nabla f}(S_g(x_{\max}), \cdots, S_g(x_{\max}))$. ☐

**Corollary 4.** *The functions, polynomials, softmax, tanh, sigmoid, $\tau_\epsilon$, GELU, SiLU, and the product, sum, compositions of them, satisfies polynomial boundedness for both the functions and their gradients, polynomial generalized continuity, and polynomial generalized smoothness.*

*Proof.* By Proposition 2, we only need to show that polynomials, softmax, tanh, sigmoid, $\tau_\epsilon$, GELU, SiLU satisfy all the properties.

First, polynomials obviously satisfies all the properties. It can be shown by simple calculations that softmax, tanh, sigmoid, $\tau_\epsilon$, and their derivatives are all upper bounded by constant, and therefore satisfies all the above properties.

For SiLU, it is the product of $x$ and sigmoid, and therefore by Proposition 2 satisfies all the above properties.

For GELU, $\sigma(x) = x\Phi(x)$, where $\Phi(x) = \int_{-\infty}^{x} \frac{e^{-s^2/2}}{\sqrt{2\pi}} ds$. Also, $\Phi'(x) = \phi(x) = \frac{e^{-x^2/2}}{\sqrt{2\pi}}$. Then $\Phi(x)$ and $\Phi'(x)$ are both upper bounded by constants, and thus satisfies all the above properties. By Proposition 2, GELU, the product of $x$ and $\Phi(x)$, also satisfies these properties. ☐

Similar to Lipschitz smoothness and generalized smoothness (Li et al., 2023), the polynomial generalized smoothness has another interpretation as follows:

**Lemma 2.** *If the function $f(x) \in C^1$ satisfies the polynomial generalized smoothness, then the following inequalities hold*

$$f(x') \le f(x) + \nabla f(x)^\top (x' - x) + \frac{S(\|x_{\max,1}\|, \cdots, \|x_{\max,d}\|)}{2} \|x - x'\|^2 \tag{2}$$

*Proof.* Since $f(x) \in C^1$ satisfies the polynomial generalized smoothness, we have

$$\|\nabla f(x) - \nabla f((1-t)x + tx')\| \le S(\cdots, \max\{\|x_i\|, \|(1-t)x_i + tx_i'\|\}, \cdots)\|x - ((1-t)x + tx')\|$$
$$\le t\, S(\cdots, \max\{\|x_i\|, \|x_i'\|\}, \cdots)\|x - x')\|$$
$$= t\, S(\|x_{\max,1}\|, \cdots, \|x_{\max,d}\|)\|x - x'\|$$

Then

$$f(x') - f(x) = \int_0^1 \nabla f((1-t)x + tx')^\top (x' - x) dt$$
$$= \int_0^1 \nabla f(x)^\top (x' - x) dt + \int_0^1 \left(\nabla f((1-t)x + tx') - \nabla f(x)\right)^\top (x' - x) dt$$
$$\le \nabla f(x)^\top (x' - x) + \|x - x'\| \int_0^1 \|\nabla f((1-t)x + tx') - \nabla f(x)\| dt$$
$$\le \nabla f(x)^\top (x' - x) + S(\|x_{\max,1}\|, \cdots, \|x_{\max,d}\|)\|x - x'\|^2 \int_0^1 t\, dt$$
$$= \nabla f(x)^\top (x' - x) + \frac{S(\|x_{\max,1}\|, \cdots, \|x_{\max,d}\|)}{2} \|x - x'\|^2$$

☐

### C.2.1 COMPARISON BETWEEN POLY-SMOOTHNESS AND GENERALIZED SMOOTHNESS

The differences between the above poly-smoothness and generalized smoothness (Li et al., 2023) lies in two parts. (1) Li et al. (2023) defined the bound of Hessian first, and therefore evaluate the smoothness function at one fixed point for any two points in its neighbourhood, while we directly consider the continuity of the gradient and evaluate the smoothness function $S(\cdot)$ at the two points. (2) Regarding the smoothness function, Li et al. (2023) employed a function of $\|\nabla f(x)\|$, while we use a polynomial of $\|x_i\|, \|x_i'\|$. Note such $S(\cdot)$ is monotonically increasing in $\mathbb{R}_{\ge 0} \times \cdots \times \mathbb{R}_{\ge 0}$.

### C.3 EXAMPLE OF ASSUMPTION 4

Assumption 4 is used in the convergence of neural networks. Before introducing the convergence proof, we first verify that the Assumption 4 can be easily satisfied:

*Proof of Lemma 1.* First, we compute the Jacobian of the neural network w.r.t. $\theta_{L-1,1}$

$$\nabla_{\theta_{L-1,1}} f(\theta; x_i)$$
$$= \nabla_{\tau_\epsilon} \varphi_L(\tau_\epsilon(u_{L,i})) \nabla_u \tau_\epsilon(u_{L,i}) \nabla_{\theta_{L-1,2}} \bar{\varphi}(u_{L-1,i})$$
$$= \theta_L \underbrace{\left( \frac{1}{\sqrt{\|u_{L,i}\|^2 + \epsilon^2}} I - \frac{1}{(\sqrt{\|u_{L,i}\|^2 + \epsilon^2})^3} u_{L,i} u_{L,i}^\top \right)}_{P_i} \underbrace{\begin{pmatrix} \sigma(\theta_{L-1,1}u_{L-1,i})^\top & & \\ & \ddots & \\ & & \sigma(\theta_{L-1,1}u_{L-1,i})^\top \end{pmatrix}}_{Q_i}$$

where in $\nabla_{\theta_{L-1,2}} \bar{\varphi}(u_{L-1,i})$, we vectorize $\theta_{L-1,2}$ in row.

Also, $\sigma(x) = x\Phi(x)$, where $\Phi(x) = \int_{-\infty}^x \frac{e^{-s^2/2}}{\sqrt{2\pi}} ds$. Let $\phi(x) = \frac{e^{-x^2/2}}{\sqrt{2\pi}}$. Then we have

$$\sigma'(x) = \Phi(x) + x\phi(x)$$
$$\sigma^{(n)}(x) = 2\phi^{(n-2)}(x) + \phi^{(n-1)}(x)$$
$$= \left( 2(-1)^{n-2} H_{n-2}(x) + (-1)^{n-1} H_{n-1}(x) \right) \phi(x)$$

where $H_n(x)$ is the probabilist's Hermite polynomial.

By Taylor expansion,

$$\sigma(v^\top u + a) = \sum_{n=0}^{\infty} \frac{\sigma^{(n)}(a)}{n!} (v^\top u)^n$$

Consider $\sigma(t\theta u)$ near $t = 0$. Choose $0 < t_1 < t_2 < \cdots < t_N \ll 1$. For probabilist's Hermite polynomials,

$$H_{2n}(0) \neq 0, H_{2n-1}(0) = 0, \ \forall n \geq 1.$$

Thus,

$$\sigma(0) = 0, \ \sigma^{(n)}(0) \neq 0, \forall n \geq 1.$$

Let $\theta_{L-1,1,j}$ be the $j$th row of $\theta_{L-1,1}$. Then

$$\begin{pmatrix} \sigma(t_1\theta_{L-1,1,j}u) \\ \vdots \\ \sigma(t_N\theta_{L-1,1,j}u) \end{pmatrix} = \begin{pmatrix} \sum_{n=0}^{\infty} \frac{\sigma^{(n)}(0)}{n!}(t_1\theta_{L-1,1,j}u)^n \\ \vdots \\ \sum_{n=0}^{\infty} \frac{\sigma^{(n)}(0)}{n!}(t_N\theta_{L-1,1,j}u)^n \end{pmatrix} = \sum_{n=1}^{N} \frac{\sigma^{(n)}(0)}{n!}(\theta_{L-1,1,j}u)^n \begin{pmatrix} t_1^n \\ \vdots \\ t_N^n \end{pmatrix} + \mathcal{O}\begin{pmatrix} t_1^{N+1} \\ \vdots \\ t_N^{N+1} \end{pmatrix}$$

Note

$$\det \begin{pmatrix} t_1 & \cdots & t_1^N \\ \vdots & & \vdots \\ t_N & \cdots & t_N^N \end{pmatrix} = \prod_{i=1}^{N} t_i \prod_{1 \leq j < k \leq N} (t_k - t_j) > 0.$$

Thus the $N$ vectors $\begin{pmatrix} t_1^n \\ \vdots \\ t_N^n \end{pmatrix}$ are linearly independent.

We then choose $u$ s.t. $\theta_{L-1,1}u$ has at least $N$ distinct non-zero elements. We claim that the set of all $\theta_{L-1,1}$ such that there is no $u$ s.t. $\theta_{L-1,1}u$ has at least $N$ distinct non-zero elements, is measure-zero in $\mathbb{R}^{|\theta_{L-1,1}|}$. We denote the set to be $\mathcal{B}_{\theta_{L-1,1}}$.

Consider $v \in \mathbb{R}^m$ in the column space of $\theta_{L-1,1}$. Fix a partition of $m$ elements into at most $N-1$ blocks, where in each block of $v$ is constant. Such set of vectors is of dimensional $N-1$. Moreover, the number of such partition is finite. Therefore, $\text{Leb}_{|\theta_{L-1,1}|}(\mathcal{B}_{\theta_{L-1,1}}) = 0$.

WOLG, assume the first $N$ elements are distinct. Then, similarly, we have that the $N$ vectors $\begin{pmatrix} (\theta_{L-1,1,1}u)^n \\ \vdots \\ (\theta_{L-1,1,N}u)^n \end{pmatrix}$ are linearly independent. Thus

$$\det \begin{pmatrix} \frac{\sigma^{(1)}(0)}{1!}(\theta_{L-1,1,1}u) & \cdots & \frac{\sigma^{(N)}(0)}{N!}(\theta_{L-1,1,1}u)^N \\ \vdots & & \vdots \\ \frac{\sigma^{(1)}(0)}{1!}(\theta_{L-1,1,N}u) & \cdots & \frac{\sigma^{(N)}(0)}{N!}(\theta_{L-1,1,N}u)^N \end{pmatrix}$$

$$= \prod_{i=1}^N \frac{\sigma^{(i)}(0)}{i!} \det \begin{pmatrix} (\theta_{L-1,1,1}u) & \cdots & (\theta_{L-1,1,1}u)^N \\ \vdots & & \vdots \\ (\theta_{L-1,1,N}u) & \cdots & (\theta_{L-1,1,N}u)^N \end{pmatrix} \neq 0.$$

Therefore, $\begin{pmatrix} \sigma(t_1\theta_{L-1,1}u)^\top \\ \vdots \\ \sigma(t_N\theta_{L-1,1}u)^\top \end{pmatrix}$ is full rank $N$, and then $\begin{pmatrix} Q_1 \\ \vdots \\ Q_N \end{pmatrix}$ is full rank $Nd$.

Also, since $P_i$ is invertible for any $u_{L,i}$, we have $\begin{pmatrix} B_1 & & \\ & \ddots & \\ & & B_N \end{pmatrix}$ is invertible. Thus

$$\begin{pmatrix} \nabla_{\tau_\epsilon}\varphi_L(\tau_\epsilon(u_{L,1}))\nabla_u\tau_\epsilon(u_{L,1})\nabla_{\theta_{L-1,2}}\bar\varphi(u_{L-1,1}) \\ \vdots \\ \nabla_{\tau_\epsilon}\varphi_L(\tau_\epsilon(u_{L,N}))\nabla_u\tau_\epsilon(u_{L,N})\nabla_{\theta_{L-1,2}}\bar\varphi(u_{L-1,N}) \end{pmatrix} = \begin{pmatrix} B_1 & & \\ & \ddots & \\ & & B_N \end{pmatrix}\begin{pmatrix} Q_1 \\ \vdots \\ Q_N \end{pmatrix}$$

is full rank $Nd$, where $u_{L-1,i} = t_i u$. $\qquad\square$

# D  MINIMUM DISTANCE AND BIASED DISTRIBUTION: PROOF OF THEOREM 1

We now present the complete version of Theorem 1.

**Theorem 4.** *Suppose Assumption 1 holds.*

1. *For any biased distribution such that there exists a ball $B\left(C\left(\frac{-\log\delta}{\bar\pi_{\max}(N-1)V_d}\right)^{1/d}\right) \subseteq \Omega_{\max}$, and $\frac{\pi_{\max}}{\bar\pi_{\max}} = \mathcal{O}(1)$, we have with probability at least $1-2\delta$,*

$$\left(\frac{2\delta}{\pi_{\max}N(N-1)V_d}\right)^{1/d} \leq h_{\min} \leq C\left(\frac{-\log\delta}{\bar\pi_{\max}(N-1)V_d}\right)^{1/d}$$

*where $V_d = \frac{\pi^{d/2}}{\Gamma(d/2+1)}$ is the volumn of d-dimensional unit ball, and $C > 0$ is some universal constant.*

2. *For general distribution, we have with probability at least $1-2\delta$,*

$$\left(\frac{2\delta}{\pi_{\max}N(N-1)V_d}\right)^{1/d} \leq h_{\min} \leq \left(\frac{-\log\delta}{\pi_{\min}(N-1)V_d}\right)^{1/d}.$$

*Proof.* First fix any point $x_i = x$. For this proof, we consider some small enough $r \in (0,1)$, and then

$$\pi_{\min}V_d r^d \leq \mathbb{P}(\|x_j - x_i\| < r | x_i = x) = \int_{B(x,r)} \pi(z)dz \leq \pi_{\max}V_d r^d \tag{3}$$

where $V_d = \frac{\pi^{d/2}}{\Gamma(d/2+1)}$ is the volumn of $d$-dimensional unit ball.

We would like to lower bound

$$\mathbb{P}(h_{\min} \geq r) = \mathbb{P}(\|x_i - x_j\| \geq r, \; \forall 1 \leq i < j \leq N)$$

$$= \mathbb{E}\big[\mathbb{1}_{\{\|x_i-x_j\|\geq r, \; \forall 1\leq i<j\leq N\}}\big]$$

$$= \mathbb{E}\big[\mathbb{1}_{\{\|x_1-x_2\|\geq r\}}\mathbb{1}_{\{\|x_3-x_i\|\geq r, \; \forall i=1,2\}}\cdots\mathbb{1}_{\{\|x_N-x_i\|\geq r, \; \forall i=1,\cdots,N-1\}}\big]$$

$$= \mathbb{E}\big[\mathbb{E}[\mathbb{1}_{\{\|x_1-x_2\|\geq r\}}|x_1]\cdot\mathbb{E}[\mathbb{1}_{\{\|x_3-x_i\|\geq r, \; \forall i=1,2\}}|x_1,x_2]\cdots\mathbb{E}[\mathbb{1}_{\{\|x_N-x_i\|\geq r, \; \forall i=1,\cdots,N-1\}}|x_1,\cdots,x_{N-1}]\big]$$

where the last inequality follows from the properties of conditional expectation.

Also, we have

$$\mathbb{E}\big[\mathbb{1}_{\{\|x_1-x_2\|\geq r\}}|x_1\big] = \mathbb{P}\big(\|x_1 - x_2\| \geq r|x_1\big)$$
$$\geq 1 - \pi_{\max}V_d r^d$$

where the inequality follows from (3), and similarly,

$$\mathbb{E}\big[\mathbb{1}_{\{\|x_s-x_i\|\geq r, \ \forall i=1,\cdots,s-1\}}|x_1,\cdots,x_{s-1}\big] = \mathbb{P}\big(\|x_s - x_i\| \geq r, \ \forall i = 1,\cdots,s-1|x_1,\cdots,x_{s-1}\big)$$
$$\geq 1 - (s-1)\pi_{\max}V_d r^d.$$

Therefore, for some small $r$, we have

$$\mathbb{P}(h_{\min} \geq r) \geq \mathbb{E}\Bigg[\prod_{i=1}^{N-1}\big(1 - i\,\pi_{\max}V_d r^d\big)\Bigg] = \prod_{i=1}^{N-1}\big(1 - i\,\pi_{\max}V_d r^d\big)$$
$$\geq 1 - \sum_{i=1}^{N-1}i\,\pi_{\max}V_d r^d = 1 - \frac{N(N-1)}{2}\pi_{\max}V_d r^d.$$

Thus with probability at least $1 - \delta$, we have

$$h_{\min} \geq \left(\frac{2\delta}{\pi_{\max}N(N-1)V_d}\right)^{1/d}.$$

For the other side, for biased distribution satisfying the assumptions stated in the theorem, we consider

$$r \leq 2C\left(\frac{-\log\delta}{\bar\pi_{\max}(N-1)V_d}\right)^{1/d}, \text{ and } \Omega_{\max,B} = B(\frac{r}{2}) \subseteq \Omega_{\max}.$$

Then,

$$\mathbb{P}(h_{\min} \leq r) = \mathbb{P}(\exists i,j, s.t. \|x_i - x_j\| \leq r)$$
$$\geq \mathbb{P}(\exists \text{ at least two points } x_i, x_j \in \Omega_{\max,B}, i.e., \|x_i - x_j\| \leq r)$$
$$= 1 - \mathbb{P}(x_i \notin \Omega_{\max,B}, \ \forall i) - \mathbb{P}(\exists i, s.t. x_i \in \Omega_{\max,B}, \ x_j \notin \Omega_{\max,B}, \ \forall j \neq i)$$
$$\geq 1 - \left((1 - \bar\pi_{\max}|\Omega_{\max,B}|)^N + N(1 - \bar\pi_{\max}|\Omega_{\max,B}|)^{N-1}\pi_{\max}|\Omega_{\max,B}|\right)$$
$$= 1 - (1 - \bar\pi_{\max}|\Omega_{\max,B}|)^{N-1}(1 - \bar\pi_{\max}|\Omega_{\max,B}| + N\pi_{\max}|\Omega_{\max,B}|)$$
$$\geq 1 - e^{-(N-1)\bar\pi_{\max}|\Omega_{\max,B}|}(1 - \bar\pi_{\max}|\Omega_{\max,B}| + N\pi_{\max}|\Omega_{\max,B}|)$$

where the last inequality follows from $(1-x)^N \leq e^{-Nx}$.

Then, we have with probability at least $1 - \delta$,

$$h_{\min} \leq C\left(\frac{-\log\delta}{\bar\pi_{\max}(N-1)V_d}\right)^{1/d},$$

where this inequality follows from $|\Omega_{\max,B}| = V_d(\frac{r}{2})^d$ and $C > 0$ is some universal constant.

The last conclusion follows from taking union bound of the above two results.

For general distribution,

$$\mathbb{E}\big[\mathbb{1}_{\{\|x_1-x_2\|\geq r\}}|x_1\big] = \mathbb{P}\big(\|x_1 - x_2\| \geq r|x_1\big)$$
$$\leq 1 - \pi_{\min}V_d r^d$$

where the inequality follows from (3), and similarly,

$$\mathbb{E}\big[\mathbb{1}_{\{\|x_s-x_i\|\geq r, \ \forall i=1,\cdots,s-1\}}|x_1,\cdots,x_{s-1}\big] = \mathbb{P}\big(\|x_s - x_i\| \geq r, \ \forall i = 1,\cdots,s-1|x_1,\cdots,x_{s-1}\big)$$
$$\leq 1 - \pi_{\min}V_d r^d.$$

Therefore, for some small $r$, we have

$$\mathbb{P}(h_{\min} \geq r) \leq (1 - \pi_{\min} V_d r^d)^{N-1}$$

Thus with probability at least $1 - \delta$, we have

$$h_{\min} < \left(\frac{1 - \delta^{\frac{1}{N-1}}}{\pi_{\min} V_d}\right)^{1/d} \leq \left(\frac{-\log \delta}{\pi_{\min} V_d (N-1)}\right)^{1/d}$$

where the last inequality follows from $1 - e^{-x} \leq x$ for $x > 0$.

$\square$

# E   OPTIMIZATION: PROOF OF THEOREM 2 AND COROLLARY 3

We recall that the architecture of neural network is defined as follows

$$u_{0,i} = x_i$$
$$u_{\ell+1,i} = u_{\ell,i} + \varphi_\ell(\theta; u_{\ell,i}), \ \ell = 0, \cdots, L - 1$$
$$f(x_i) = \varphi_L(\theta; u_{L,i}).$$

The loss function of each data point is

$$l(f(x_i), y_i) = \frac{1}{2}\|y_i - f(x_i)\|^2$$

and the total loss is

$$\mathcal{L}(f(x_i), y_i) = \frac{1}{N}\sum_{i=1}^{N} l(f(x_i), y_i).$$

We then compute the Jacobian of the following maps that will be used in the proof:

$$\nabla_f l(x_i) = (f(x_i) - y_i)^\top$$

$$\nabla_{\theta_j} f(x_i) = \sum_{\ell \in \mathcal{J}_j} \nabla_u \bar{\varphi}_L(u_{L,i})(I + \nabla_u \bar{\varphi}_{L-1}(u_{L-1,i})) \cdots (I + \nabla_u \bar{\varphi}_{\ell+1}(u_{\ell+1,i})) \nabla_{\theta_j} \bar{\varphi}_\ell(u_{\ell,i}; \theta)$$

where $\nabla_u \bar{\varphi}_\ell(u_\ell)$ is the Jacobian of the map $\bar{\varphi}_\ell(u_\ell)$ w.r.t. $u_\ell$, $\nabla_{\theta_\ell} f(x_i)$ is also the Jacobian; the set $\mathcal{J}_j$ denotes the indices of layers in which $\theta_j$ appears.

Then

$$\nabla_{\theta_j} l(x_i) = \nabla_f l(x_i) \nabla_{\theta_\ell} f(x_i)$$
$$= \sum_{\ell \in \mathcal{J}_j} \nabla_f l(x_i) \nabla_u \bar{\varphi}_L(u_{L,i})(I + \nabla_u \bar{\varphi}_{L-1}(u_{L-1,i})) \cdots (I + \nabla_u \bar{\varphi}_{\ell+1}(u_{\ell+1,i})) \nabla_{\theta_j} \bar{\varphi}_\ell(u_{\ell,i}; \theta)$$

and we have

$$\nabla_{\theta_j}\mathcal{L} = \frac{1}{N}\sum_{i=1}^{N} \nabla_{\theta_j} l(x_i).$$

As shown in equation (1), we consider the case where the weights at each layer are distinct, i.e., $\mathcal{J}_j = \{j\}$.

In this section, we also define the following function for any matrix function $M(x) \in \mathbb{R}^{n \times m}$ with $m \geq n$

$$\det_{\mathrm{r}}(M(x)) = \sum_{i=1}^{\binom{m}{n}} (\det M_i(x))^2,$$

where $M_i(x) \in \mathbb{R}^{n \times n}$ is the matrix with $n$ columns of $M(x)$. Then $\det_{\mathrm{r}}(M(x)) \neq 0$ if and only if $M(x)$ is full rank at $x$.

The following is a more complete version of the convergence result in Theorem 2, combining Corollary 3. We also specify that the dense set of $\eta$ is indeed residual (see more details in Appendix E.3).

**Theorem 5.** *Suppose Assumptions 1 to 4 hold. Consider the initialization $\theta^0$ in $\mathbb{R}^{\dim\theta}$ except for a measure-zero set. Then, under some arbitrarily small adjustment on the scale of $\varphi_\ell$ for $\ell = 0, \cdots, L-1$, with probability 1 over the joint distribution $\pi \times \cdots \times \pi$ of the input data $x_1, \cdots, x_N$, we have the following results:*

1. *When $\eta$ is in some residual set of $\left(0, \min\left\{\frac{2-\delta}{\mathsf{L}_1}, 1\right\}\right)$, where*

$$\mathsf{L}_1 = S\left(\cdots, \|\theta_i^0\| + \sqrt{\frac{2C}{\delta}\mathcal{L}(\theta^0)} + \eta \max_{\|\theta\| \leq \|\theta^0\| + \sqrt{\frac{2C}{\delta}\mathcal{L}(\theta^0)}} \|\nabla_{\theta_i}\mathcal{L}(\theta)\|, \cdots\right)$$

*for some polynomial $S(\cdot)$ with positive coefficients, we have*

$$\mathcal{L}(\theta^k) \leq \prod_{s=0}^{k-1}\left(1 - \eta\left(1 - \frac{\eta\mathsf{L}_1}{2}\right)\frac{\mu_{\mathrm{low},s,X}}{N}\right)\mathcal{L}(\theta^0), \ \forall\, k \leq T = \frac{C}{\eta},$$

*where $\mu_{\mathrm{low},s,X} > 0$ is some strictly positive constant depending on $\theta^s$ and $x_1, \cdots, x_N$, and $C > 0$ is universal constant.*

2. *Furthermore, let $R = \sqrt{\|\theta^0 - \theta^*\|^2 + \frac{4\rho+2}{\delta}\mathcal{L}(\theta^0)}$ for some stationary point $\theta^*$. Assume $\mathcal{L}(\theta)$ satisfies the $(\theta^0, \theta^*, R, \rho, \epsilon_{\mathcal{L}})$-dissipative condition in Definition 5. When $\eta$ is in some residual set of $\left(0, \min\left\{\frac{2-\delta}{\mathsf{L}_2}, 1\right\}\right)$, where*

$$\mathsf{L}_2 = S\left(\cdots, \|\theta_i^*\| + R + \max_{\theta \in B_{\theta^*}(R)} \|\nabla_{\theta_i}\mathcal{L}(\theta)\|, \cdots\right),$$

*GD will converge to $\|\nabla\mathcal{L}(\theta)\| \leq \epsilon_{\mathcal{L}}$, and*

$$\mathcal{L}(\theta^k) \leq \prod_{s=0}^{k-1}\left(1 - \eta\left(1 - \frac{\eta\mathsf{L}_2}{2}\right)\frac{\mu_{\mathrm{low},s,X}}{N}\right)\mathcal{L}(\theta^0), \ \forall\, k \geq 1 \text{ and } \|\nabla\mathcal{L}(\theta^k)\| > \epsilon_{\mathcal{L}}.$$

3. *Fix $x_i$, and let $\mathcal{D}_{i,H} = \{j \mid h_{ij} = \|x_i - x_j\| \leq H, \ \forall\, j \neq i\}$. Then for both the above cases, for any $k \geq 1$, there exists $r_{k,i,H} > 0$ and $L_{k,i,H} > 0$, s.t., when $\sqrt{\sum_{j \in \mathcal{D}_{i,H}} h_{ij}^2} \leq r_{k,i,H}$, we have*

$$\mu_{\mathrm{low,s,X}} \leq L_{k,i,H}\sqrt{\sum_{j \in \mathcal{D}_{i,H}} h_{ij}^2}, \ \forall s \leq k.$$

*Specifically, if $H = h_{\min}$ and $\mathcal{D}_{i,H} \neq \varnothing$, there exists $L_{k,i}, r_{k,i} > 0$, s.t., when $h_{\min} \leq r_{k,i}$, we have*

$$\mu_{\mathrm{low,s,X}} \leq L_{k,i}h_{\min}, \ \forall s \leq k.$$

*Proof.* Define the GD iteration map to be

$$\Psi(\theta) = \theta - \eta\nabla_\theta\mathcal{L}(\theta), \text{ and } \Psi^k(\theta) = \underbrace{\Psi \circ \cdots \circ \Psi}_{k}(\theta).$$

First, consider fixed $x_1, \cdots, x_N$. Let

$$\mathcal{B}_{\theta,\mathrm{rank}} = \{\theta : \nabla_\theta\Psi(\theta) \text{ is not full rank}\}$$

By Lemma 11, we know that for $\eta \in \mathcal{D}_{\eta,0}$, where $\mathcal{D}_{\eta,0}$ is a residual set of $(0, \min\{\frac{2}{L}, 1\})$, we have

$$\mathrm{Leb}_{\dim\theta}(\mathcal{B}_{\theta,\mathrm{rank}}) = 0.$$

Next we would like to show that along the trajectory, i.e., for any $k$, the union of "bad" sets of the initial condition $\theta$ leading to the degeneracy of the map $\Psi^k$ is a measure-zero set.

Now consider any measure-zero set $\mathcal{B}_{\theta,0}$ in $\mathbb{R}^{\dim\theta}$. By Theorem 8,

$$\mathrm{Leb}_{\dim\theta}(\Psi^{-1}(\mathcal{B}_{\theta,0})) = 0.$$

We then would like to prove that $(\Psi^k)^{-1}(\mathcal{B}_{\theta,0})$ is measure-zero for any $k \geq 1$. Suppose $(\Psi^k)^{-1}(\mathcal{B}_{\theta,0})$ is measure-zero. We would like to show that $(\Psi^{k+1})^{-1}(\mathcal{B}_{\theta,0})$ is measure zero. This is obvious since $(\Psi^{k+1})^{-1}(\mathcal{B}_{\theta,0}) = \Psi^{-1}((\Psi^k)^{-1}(\mathcal{B}_{\theta,0}))$.

Let $\mathcal{B}_{\theta,0}$ be the union of all the bad measure-zero sets of $\theta$ in Assumption 4 and Lemmas 4, including $\mathcal{B}_{\theta,\text{rank}}$, and there are finite union of such sets, which implies $\text{Leb}_{\dim\theta}(\mathcal{B}_{\theta,0}) = 0$. Let

$$\mathcal{B}_\theta = \bigcup_{k=1}^{\infty} (\Psi^k)^{-1}(\mathcal{B}_{\theta,0}).$$

Then

$$\text{Leb}_{\dim\theta}(\mathcal{B}_\theta) = \text{Leb}_{\dim\theta}\left(\bigcup_{k=1}^{\infty} (\Psi^k)^{-1}(\mathcal{B}_{\theta,0})\right) \leq \sum_{k=1}^{\infty} 0 = 0,$$

namely, the initial condition set of $\theta$ where there exists some iteration $k$ s.t. $\theta^k \in \mathcal{B}_{\theta,0}$, is measure-zero. In the rest of the proof, we just consider $\theta^0 \in \mathbb{R}^{\dim\theta}\backslash\mathcal{B}_\theta$.

Next we consider the bad set of $(x_1, \cdots, x_N)$. By Lemma 4, for any $\theta^0$, the bad set of $(x_1, \cdots, x_N)$, denoted as $\mathcal{B}_{X,0}$, is measure-zero,

$$\text{Leb}_{Nd}(\mathcal{B}_{X,0}) = 0.$$

Also, there are at most countable $\theta^k$ in the GD trajectory, and thus all such bad sets $\bigcup_{k=0}^{\infty} \mathcal{B}_{X,0}$ satisfy

$$\text{Leb}_{Nd}(\bigcup_{k=0}^{\infty} \mathcal{B}_{X,k}) \leq \sum_{k=0}^{\infty} 0 = 0.$$

Since $(x_1, \cdots, x_N) \sim \pi \times \cdots \times \pi$, and $\pi \ll \text{Leb}$, we have

$$(\pi \times \cdots \times \pi)(\mathcal{B}_X) = 0,$$

namely, with probability 0, $(x_1, \cdots, x_N)$ will be in this set.

Then, consider some initial conditions $\theta^0$ except for a measure-zero set, and consider some dataset $\{(x_i, y_i)\}_{i=1}^{N}$ chosen with probability 1 over the joint distribution. By Lemma 7 and Lemma 2, we have for $i = 1, 2$,

$$\mathcal{L}(\theta^{k+1}) \leq \mathcal{L}(\theta^k) + \nabla\mathcal{L}(\theta^k)^\top(\theta^{k+1} - \theta^k) + \frac{S_k}{2}\|\theta^k - \theta^{k+1}\|^2$$

$$= \mathcal{L}(\theta^k) - \eta\left(1 - \frac{\eta S_k}{2}\right)\|\nabla\mathcal{L}(\theta^k)\|^2$$

$$\leq \left(1 - \eta\left(1 - \frac{\eta S_k}{2}\right)\frac{\mu_{\text{low},k,X}}{N}\right)\mathcal{L}(\theta^k)$$

$$\leq \left(1 - \eta\left(1 - \frac{\eta \mathsf{L}_i}{2}\right)\frac{\mu_{\text{low},k,X}}{N}\right)\mathcal{L}(\theta^k)$$

$$\leq \prod_{s=0}^{k}\left(1 - \eta\left(1 - \frac{\eta \mathsf{L}_i}{2}\right)\frac{\mu_{\text{low},s,X}}{N}\right)\mathcal{L}(\theta^0)$$

where the second inequality follows from Lemma 4, the third inequality follows from Lemma 3 and $S(a_1, \cdots, a_{n_\theta}) \leq \mathsf{L}$ by the monotonicity of $S(\cdot)$ in $\mathbb{R}_{\geq 0} \times \cdots \times \mathbb{R}_{\geq 0}$; $\mu_{\text{low},k,X} > 0$ is some strictly positive constant depending on $\theta^k$ and $x_1, \cdots, x_N$.

By Lemma 3, when $i = 1$, the above inequality holds for $k + 1 \leq T = \frac{C}{\eta}$. When $i = 2$ with $(\theta^0, \theta^*, R, \rho, \epsilon_\mathcal{L})$-dissipative condition, it holds for any $k \geq 1$.

Then, under $(\theta^0, \theta^*, R, \rho, \epsilon_\mathcal{L})$-dissipative condition, GD will converge to the region $\|\nabla\mathcal{L}(\theta)\| \leq \epsilon_\mathcal{L}$. If not, we have $\mu_{\text{low},k,X} > 0$ because by Assumption 4 and Lemma 4, the $\theta$ s.t. $\mu_{\text{low},\theta,X} = 0$ lies in some measure-zero set and we remove any finite-step convergence to this set at the beginning of this proof. Thus $\left(1 - \eta\left(1 - \frac{\eta \mathsf{L}_2}{2}\right)\frac{\mu_{\text{low},k,X}}{N}\right) < 1$, which means $\mathcal{L}(\theta^k)$ will keep decreasing. Contradiction.

The last statement follows from Lemma 4, and take $L_{k,i,H} = \max_{s \le k} L_{s,i,H}$ and $r_{k,i,H} = \min_{s \le k} r_{s,i,H}$.

$\square$

The following lemma shows that the loss function is Lipschitz smooth along the GD iteration.

**Lemma 3.** *Assume $\mathcal{L}(\theta)$ satisfies polynomial generalized smoothness.*

1. *Without $(\theta^0, \theta^*, R, \rho, \epsilon_{\mathcal{L}})$-dissipative condition, let $\eta \le \min\{\frac{2-\delta}{\mathsf{L}_2}, 1\}$, where*

$$\mathsf{L}_2 = S\left(\cdots, \|\theta_i^0\| + \sqrt{\frac{2C}{\delta}\mathcal{L}(\theta^0)} + \max_{\|\theta\| \le \|\theta^0\| + \sqrt{\frac{2C}{\delta}\mathcal{L}(\theta^0)}} \|\nabla_{\theta_i}\mathcal{L}(\theta)\|, \cdots\right).$$

   *Then*

$$S_k \le \mathsf{L}_2, \ \forall\, k \le T = \frac{C}{\eta}.$$

2. *With $(\theta^0, \theta^*, R, \rho, \epsilon_{\mathcal{L}})$-dissipative condition, let $\eta \le \min\{\frac{2-\delta}{\mathsf{L}_2}, 1\}$, where*

$$\mathsf{L}_2 = S\left(\cdots, \|\theta_i^*\| + \sqrt{\|\theta^0 - \theta^*\|^2 + \frac{4\rho+2}{\delta}\mathcal{L}(\theta^0)} + \max_{\theta \in B_{\theta^*}\left(\sqrt{\|\theta^0-\theta^*\|^2+\frac{4\rho+2}{\delta}\mathcal{L}(\theta^0)}\right)} \|\nabla_{\theta_i}\mathcal{L}(\theta)\|, \cdots\right).$$

   *Then*

$$S_k \le \mathsf{L}_2, \ \forall\, k \ge 0 \text{ and } \|\nabla\mathcal{L}(\theta^k)\| \ge \epsilon_{\mathcal{L}}.$$

*Proof.* First, note

$$\max\{\|\theta_i^k\|, \|\theta_i^{k+1}\|\} \le \|\theta_i^k\| + \eta\|\nabla\mathcal{L}_{\theta_i}(\theta^k)\|.$$

By the monotonicity of $S(\cdot)$, we only need to show $\|\theta_i^k\|$ is bounded under the two cases.

Without the dissipative assumption, we would like to show that

$$\|\theta_i^k\| \le \|\theta_i^0\| + \sqrt{\frac{2C}{\delta}\mathcal{L}(\theta^0)}, \ \forall k \le T = \frac{C}{\eta}.$$

Suppose the above inequality holds for all $\theta_i^j$, where $j \le k$ and $i = 1, \cdots, n_\theta$. Then by $\eta \le 1$, we have

$$S_k \le \bar{S}_k = S(\cdots, \|\theta_i^k\| + \eta\|\nabla_{\theta_i}\mathcal{L}(\theta^k)\|, \cdots))$$

$$\le \mathsf{L}_2 = S(\cdots, \|\theta_i^0\| + \sqrt{\frac{2C}{\delta}\mathcal{L}(\theta^0)} + \max_{\|\theta\| \le \|\theta^0\| + \sqrt{\frac{2C}{\delta}\mathcal{L}(\theta^0)}} \|\nabla_{\theta_i}\mathcal{L}(\theta)\|, \cdots))$$

namely,

$$\eta \le \frac{2-\delta}{\mathsf{L}_2} \le \frac{2-\delta}{\bar{S}_k}.$$

By Lemma 7 and Lemma 2, we have

$$0 \le \mathcal{L}(\theta^{k+1}) \le \mathcal{L}(\theta^k) + \nabla\mathcal{L}(\theta^k)^\top(\theta^{k+1} - \theta^k) + \frac{S_k}{2}\|\theta^k - \theta^{k+1}\|^2$$

$$= \mathcal{L}(\theta^k) - \eta\left(1 - \frac{\eta S_k}{2}\right)\|\nabla\mathcal{L}(\theta^k)\|^2$$

$$\le \mathcal{L}(\theta^0) - \eta\sum_{j=0}^{k}\left(1 - \frac{\eta S_j}{2}\right)\|\nabla\mathcal{L}(\theta^j)\|^2$$

$$\le \mathcal{L}(\theta^0) - \frac{\delta}{2}\eta\sum_{j=0}^{k}\|\nabla\mathcal{L}(\theta^j)\|^2,$$

namely,

$$\sum_{j=0}^{k} \|\nabla \mathcal{L}(\theta^j)\|^2 \le \frac{2}{\delta \eta} (\mathcal{L}(\theta^0) - \mathcal{L}(\theta^{k+1})) \le \frac{2}{\delta \eta} \mathcal{L}(\theta^0). \tag{4}$$

Then

$$\|\theta_i^{k+1}\| = \|\theta_i^k - \eta \nabla_{\theta_i} \mathcal{L}(\theta^k)\| = \left\| \theta_i^0 - \eta \sum_{j=0}^{k} \nabla_{\theta_i} \mathcal{L}(\theta^j) \right\|$$

$$\le \|\theta_i^0\| + \eta \sum_{j=0}^{k} \left\| \nabla \mathcal{L}(\theta^j) \right\|$$

$$\le \|\theta_i^0\| + \eta \sqrt{k \sum_{j=0}^{k} \|\nabla \mathcal{L}(\theta^j)\|^2}$$

$$\le \|\theta_i^0\| + \sqrt{\frac{2}{\delta} k \eta \mathcal{L}(\theta^0)}$$

$$\le \|\theta_i^0\| + \sqrt{\frac{2C}{\delta} \mathcal{L}(\theta^0)}$$

where the second inequality follows from Cauchy-Schwartz inequality, the third inequality follows from (4), and last inequality follows from $k \le T = \frac{C}{\eta}$.

Then the above derivation for $\theta^k$ also holds for $\theta^{k+1}$. Therefore $S_{k+1} \le \mathsf{L}_2$.

Next, under the dissipative condition, we would like to show that

$$\theta_i^k \in B_{\theta_i^*} \left( \sqrt{\|\theta^0 - \theta^*\|^2 + \frac{4\rho+2}{\delta} \mathcal{L}(\theta^0)} \right), \ i.e., \ \|\theta_i^k - \theta_i^*\|^2 \le \|\theta^0 - \theta^*\|^2 + \frac{4\rho+2}{\delta} \mathcal{L}(\theta^0)$$

Suppose the above inequality holds for all $j \le k$. Then we have

$$S_k \le \bar{S}_k = S(\cdots, \|\theta_i^k\| + \eta \|\nabla_{\theta_i} \mathcal{L}(\theta^k)\|, \cdots)$$

$$\le \mathsf{L}_2 = S\left(\cdots, \|\theta_i^*\| + \sqrt{\|\theta^0 - \theta^*\|^2 + \frac{4\rho+2}{\delta} \mathcal{L}(\theta^0)} + \max_{\theta \in B_{\theta^*}\left(\sqrt{\|\theta^0-\theta^*\|^2+\frac{4\rho+2}{\delta}\mathcal{L}(\theta^0)}\right)} \|\nabla_{\theta_i} \mathcal{L}(\theta)\|, \cdots\right)$$

namely,

$$\eta \le \frac{2-\delta}{\mathsf{L}_2} \le \frac{2-\delta}{\bar{S}_k}.$$

Then, consider the $k + 1$th iteration

$$\|\theta^{k+1} - \theta^*\|^2 = \|\theta^k - \theta^* - \eta \nabla \mathcal{L}(\theta^k)\|^2$$

$$= \|\theta^k - \theta^*\|^2 - 2\eta \nabla \mathcal{L}(\theta^k)^\top (\theta^k - \theta^*) + \eta^2 \|\nabla \mathcal{L}(\theta^k)\|^2$$

$$\le \|\theta^k - \theta^*\|^2 + \eta(2\rho + \eta) \|\nabla \mathcal{L}(\theta^k)\|^2$$

$$\le \|\theta^0 - \theta^*\|^2 + \eta(2\rho + \eta) \sum_{j=0}^{k} \|\nabla \mathcal{L}(\theta^j)\|^2$$

$$\le \|\theta^0 - \theta^*\|^2 + \frac{4\rho + 2\eta}{\delta} \mathcal{L}(\theta^0)$$

where the first inequality follows from Definition 5, and the last inequality follows from (4), which also holds true when $\mathsf{L}_2$ is replaced by $\mathsf{L}_2$.

Then together with $\eta \le 1$ and

$$\|\theta_i^k\| \le \|\theta_i^*\| + \|\theta_i^k - \theta_i^*\| \le \|\theta_i^*\| + \|\theta^k - \theta^*\|,$$

we obtain $S_k \le \mathsf{L}_2$ for all k. $\qquad \square$

### E.1 LOWER BOUND OF GRADIENT

In this section, we use $\theta_i$ and $\theta_{\bar{\ell},i}$ to represent the scalar elements of $\theta$ and $\theta_{\bar{\ell}}$ respectively. The main lemma in this section is as follows.

**Lemma 4.** *Under Assumption 4, and some small adjustment of the scale of $\varphi_\ell$ for $\ell = 0, \cdots, L-1$, given fixed $\theta$ except for a measure-zero set in $\mathbb{R}^{\dim\theta}$, and $X = (x_1, \cdots, x_N)$ except for a measure-zero set in $\mathbb{R}^{Nd}$ that depends on $\theta$, the gradient satisfies the following inequality*

$$\|\nabla_\theta \mathcal{L}(\theta; X)\|^2 \geq \frac{\mu_{\mathrm{low},\theta,\mathrm{X}}}{N} \mathcal{L}(\theta; X)$$

*where $\mu_{\mathrm{low},\theta,\mathrm{X}} > 0$ is a strictly positive constant depending on $\theta$ and $X$.*

*Moreover, fix $x_i$, and let $\mathcal{D}_{i,H} = \{j \,|\, h_{ij} = \|x_i - x_j\| \leq H, \,\forall j \neq i\}$. Then there exists $r_{\theta,i,H} > 0$ and $L_{\theta,i,H} > 0$, s.t., when $\sqrt{\sum_{j\in\mathcal{D}_{i,H}} h_{ij}^2} \leq r_{\theta,i,H}$, we have*

$$\mu_{\mathrm{low},\theta,\mathrm{X}} \leq L_{\theta,i,H} \sqrt{\sum_{j\in\mathcal{D}_{i,H}} h_{ij}^2}.$$

*Specifically, if $H = h_{\min}$ and $\mathcal{D}_{i,H} \neq \varnothing$, there exists $L_{\theta,i}, r_{\theta,i} > 0$, s.t., when $h_{\min} \leq r_{\theta,i}$, we have*

$$\mu_{\mathrm{low},\theta,\mathrm{X}} \leq L_{\theta,i} h_{\min}.$$

*Proof.* Consider

$$\|\nabla_\theta \mathcal{L}(\theta; X)\|^2 = \left\| \frac{1}{N} \sum_{i=1}^N \nabla_f l(x_i) \nabla_\theta f(x_i) \right\|^2$$

$$\geq \frac{1}{N^2} \mu_{\mathrm{low},\theta,\mathrm{X}} \sum_{i=1}^N \|\nabla_f l(x_i)\|^2 = \frac{\mu_{\mathrm{low},\theta,\mathrm{X}}}{N} \mathcal{L}(\theta; X)$$

where the first inequality follows from Lemma 6, and $\mu_{\mathrm{low},\theta,\mathrm{X}} > 0$ is a constant depending on $\theta$ and $X$. The second statement follows from Lemma 6. $\qquad\square$

Before presenting other lemmas, we first introduce the definition of a frame as follows.

**Definition 6** (Frame). *The set of vectors $\{e_k\}$ in an inner-product vector space $V$ is a frame of $V$ if there exists constants $0 < \mu_{\mathrm{low}} \leq \mu_{\mathrm{up}} < \infty$, s.t.,*

$$\mu_{\mathrm{low}} \|v\|^2 \leq \sum_k |\langle v, e_k \rangle|^2 \leq \mu_{\mathrm{up}} \|v\|^2, \;\; \forall v \in V.$$

*The frame operator $T : V \to V$ is defined as*

$$Tv = \sum_k \langle v, e_k \rangle e_k = \left( \sum_k e_k e_k^\top \right) v.$$

Then the gradient of the network $f$ at all data points forms a frame in $\mathbb{R}^{Nd}$:

**Lemma 5.** *Under Assumption 4 and some small adjustment of the scale of $\varphi_\ell$ for $\ell = 0, \cdots, L-1$, for any fixed $\theta$ except for a measure-zero set, $\left\{ \begin{pmatrix} \nabla_{\theta_i} f(x_1) \\ \vdots \\ \nabla_{\theta_i} f(x_N)) \end{pmatrix} \right\}_{i=1}^{\dim\theta}$ form a frame except for a measure-zero set of $(x_1, \cdots, x_N)$ in $\mathbb{R}^{Nd}$.*

*Proof.* By Assumption 4, there exists $\bar{\ell} \in \{0, \cdots, L-1\}$ and $U = (u_1, \cdots, u_N) \in \mathbb{R}^{Nd}, \theta$, s.t.

$$\det_{\mathrm{r}} \begin{pmatrix} \nabla_{\theta_{\bar{\ell}}} \tilde{f}_{\bar{\ell}}(\theta; u_1) \\ \vdots \\ \nabla_{\theta_{\bar{\ell}}} \tilde{f}_{\bar{\ell}}(\theta; u_N) \end{pmatrix} \neq 0,$$

where $\tilde{f}_{\bar{\ell}}(\theta; u_{\bar{\ell}}) = f(\theta; x)$.

Then by Theorem 7,

$$\mathrm{Leb}_{Nd}\left(\left\{U \in \mathbb{R}^{Nd} : \det_{\mathrm{r}}\begin{pmatrix}\nabla_{\theta_{\bar{\ell}}}\tilde{f}_{\bar{\ell}}(\theta; u_1) \\ \vdots \\ \nabla_{\theta_{\bar{\ell}}}\tilde{f}_{\bar{\ell}}(\theta; u_N)\end{pmatrix} = 0\right\}\right) = 0,$$

namely,

$$\mathrm{Leb}_{Nd}\left(\left\{U \in \mathbb{R}^{Nd} : \begin{pmatrix}\nabla_{\theta_{\bar{\ell}}}\tilde{f}_{\bar{\ell}}(\theta; u_1) \\ \vdots \\ \nabla_{\theta_{\bar{\ell}}}\tilde{f}_{\bar{\ell}}(\theta; u_N)\end{pmatrix} \text{ is not full rank}\right\}\right) = 0$$

Next consider the map $U = U(X) = \begin{pmatrix}u_{\bar{\ell}}(x_1) \\ \vdots \\ u_{\bar{\ell}}(x_N)\end{pmatrix} : \mathbb{R}^{Nd} \to \mathbb{R}^{Nd}$. By Lemma 10,

$$\mathrm{Leb}_{Nd}\left(\left\{(x_1, \cdots, x_N) \in \mathbb{R}^{Nd} : \begin{pmatrix}\nabla_{\theta_{\bar{\ell}}}\tilde{f}_{\bar{\ell}}(\theta; u(x_1)) \\ \vdots \\ \nabla_{\theta_{\bar{\ell}}}\tilde{f}_{\bar{\ell}}(\theta; u(x_N))\end{pmatrix} \text{ is not full rank}\right\}\right) = 0$$

Therefore, for any fixed $\theta$ except for a measure-zero set, $\left\{\begin{pmatrix}\nabla_{\theta_i}f(x_1) \\ \vdots \\ \nabla_{\theta_i}f(x_N))\end{pmatrix}\right\}_{i=1}^{\dim\theta}$ forms a frame except for a measure-zero set of $(x_1, \cdots, x_N)$. $\qquad\square$

**Lemma 6.** *Under Assumption 4 and some small adjustment of the scale of $\varphi_\ell$ for $\ell = 0, \cdots, L-1$, consider fixed $\theta$ except for a measure-zero set. The lower frame bound $\mu_{\mathrm{low},\theta,\mathrm{X}}$ is strictly positive except for a measure-zero set of $(x_1, \cdots, x_N)$ in $\mathbb{R}^{Nd}$.*

*Moreover, fix $x_i$, and let $\mathcal{D}_{i,H} = \{j \,|\, h_{ij} = \|x_i - x_j\| \le H,\ \forall j \ne i\}$. Then there exists $r_{\theta,i,H} > 0$ and $L_{\theta,i,H} > 0$, s.t., when $\sqrt{\sum_{j\in\mathcal{D}_{i,H}} h_{ij}^2} \le r_{\theta,i,H}$, we have*

$$\mu_{\mathrm{low},\theta,\mathrm{X}} \le L_{\theta,i,H}\sqrt{\sum_{j\in\mathcal{D}_{i,H}} h_{ij}^2}.$$

*Specifically, if $H = h_{\min}$ and $\mathcal{D}_{i,H} \ne \varnothing$, there exists $L_{\theta,i}, r_{\theta,i} > 0$, s.t., when $h_{\min} \le r_{\theta,i}$, we have*

$$\mu_{\mathrm{low},\theta,\mathrm{X}} \le L_{\theta,i}h_{\min}.$$

*Proof.* Let $A(\theta, X)$ be the frame operator, i.e.,

$$A(\theta, X) = \sum_{i=1}^{\dim\theta}\begin{pmatrix}\nabla_{\theta_i}f(x_1) \\ \vdots \\ \nabla_{\theta_i}f(x_N)\end{pmatrix}\begin{pmatrix}\nabla_{\theta_i}f(x_1) \\ \vdots \\ \nabla_{\theta_i}f(x_N)\end{pmatrix}^\top = \begin{pmatrix}\nabla_{\theta}f(x_1) \\ \vdots \\ \nabla_{\theta}f(x_N)\end{pmatrix}\begin{pmatrix}\nabla_{\theta}f(x_1) \\ \vdots \\ \nabla_{\theta}f(x_N)\end{pmatrix}^\top$$

We would like to bound $\lambda_{\min}(A)$. Consider any fixed $\theta$ except for a measure-zero set. Let

$$B(\theta, X) = \sum_{i=1}^{\dim\theta_{\bar{\ell}}}\begin{pmatrix}\nabla_{\theta_{\bar{\ell},i}}f(x_1) \\ \vdots \\ \nabla_{\theta_{\bar{\ell},i}}f(x_N)\end{pmatrix}\begin{pmatrix}\nabla_{\theta_{\bar{\ell},i}}f(x_1) \\ \vdots \\ \nabla_{\theta_{\bar{\ell},i}}f(x_N)\end{pmatrix}^\top$$

$$= \sum_{i=1}^{\dim\theta_{\bar{\ell}}}\begin{pmatrix}\nabla_{\theta_{\bar{\ell},i}}\tilde{f}_{\bar{\ell}}(\theta; u_{\bar{\ell}}(x_1)) \\ \vdots \\ \nabla_{\theta_{\bar{\ell},i}}\tilde{f}_{\bar{\ell}}(\theta; u_{\bar{\ell}}(x_N))\end{pmatrix}\begin{pmatrix}\nabla_{\theta_{\bar{\ell},i}}\tilde{f}_{\bar{\ell}}(\theta; u_{\bar{\ell}}(x_1)) \\ \vdots \\ \nabla_{\theta_{\bar{\ell},i}}\tilde{f}_{\bar{\ell}}(\theta; u_{\bar{\ell}}(x_N))\end{pmatrix}^\top$$

We also denote

$$\tilde{B}(\theta, U) = \tilde{B}(\theta, (u_{\bar{\ell},1}, \cdots, u_{\bar{\ell},N})) \stackrel{\Delta}{=} B(\theta, X)$$

Then obviously, we have

$$\lambda_{\min}(A) \geq \lambda_{\min}(B) = \lambda_{\min}(\tilde{B}).$$

By Lemma 5, $\lambda_{\min}(B) > 0$ except for a measure-zero set of $(x_1, \cdots, x_N)$ in $\mathbb{R}^{Nd}$.

In the end, we would like to show the dependence on $h_{\min}$. First, consider $h_{\min} = 0$, i.e., there exists $i, j$ s.t. $x_i = x_j$. Then obviously $A(\theta, X)$ is degenerate, and $\sigma_{\min}(A(\theta, X)) = 0$.

Next, let $\mathcal{D}_{i,H} = \{j \,|\, h_{ij} = \|x_i - x_j\| \leq H, \ \forall j \neq i\}$. WLOG, assume that $\mathcal{D}_{N-k,H} = \{N-k+1, \cdots, N\}$ for some $k$, and consider $X_1 = \begin{pmatrix} x_1 \\ \vdots \\ x_{N-k} \end{pmatrix}$, and $X_2 = \begin{pmatrix} x_{N-k+1} \\ \vdots \\ x_N \end{pmatrix}$.

Since $A(\theta, X)$ is symmetric, its eigenvalues are real for any $X$ and $\theta$. By Theorem 4.1 in Kurdyka & Paunescu (2008), we have that the map from $X$ to all the eigenvalues listed as a column is locally Lipschitz. Therefore, there exists $r_{\theta,X,k} > 0$ and $L_{\theta,X,k} > 0$, s.t., for any $X_2 \in B_{X_2'}(r_{\theta,X,k})$, where $X_2' = \begin{pmatrix} x_{N-k} \\ \vdots \\ x_{N-k} \end{pmatrix}$, i.e., $\sqrt{\sum_{j \in \mathcal{D}_{N-k,H}} h_{N-k,j}^2} \leq r_{\theta,X,k}$, we have

$$\lambda_{\min}(\theta, X_1, X_2) \leq L_{\theta,X,k} \|X_2' - X_2\| = L_{\theta,X,k} \sqrt{\sum_{j \in \mathcal{D}_{N-k,H}} h_{N-k,j}^2} = L_{\theta,X,k} \sqrt{\sum_{j=0}^{k-1} h_{N-k,N-j}^2}.$$

Specifically, if $H = h_{\min}$ and $\mathcal{D}_{N-k,H} \neq \varnothing$, we have

$$\lambda_{\min}(\theta, X_1, X_2) \leq L_{\theta,X} h_{\min},$$

for $h_{\min} \leq r_{\theta,X}$ and $L_{\theta,X}, r_{\theta,X} > 0$. $\qquad\square$

### E.2 Polynomial generalized smoothness of the loss

In this section, we denote $\theta_{\max,i} = \arg\max\{\|\theta\|, \|\theta'\|\}$, and $u_{\max} = \arg\max\{\|u\|, \|u'\|\}$. All the constants $C$'s are $\geq 0$ and independent of $\theta, \bar{\varphi}, u$; all the $p$'s in the subscripts are in $\mathbb{N}$. We denote $u_{\ell,i} = u_{\ell,i}(\theta)$ to emphasize the dependency on $\theta$, especially when comparing $u_{\ell,i}$ under different $\theta$'s. Also, note that $S(\cdot)$ has positive coefficients. Therefore $S(\cdot)$ is monotonically increasing on $\mathbb{R}_{\geq 0} \times \cdots \times \mathbb{R}_{\geq 0}$, and

$$S(a_1, \cdots, a_i, \cdots, a_d) \leq S(|a_1|, \cdots, |a_i|, \cdots, |a_d|).$$

Below is the general lemma about the poly-smoothness of the neural network.

**Lemma 7.** *Under Assumption 3, $l(\theta; x_i)$ satisfies generalized smoothness*

$$\|\nabla_\theta l(\theta; x_i) - \nabla_\theta l(\theta'; x_i)\| \leq S_l(\|\theta_{\max,1}\|, \cdots, \|\theta_{\max,n_\theta}\|) \|\theta - \theta'\|, \tag{5}$$

*and consequently,*

$$\|\nabla_\theta \mathcal{L}(\theta) - \nabla_\theta \mathcal{L}(\theta')\| \leq S(\|\theta_{\max,1}\|, \cdots, \|\theta_{\max,n_\theta}\|) \|\theta - \theta'\| \tag{6}$$

*where $S_l(\|\theta_{\max,1}\|, \cdots, \|\theta_{\max,n_\theta}\|), S(\|\theta_{\max,1}\|, \cdots, \|\theta_{\max,n_\theta}\|)$ are some polynomials with positive coefficient.*

*Proof.* The proof follows directly from Proposition 2. $\qquad\square$

Next, we aim at deriving the polynomial function precisely under the neural networks where there is normalization $\tau_\epsilon$ in each layer

$$\begin{aligned}
u_{0,i} &= x_i; \\
u_{\ell+1,i} &= u_{\ell,i} + \bar{\varphi}_\ell(\theta_\ell; u_{\ell,i}), \forall \ell = 0, \cdots, L-1; \\
f(\theta; x_i) &= \bar{\varphi}_L(\theta_L; u_{L,i})
\end{aligned} \tag{7}$$

where $\theta = (\theta_0, \cdots, \theta_L)$ is the collection of weights with the $\theta_i$ corresponding to the $i$th layer, and $\bar{\varphi}_\ell(\theta_\ell; u) = \varphi_\ell(\theta_\ell; \tau_\epsilon(u))$ with $\varphi_\ell(\theta_\ell; \cdot) : \mathbb{R}^d \to \mathbb{R}^d$ and the normalization function $\tau_\epsilon$ (see Section 3) for $\ell = 0, \cdots, L$. . Before the precise derivation, we provide a more detailed version of Assumption 3 as follows:

**Assumption 5.** *Assume for all $\ell = 0, \cdots, L$,*

$$\|\varphi_\ell(\theta; u)\| \le \sum_{t=1}^{n_{0,\ell}} C_{0,\ell,t} \prod_{i=1}^{n_\theta} \|\theta_i\|^{p_{0,\ell,t,i}} \|u\|^{p_{0,\ell,t}} \tag{8}$$

$$\|\nabla_{\theta_\ell} \varphi_\ell(\theta; u)\| \le \sum_{t=1}^{n_{1,\ell}} C_{1,\ell,t} \prod_{i=1}^{n_\theta} \|\theta_i\|^{p_{1,\ell,t,i}} \|u\|^{p_{1,\ell,t}} \tag{9}$$

$$\|\nabla_u \varphi_\ell(\theta; u)\| \le \sum_{t=1}^{n_{2,\ell}} C_{2,\ell,t} \prod_{i=1}^{n_\theta} \|\theta_i\|^{p_{2,\ell,t,i}} \|u\|^{p_{2,\ell,t}} \tag{10}$$

$$\|\bar\varphi_\ell(\theta; u) - \bar\varphi_\ell(\theta'; u')\| \le \left( C_{1,1,\ell} \prod_{i=1}^{n_\theta} \|\theta_{\max,i}\|^{p_{1,1,\ell,i}} \|u_{\max}\|^{p_{1,1,\ell}} + C_{1,1,\ell,0} \right) \|\theta - \theta'\|$$
$$+ \left( C_{1,2,\ell} \prod_{i=1}^{n_\theta} \|\theta_{\max,i}\|^{p_{1,2,\ell,i}} \|u_{\max}\|^{p_{1,2,\ell}} + C_{1,2,\ell,0} \right) \|u - u'\| \tag{11}$$

$$\|\nabla_{\theta_\ell} \bar\varphi_\ell(\theta; u) - \nabla_{\theta_\ell} \bar\varphi_\ell(\theta'; u')\| \le \left( C_{2,1,\ell} \prod_{i=1}^{n_\theta} \|\theta_{\max,i}\|^{p_{2,1,\ell,i}} \|u_{\max}\|^{p_{2,1,\ell}} + C_{2,1,\ell,0} \right) \|\theta - \theta'\|$$
$$+ \left( C_{2,2,\ell} \prod_{i=1}^{n_\theta} \|\theta_{\max,i}\|^{p_{2,2,\ell,i}} \|u_{\max}\|^{p_{2,2,\ell}} + C_{2,2,\ell,0} \right) \|u - u'\|$$
$$\tag{12}$$

$$\|\nabla_u \bar\varphi_\ell(\theta; u) - \nabla_u \bar\varphi_\ell(\theta'; u')\| \le \left( C_{3,1,\ell} \prod_{i=1}^{n_\theta} \|\theta_{\max,i}\|^{p_{3,1,\ell,i}} \|u_{\max}\|^{p_{3,1,\ell}} + C_{3,1,\ell,0} \right) \|\theta - \theta'\|$$
$$+ \left( C_{3,2,\ell} \prod_{i=1}^{n_\theta} \|\theta_{\max,i}\|^{p_{3,2,\ell,i}} \|u_{\max}\|^{p_{3,2,\ell}} + C_{3,2,\ell,0} \right) \|u - u'\|$$
$$\tag{13}$$

**Lemma 8.** *Under the Assumption 5, we have that, for all $\ell = 0, \cdots, L$,*

$$\|\bar\varphi_\ell(\theta; u)\| \le \sum_{t=1}^{n_{0,\ell}} C_{0,\ell,t} \prod_{i=1}^{n_\theta} \|\theta_i\|^{p_{0,\ell,t,i}} \tag{14}$$

$$\|\nabla_{\theta_j} \bar\varphi_\ell(\theta; u)\| \le \sum_{t=1}^{n_{1,\ell}} C_{1,\ell,t} \prod_{i=1}^{n_\theta} \|\theta_i\|^{p_{1,\ell,t,i}} \tag{15}$$

$$\|\nabla_u \bar\varphi_\ell(\theta; u)\| \le \sum_{t=1}^{n_{2,\ell}} C_{2,\ell,t} \prod_{i=1}^{n_\theta} \|\theta_i\|^{p_{2,\ell,t,i}} \tag{16}$$

$$\|u_{\ell,i}\| \le \|x_i\| + \sum_{j=0}^{\ell-1} \sum_{t=1}^{n_{0,j}} C_{0,j,t} \prod_{i=1}^{n_\theta} \|\theta_i\|^{p_{0,j,t,i}} \tag{17}$$

$$\|u_{\ell,i}(\theta) - u_{\ell,i}(\theta)\| \le \sum_{j=0}^{g(\ell)} C_{u,j,i} \prod_{q=1}^{n_\theta} \|\theta_{\max,q}\|^{p_{u,j,q}} \|\theta - \theta'\| \tag{18}$$

*Proof.* By the definition of $\epsilon-$normalization, we have

$$\|\tau_\epsilon(u)\| = \left\| \frac{u}{\sqrt{\|u\|^2 + \epsilon^2}} \right\| \le 1.$$

Then we have

$$\|\bar{\varphi}_\ell(\theta; u)\| = \|\varphi(\theta; \tau_\epsilon(u))\| \le \sum_{t=1}^{n_{0,\ell}} C_{0,\ell,t} \prod_{i=1}^{n_\theta} \|\theta_i\|^{p_{0,\ell,t,i}} \|\tau_\epsilon(u)\|^{p_{0,\ell,t}} \le \sum_{t=1}^{n_{0,\ell}} C_{0,\ell,t} \prod_{i=1}^{n_\theta} \|\theta_i\|^{p_{0,\ell,t,i}}.$$

The inequality of $\nabla_{\theta_j} \bar{\varphi}_\ell(\theta; u)$ follows the same idea. For $\nabla_u \bar{\varphi}_\ell(\theta; u)$, first consider

$$\nabla \tau_\epsilon(u) = \frac{1}{\sqrt{\|u\|^2 + \epsilon^2}} I - \frac{1}{(\sqrt{\|u\|^2 + \epsilon^2})^3} u u^\top$$

Then by Wely's theorem,

$$\|\nabla \tau_\epsilon(u)\| \le \frac{1}{\sqrt{\|u\|^2 + \epsilon^2}} + \frac{\|u\|^2}{(\sqrt{\|u\|^2 + \epsilon^2})^3} \le \frac{1}{\epsilon}$$

where the equality of the last inequality is achieved at $\|u\| = 0$. Then

$$\|\nabla_u \bar{\varphi}_\ell(\theta; u)\| = \|\nabla_{\tau_\epsilon(u)} \varphi_\ell(\theta; \tau_\epsilon(u)) \nabla \tau_\epsilon(u)\|$$
$$\le \|\nabla_{\tau_\epsilon(u)} \varphi_\ell(\theta; \tau_\epsilon(u))\| \|\nabla \tau_\epsilon(u)\|$$
$$\le \sum_{t=1}^{n_{2,\ell}} C_{2,\ell,t} \prod_{i=1}^{n_\theta} \|\theta_i\|^{p_{2,\ell,t,i}}$$

where $C_{2,\ell,t}$ depends on $\epsilon$ and the last inequality follows the same idea as the previous two inequalities.

For the upper bound of $u_{\ell,i}$, note that

$$u_{\ell+1,i} = u_{\ell,i} + \bar{\varphi}_\ell(\theta; u_{\ell,i}) = \cdots = u_{0,i} + \sum_{j=0}^{\ell} \bar{\varphi}_j(\theta; u_{j,i}).$$

Then,

$$\|u_{\ell,i}\| \le \|x_i\| + \sum_{j=0}^{\ell-1} \|\bar{\varphi}_j(\theta; u_{j,i})\| \le \|x_i\| + \sum_{j=0}^{\ell-1} \sum_{t=1}^{n_{0,j}} C_{0,j,t} \prod_{i=1}^{n_\theta} \|\theta_i\|^{p_{0,j,t,i}}$$

where the second inequality follows from (14).

Also,

$$\|u_{\ell,i}(\theta) - u_{\ell,i}(\theta')\| = \|x_0 + \sum_{j=0}^{\ell-1} \bar{\varphi}_j(\theta; u_{j,i}(\theta)) - x_0 - \sum_{j=0}^{\ell-1} \bar{\varphi}_j(\theta'; u_{j,i}(\theta'))\|$$
$$\le \sum_{j=0}^{\ell-1} \|\bar{\varphi}_j(\theta; u_{j,i}(\theta)) - \bar{\varphi}_j(\theta'; u_{j,i}(\theta'))\|$$
$$\le \sum_{j=0}^{\ell-1} \left( C_{1,1,j} \prod_{q=1}^{n_\theta} \|\theta_{\max,q}\|^{p_{1,1,j,q}} \|u_{\max}\|^{p_{1,1,j}} + C_{1,1,j,0} \right) \|\theta - \theta'\|$$
$$+ \sum_{j=0}^{\ell-1} \left( C_{1,2,j} \prod_{q=1}^{n_\theta} \|\theta_{\max,q}\|^{p_{1,2,j,q}} \|u_{\max}\|^{p_{1,2,j}} + C_{1,2,j,0} \right) \|u_{j,i}(\theta) - u_{j,i}(\theta')\|$$

where the first inequality follows from (14).

Then we denote the above inequality as follows

$$\underbrace{\|u_{\ell,i}(\theta) - u_{\ell,i}(\theta')\|}_{a_\ell} \le \underbrace{\sum_{j=0}^{\ell-1} \left( C_{1,1,j} \prod_{q=1}^{n_\theta} \|\theta_{\max,q}\|^{p_{1,1,j,q}} \|u_{\max}\|^{p_{1,1,j}} + C_{1,1,j,0} \right) \|\theta - \theta'\|}_{b_\ell}$$
$$+ \underbrace{\sum_{j=0}^{\ell-1} \underbrace{\left( C_{1,2,j} \prod_{q=1}^{n_\theta} \|\theta_{\max,q}\|^{p_{1,2,j,q}} \|u_{\max}\|^{p_{1,2,j}} + C_{1,2,j,0} \right)}_{\lambda_j} \underbrace{\|u_{j,i}(\theta) - u_{j,i}(\theta')\|}_{a_j}}.$$

Since $a_0 = \|x_i - x_i\| = 0$, define $b_0 = 0 \geq a_0$, and then by discrete Gronwall's inequality (Proposition 4.1 in Emmrich (1999) with $\theta = 0, \tau_j = 1$), we have

$$a_\ell \leq b_\ell + \sum_{j=0}^{\ell-1} \lambda_j b_j \prod_{s=j+1}^{\ell-1} (1 + \lambda_s),$$

namely,

$\|u_{\ell,i}(\theta) - u_{\ell,i}(\theta')\|$

$$\leq \bigg( \sum_{j=0}^{\ell-1} \big( C_{1,1,j} \prod_{q=1}^{n_\theta} \|\theta_{\max,q}\|^{p_{1,1,j,q}} \|u_{\max,j,i}\|^{p_{1,1,j}} + C_{1,1,j,0} \big)$$

$$+ \sum_{j=1}^{\ell-1} \big( C_{1,2,j} \prod_{q=1}^{n_\theta} \|\theta_{\max,q}\|^{p_{1,2,j,q}} \|u_{\max,j,i}\|^{p_{1,2,j}} + C_{1,2,j,0} \big)$$

$$\times \big( \sum_{r=0}^{j-1} C_{1,1,r} \prod_{q=1}^{n_\theta} \|\theta_{\max,q}\|^{p_{1,1,r,q}} \|u_{\max,r,i}\|^{p_{1,1,r}} + C_{1,1,r,0} \big)$$

$$\times \prod_{s=j+1}^{\ell-1} \big( 1 + C_{1,2,s} \prod_{q=1}^{n_\theta} \|\theta_{\max,q}\|^{p_{1,2,s,q}} \|u_{\max,s,i}\|^{p_{1,2,s}} + C_{1,2,s,0} \big) \bigg) \|\theta - \theta'\|$$

$$\leq \bigg( \sum_{j=0}^{\ell-1} \big( C_{1,1,j} \prod_{q=1}^{n_\theta} \|\theta_{\max,q}\|^{p_{1,1,j,q}} \big( \sum_{s=0}^{j-1} C_{0,s} \prod_{q=1}^{n_\theta} \|\theta_{\max,q}\|^{p_{0,s,q}} + \|x_i\| \big)^{p_{1,1,s}} + C_{1,1,s,0} \big)$$

$$+ \sum_{j=1}^{\ell-1} \big( C_{1,2,j} \prod_{q=1}^{n_\theta} \|\theta_{\max,q}\|^{p_{1,2,j,q}} \big( \sum_{s=0}^{j-1} C_{0,s} \prod_{q=1}^{n_\theta} \|\theta_{\max,q}\|^{p_{0,s,q}} + \|x_i\| \big)^{p_{1,2,j}} + C_{1,2,j,0} \big)$$

$$\times \big( \sum_{r=0}^{j-1} C_{1,1,r} \prod_{q=1}^{n_\theta} \|\theta_{\max,q}\|^{p_{1,1,r,q}} \big( \sum_{s=0}^{r-1} C_{0,s} \prod_{q=1}^{n_\theta} \|\theta_{\max,q}\|^{p_{0,s,q}} + \|x_i\| \big)^{p_{1,1,r}} + C_{1,1,r,0} \big)$$

$$\times \prod_{s=j+1}^{\ell-1} \big( 1 + C_{1,2,s} \prod_{q=1}^{n_\theta} \|\theta_{\max,q}\|^{p_{1,2,s,q}} \big( \sum_{t=0}^{s-1} C_{0,t} \prod_{q=1}^{n_\theta} \|\theta_{\max,q}\|^{p_{0,t,q}} + \|x_i\| \big)^{p_{1,2,s}} + C_{1,2,s,0} \big) \bigg) \|\theta - \theta'\|$$

$$= \sum_{j=0}^{g_3(\ell)} C_{u,j,i} \prod_{q=1}^{n_\theta} \|\theta_{\max,q}\|^{p_{u,j,q}} \|\theta - \theta'\|$$

where the second inequaliaty follows from (17), $g_3(\ell)$ is some polynomial of $\ell$, and $C_{u,j,i}$ is some constant depending on $\|x_i\|$. □

**Corollary 5.** *Under the same assumptions as Lemma 8, we have that, for all $\ell = 0, \cdots, L$,*

$$\|\bar\varphi_\ell(\theta; u_{\ell,i}(\theta)) - \bar\varphi_\ell(\theta'; u_{\ell,i}(\theta'))\| \leq \sum_{j=0}^{\tilde g_3(\ell)} C_{\bar\varphi,\ell,j,i} \prod_{q=1}^{n_\theta} \|\theta_{\max,q}\|^{p_{\bar\varphi,\ell,j,q}} \|\theta - \theta'\| \tag{19}$$

$$\|\nabla_{\theta_\ell} \bar\varphi_\ell(\theta; u_{\ell,i}(\theta)) - \nabla_{\theta_\ell} \bar\varphi_\ell(\theta'; u_{\ell,i}(\theta'))\| \leq \sum_{j=0}^{\tilde g_3(\ell)} C_{\nabla_\theta \bar\varphi,\ell,j,i} \prod_{q=1}^{n_\theta} \|\theta_{\max,q}\|^{p_{\nabla_\theta \bar\varphi,\ell,j,q}} \|\theta - \theta'\| \tag{20}$$

$$\|\nabla_u \bar\varphi_\ell(\theta; u_{\ell,i}(\theta)) - \nabla_u \bar\varphi_\ell(\theta'; u_{\ell,i}(\theta'))\| \leq \sum_{j=0}^{\tilde g_3(\ell)} C_{\nabla_u \bar\varphi,\ell,j,i} \prod_{q=1}^{n_\theta} \|\theta_{\max,q}\|^{p_{\nabla_u \bar\varphi,\ell,j,q}} \|\theta - \theta'\| \tag{21}$$

*where $\tilde g_3(\ell)$ is some polynomial of $\ell$, and the $C$'s depends on $\|x_i\|$.*

*Proof.* For the first inequality,

$$\|\bar{\varphi}_\ell(\theta; u_{\ell,i}(\theta)) - \bar{\varphi}_\ell(\theta'; u_{\ell,i}(\theta'))\|$$

$$\leq \Big(C_{1,1,\ell} \prod_{q=1}^{n_\theta} \|\theta_{\max,q}\|^{p_{1,1,\ell,q}} \|u_{\max}\|^{p_{1,1,\ell}} + C_{1,1,\ell,0}\Big) \|\theta - \theta'\|$$

$$+ \Big(C_{1,2,\ell} \prod_{q=1}^{n_\theta} \|\theta_{\max,q}\|^{p_{1,2,\ell,q}} \|u_{\max}\|^{p_{1,2,\ell}} + C_{1,2,\ell,0}\Big) \|u_{\ell,i}(\theta) - u_{\ell,i}(\theta')\|$$

$$\leq \Big(C_{1,1,\ell} \prod_{q=1}^{n_\theta} \|\theta_{\max,q}\|^{p_{1,1,\ell,q}} \Big(\sum_{s=0}^{j-1} C_{0,s} \prod_{q=1}^{n_\theta} \|\theta_{\max,q}\|^{p_{0,s,q}} + \|x_i\|\Big)^{p_{1,1,\ell}} + C_{1,1,\ell,0}\Big) \|\theta - \theta'\|$$

$$+ \Big(C_{1,2,\ell} \prod_{q=1}^{n_\theta} \|\theta_{\max,q}\|^{p_{1,2,\ell,q}} \Big(\sum_{s=0}^{j-1} C_{0,s} \prod_{q=1}^{n_\theta} \|\theta_{\max,q}\|^{p_{0,s,q}} + \|x_i\|\Big)^{p_{1,2,\ell}} + C_{1,2,\ell,0}\Big)$$

$$\times \sum_{j=0}^{g_3(\ell)} C_{u,j,i} \prod_{q=1}^{n_\theta} \|\theta_{\max,q}\|^{p_{u,j,q}} \|\theta - \theta'\|$$

$$= \sum_{j=0}^{\tilde{g}_3(\ell)} C_{\bar{\varphi},\ell,j,i} \prod_{q=1}^{n_\theta} \|\theta_{\max,q}\|^{p_{\bar{\varphi},\ell,j,q}} \|\theta - \theta'\|$$

where the second inequality follows from (17) and (18), $\tilde{g}_3(\ell)$ is some polynomial of $\ell$, $C_{\bar{\varphi},\ell,j,i}$ depends on $\|x_i\|$. The other two inequalities follow the same idea. $\qquad\square$

**Lemma 9.** *Under Assumption 5, $l(\theta; x_i)$ satisfies generalized smoothness*

$$\|\nabla_\theta l(\theta; x_i) - \nabla_\theta l(\theta'; x_i)\| \leq \sum_{j=1}^{n_\theta} \sum_{\ell \in \mathcal{J}_j} \sum_{j=0}^{g_4(\ell)} C_{l,\ell,j,i} \prod_{q=1}^{n_\theta} \|\theta_{\max,q}\|^{p_{l,\ell,j,q}} \|\theta - \theta'\| \tag{22}$$

*where $g_4(\ell)$ is some polynomial of $\ell$. Consequently,*

$$\|\nabla_\theta \mathcal{L}(\theta) - \nabla_\theta \mathcal{L}(\theta')\| \leq S(\|\theta_{\max,1}\|, \cdots, \|\theta_{\max,n_\theta}\|) \|\theta - \theta'\| \tag{23}$$

*where $S(\|\theta_{\max,1}\|, \cdots, \|\theta_{\max,n_\theta}\|) = \frac{1}{N} \sum_{i=1}^{N} \sum_{j=1}^{n_\theta} \sum_{\ell \in \mathcal{J}_j} \sum_{j=0}^{g_4(\ell)} C_{l,\ell,j,i} \prod_{q=1}^{n_\theta} \|\theta_{\max,q}\|^{p_{l,\ell,j,q}}$.*

*Proof.* From the calculations at the beginning of this section, we know

$$\nabla_{\theta_j} l(\theta; x_i) = \nabla_f l(\theta; x_i) \nabla_{\theta_\ell} f(x_i)$$
$$= \sum_{\ell \in \mathcal{J}_j} \nabla_f l(\theta; x_i) \nabla_u \bar{\varphi}_L(\theta; u_{L,i}) (I + \nabla_u \bar{\varphi}_{L-1}(\theta; u_{L-1,i})) \cdots (I + \nabla_u \bar{\varphi}_{\ell+1}(\theta; u_{\ell+1,i})) \nabla_{\theta_j} \bar{\varphi}_\ell(\theta; u_{\ell,i})$$

Then

$$\|\nabla_{\theta_j} l(\theta; x_i) - \nabla_{\theta_j} l(\theta'; x_i)\|$$

$$\leq \sum_{\ell \in \mathcal{J}_j} \Big\| \nabla_f l(\theta; x_i) \nabla_u \bar{\varphi}_L(\theta; u_{L,i})(I + \nabla_u \bar{\varphi}_{L-1}(\theta; u_{L-1,i})) \cdots (I + \nabla_u \bar{\varphi}_{\ell+1}(\theta; u_{\ell+1,i})) \nabla_{\theta_j} \bar{\varphi}_\ell(\theta; u_{\ell,i})$$

$$- \nabla_f l(\theta'; x_i) \nabla_u \bar{\varphi}_L(\theta'; u_{L,i})(I + \nabla_u \bar{\varphi}_{L-1}(\theta'; u_{L-1,i})) \cdots (I + \nabla_u \bar{\varphi}_{\ell+1}(\theta'; u_{\ell+1,i})) \nabla_{\theta_j} \bar{\varphi}_\ell(\theta'; u_{\ell,i}) \Big\|$$

$$= \sum_{\ell \in \mathcal{J}_j} \Big\| (\bar{\varphi}_L(\theta; u_{L,i}) - y_i)^\top \nabla_u \bar{\varphi}_L(\theta; u_{L,i})(I + \nabla_u \bar{\varphi}_{L-1}(\theta; u_{L-1,i})) \cdots (I + \nabla_u \bar{\varphi}_{\ell+1}(\theta; u_{\ell+1,i})) \nabla_{\theta_j} \bar{\varphi}_\ell(\theta; u_{\ell,i})$$

$$- (\bar{\varphi}_L(\theta'; u_{L,i}) - y_i)^\top \nabla_u \bar{\varphi}_L(\theta'; u_{L,i})(I + \nabla_u \bar{\varphi}_{L-1}(\theta'; u_{L-1,i})) \cdots (I + \nabla_u \bar{\varphi}_{\ell+1}(\theta'; u_{\ell+1,i})) \nabla_{\theta_j} \bar{\varphi}_\ell(\theta'; u_{\ell,i}) \Big\|$$

$$\leq \sum_{\ell \in \mathcal{J}_j} (\|\bar{\varphi}_L(\theta; u_{L,i})\| + \|y_i\|) \|\nabla_u \bar{\varphi}_L(\theta; u_{L,i})\|(1 + \|\nabla_u \bar{\varphi}_{L-1}(\theta; u_{L-1,i})\|) \cdots$$

$$\times (1 + \|\nabla_u \bar{\varphi}_{\ell+1}(\theta; u_{\ell+1,i}))\|) \|\nabla_{\theta_j} \bar{\varphi}_\ell(\theta; u_{\ell,i}) - \nabla_{\theta_j} \bar{\varphi}_\ell(\theta'; u_{\ell,i})\|$$

$$+ \cdots + (\|\bar{\varphi}_L(\theta; u_{L,i})\| + \|y_i\|) \|\nabla_u \bar{\varphi}_L(\theta; u_{L,i})\|(1 + \|\nabla_u \bar{\varphi}_{L-1}(\theta; u_{L-1,i})\|) \cdots$$

$$\times \|\nabla_u \bar{\varphi}_s(\theta; u_{s,i}) - \nabla_u \bar{\varphi}_s(\theta'; u_{s,i})\| \cdots (1 + \|\nabla_u \bar{\varphi}_{\ell+1}(\theta'; u_{\ell+1,i})\|) \|\nabla_{\theta_j} \bar{\varphi}_\ell(\theta'; u_{\ell,i})\|$$

$$+ \cdots + \|\bar{\varphi}_L(\theta; u_{L,i}) - \bar{\varphi}_L(\theta'; u_{L,i})\| \|\nabla_u \bar{\varphi}_L(\theta'; u_{L,i})\|(1 + \|\nabla_u \bar{\varphi}_{L-1}(\theta'; u_{L-1,i})\|) \cdots$$

$$\times (1 + \|\nabla_u \bar{\varphi}_{\ell+1}(\theta'; u_{\ell+1,i})\|) \|\nabla_{\theta_j} \bar{\varphi}_\ell(\theta'; u_{\ell,i})\|$$

$$\leq \sum_{\ell \in \mathcal{J}_j} \sum_{j=0}^{g_4(\ell)} C_{l,\ell,j,i} \prod_{q=1}^{n_\theta} \|\theta_{\max,q}\|^{p_{l,\ell,j,q}} \|\theta - \theta'\|$$

where the second inequality follows from triangular inequality, and the last inequality follows from Lemma 8 and Corollary 5; $C_{l,\ell,j,i}$ is a constant that depends on $\|x_i\|$ and $\|y_i\|$, and $g_4(\ell)$ is a polynomial of $\ell$.

Then

$$\|\nabla_\theta l(\theta; x_i) - \nabla_\theta l(\theta'; x_i)\| \leq \sum_{j=1}^{n_\theta} \|\nabla_{\theta_j} l(\theta; x_i) - \nabla_{\theta_j} l(\theta'; x_i)\|$$

$$\leq \sum_{j=1}^{n_\theta} \sum_{\ell \in \mathcal{J}_j} \sum_{j=0}^{g_4(\ell)} C_{l,\ell,j,i} \prod_{q=1}^{n_\theta} \|\theta_{\max,q}\|^{p_{l,\ell,j,q}} \|\theta - \theta'\|$$

and

$$\|\nabla_\theta \mathcal{L}(\theta) - \nabla_\theta \mathcal{L}(\theta')\| \leq \frac{1}{N} \sum_{i=1}^{N} \|\nabla_\theta l(\theta; x_i) - \nabla_\theta l(\theta'; x_i)\|$$

$$\leq \frac{1}{N} \sum_{i=1}^{N} \sum_{j=1}^{n_\theta} \sum_{\ell \in \mathcal{J}_j} \sum_{j=0}^{g_4(\ell)} C_{l,\ell,j,i} \prod_{q=1}^{n_\theta} \|\theta_{\max,q}\|^{p_{l,\ell,j,q}} \|\theta - \theta'\|$$

$$\square$$

### E.3 ADDITIONAL LEMMAS IN MEASURE THEORY AND DIFFERENTIAL TOPOLOGY

In this section, we first introduce some basic concepts and theorems in measure theory and differential topology, and then state the supplementary lemmas of the proof.

#### E.3.1 BASIC DEFINITIONS AND THEOREMS

We first introduce the parametric transversality theorem (Theorem 6), together with the necessary concepts including residual set and transversality. We then state other useful theorems that will be applied in the lemmas of the following section

We first introduce the notions of nowhere dense set and meager set in order to define the residual set.

**Definition 7** (Nowhere dense set). *A set is called nowhere dense if it is not dense in any nonempty open subset, or equivalently, the interior of its closure is empty.*

**Definition 8** (Meager set). *A set is called meager if it is a countable union of nowhere dense sets.*

**Definition 9** (Residual set). *A set is called residual if its complement is a countable union of nowhere dense sets, or equivalently, its complement is meager.*

**Proposition 3.** *The meager set and residual set have the following properties:*

*1. All countable unions of meager sets are meager.*

*2. Consider a continuous function $f : (a, \infty) \to \mathbb{R}$ for some $a > 0$. Then if $X \subseteq (a, \infty)$ is residual in $(a, \infty)$, then $f(X)$ is residual in $(0, 1/a)$.*

We then introduce transversality of a map.

**Definition 10.** *(Transversality) A map $f : M \to N$ is transversal to a submanifold $A \subseteq N$ if for any $x \in f^{-1}(A)$, we have $\mathrm{Im}(Df_x) + T_{f(x)}A = T_{f(x)}N$, i.e., the tangent space of $N$ at $f(x)$ is spanned by the image of the differential of $f$ at $x$ and the tangent space of $A$ at $f(x)$.*

In this paper, we only consider the case where the submanifold $A = \{0\}$. Then, transversality is equivalent to non-degeneracy of the map.

**Theorem 6** (Parametric transversality theorem (Hirsch, 2012)). *Let $V, M, N$ be $C^r$ manifold without boundary and $A \subset N$ a $C^r$ submanifold. Let $F : V \to C^r(M, N)$ satisfy the following conditions:*

- *$F^{\mathrm{ev}} : V \times M \to N$, $(v, x) \mapsto F_v(x)$, is $C^r$;*

- *$F^{\mathrm{ev}}$ is transverse to $A$;*

- *$r > \max\{0, \dim N + \dim A - \dim M\}$.*

*Then the set $\{v \in V : F_v \text{ is transverse to } A\}$ is residual and therefore dense.*

In addition to the parametric transversality theorem, we also employ the following useful results.

**Theorem 7** (Mityagin (2015)). *Let $f(x)$ be a real analytic function on (a connected open domain $U$ of) $\mathbb{R}^d$. If $f$ is not identically zero, then its zero set*

$$f^{-1}(0) = \{x \in U : f(x) = 0\}$$

*has a zero measure, i.e., $\mathrm{Leb}(f^{-1}(0)) = 0$.*

**Theorem 8** (Theorem G.3 in Wang et al. (2022b)). *Let $f : \mathbb{R}^d \to \mathbb{R}^d$ and $f \in \mathcal{C}^1$. If the set of critical points of $f$ is a null-set, i,e.,*

$$\mathcal{L}(\{x \in \mathbb{R}^d : \nabla f(x) \text{ is not invertible}\}) = 0,$$

*then $\mathcal{L}(f^{-1}(B)) = 0$ for any null-set $B$.*

**Theorem 9** (Lemma 3 in Rader (1973)). *Consider a function $f : U \subseteq \mathbb{R}^d \to \mathbb{R}^d$ that is pointwise Lipschitz, i.e., for any $x \in U$, there exists a neighbourhood $U_x$ of $x$ and a constant $L_x$, s.t.*

$$\|f(x) - f(y)\| \le L_x \|x - y\|, \ \forall y \in U_x.$$

*Then its image of a measure-zero set is still measure zero.*

**Theorem 10** (Preimage Theorem). *If $y$ is a regular value of a smooth map $f : X \to Y$, then the preimage $f^{-1}(y)$ is a submanifold of $X$, with $\dim f^{-1}(y) = \dim X - \dim Y$.*

### E.3.2 SUPPLEMENTARY LEMMAS OF THE PROOF

We then show the supplementary lemmas for the convergence of neural network (1) by applying the above theorems.

**Lemma 10.** *Fix some $\theta$. Under Assumption 2,*

*1. Let $u_\ell = u_\ell(x) : \mathbb{R}^d \to \mathbb{R}^d$. The preimage of $u_\ell$ for any measure-zero set in $\mathbb{R}^d$ is a measure-zero set in $\mathbb{R}^d$, for all $\ell = 0, \cdots, L - 1$.*

2. Let $U_\ell(x_1, \cdots, x_n) = \begin{pmatrix} u_\ell(x_1) \\ :u_\ell(x_N) \end{pmatrix} : \mathbb{R}^{Nd} \to \mathbb{R}^{Nd}$. The preimage of $U_\ell$ for any measure-zero set in $\mathbb{R}^{Nd}$ is a measure-zero set in $\mathbb{R}^{Nd}$, for all $\ell = 0, \cdots, L-1$.

*Proof.* Suppose the critical point set of $u_\ell$ is measure zero, and thus by Theorem 8, the preimage of $u_\ell$ for any measure zero set is a null set. Consider

$$u_{\ell+1} = u_{\ell+1}(x),$$

where its Jacobian is

$$\nabla u_{\ell+1}(x) = (I + \nabla_u \bar{\varphi}_\ell(u_\ell)) \cdots (I + \nabla_u \bar{\varphi}_0(u_0)).$$

We would like to prove that the critical point set of $u_{\ell+1}$ is measure zero, i.e.,

$$\{x \in \mathbb{R}^d : \nabla u_{\ell+1}(x) \text{ is not full rank}\}.$$

Note that

$$\nabla u_{\ell+1}(x) = (I + \nabla_u \bar{\varphi}_\ell(u_\ell)) \nabla u_\ell(x),$$

and thus

$$\begin{aligned} &\{x \in \mathbb{R}^d : \nabla u_{\ell+1}(x) \text{ is not full rank}\} \\ &= \{x \in \mathbb{R}^d : I + \nabla_u \bar{\varphi}_\ell(u_\ell(x)) \text{ is not full rank}\} \cup \{x \in \mathbb{R}^d : \nabla u_\ell(x) \text{ is not full rank}\}. \end{aligned}$$

By our assumption, the set

$$\text{Leb}_d\big(\{x \in \mathbb{R}^d : \nabla u_\ell(x) \text{ is not full rank}\}\big) = 0.$$

Also, by Lemma 11,

$$\text{Leb}_d\big(\{u \in \mathbb{R}^d : I + \nabla_u \bar{\varphi}_\ell(u) \text{ is not full rank}\}\big) = 0,$$

and then by Theorem 8, the preimage of the above measure zero set is still measure zero, i.e.,

$$\text{Leb}_d\big(\{x \in \mathbb{R}^d : I + \nabla_u \bar{\varphi}_\ell(u_\ell(x)) \text{ is not full rank}\}\big) = 0.$$

Thus the critical point set of $u_{\ell+1}$ is measure zero, i.e.

$$\text{Leb}_d\big(\{x \in \mathbb{R}^d : \nabla u_{\ell+1}(x) \text{ is not full rank}\}\big) \le 0 + 0 = 0.$$

By Theorem 8, the first conclusion holds for $u_{\ell+1}$.

For the second statement, the Jacobian of $U$ is

$$\begin{aligned} \nabla U_{\ell+1}(x_1, \cdots, x_N) &= \begin{pmatrix} \nabla u_{\ell+1}(x_1) & & \\ & \ddots & \\ & & \nabla u_{\ell+1}(x_N) \end{pmatrix} \\ &= \begin{pmatrix} I + \nabla_u \bar{\varphi}_\ell(u_\ell(x_1)) & & \\ & \ddots & \\ & & I + \nabla_u \bar{\varphi}_\ell(u_\ell(x_N)) \end{pmatrix} \nabla U_\ell(x_1, \cdots, x_N) \\ &= \left( I_{Nd} + \begin{pmatrix} \nabla_u \bar{\varphi}_\ell(u_\ell(x_1)) & & \\ & \ddots & \\ & & \nabla_u \bar{\varphi}_\ell(u_\ell(x_N)) \end{pmatrix} \right) \nabla U_\ell(x_1, \cdots, x_N). \end{aligned}$$

Then the second statement follows a similar derivation as the first one. $\quad\square$

**Lemma 11.** *Let $f(x, y) : \mathbb{R}^m \times \mathbb{R}^{n_2} \to \mathbb{R}^m$ be an analytic function. Consider $g(x, y, \alpha) = \alpha x + f(x, y)$ where $\alpha \ge c > 0$ for some constant $c < 1$. Then the following statements hold for (1) fixed $y$ and any $x$ except for a measure-zero set, and (2) fixed $x$ and any $y$ except for a measure-zero set:*

1. *There exists a residual set in the neighborhood of 0, such that, when $\epsilon_0$ is in this set, the Jacobian w.r.t. $x$ of either $x + f(x, y)$ or $x + (1 + \epsilon_0) f(x, y)$ is full rank.*

2. *There exists a residual set of $(0, \frac{1}{c})$, such that, when $\eta$ is in this residual set, the Jacobian of $x + \eta f(x, y)$ w.r.t. $x$ is full rank.*

*Proof.* First, consider

$$F(\alpha, x, y) = \det\left(\alpha I + \nabla_x f(x, y)\right) = \prod_{i=1}^{m}(\alpha + \lambda_i(x, y)),$$

where $\lambda_i(x, y)$ is the eigenvalue of $\nabla_x f(x, y)$. By Proposition 1, $F(\alpha, x, y)$ is a analytic function of $\alpha, x, y$.

Fix $x, y$. Then $F(\alpha, x, y)$ is a polynomial of $\alpha$ and the critical points of $F(\alpha, x, y)$ are isolated. Thus $F'_\alpha(\alpha, x, y) \neq 0$ for all alpha except for a measure 0 set of isolated points.

Next we no longer fix $x, y$, let $A = \{0\}$, and consider any open interval $I$ of $\alpha$ that does not contain the critical points, which is a manifold. Then from the above discussion, $\nabla F(\alpha, x, y) \neq \mathbf{0}$, namely $F$ is surjective and thus transverse to $A$. Therefore, by Theorem 6, the set $\mathcal{S}_\alpha = \{\alpha \in I : F_\alpha$ is transverse to $A\}$ is residual and dense, which also implies that $F_\alpha(x, y) = \det\left(\alpha I + \nabla_x f(x, y)\right)$ is transverse to $A$ for a residual set of $\alpha$ in $(c, \infty)$. Then, for any fixed $\alpha$ in this residual set, $\nabla F_\alpha(x, y)$ is surjective for any $(x, y) \in F_\alpha^{-1}(0)$, and hence 0 is a regular point. Then by Theorem 10, $F_\alpha^{-1}(0)$ is a submanifold of codimension 1, and therefore

$$\mathrm{Leb}_{mn_2}(F_\alpha^{-1}(0)) = 0.$$

Then we consider fixing $x$ or $y$, i.e., $F_{\alpha,x}(y)$ and $F_{\alpha,y}(x)$. The above reasoning also holds true for the two functions, and therefore

$$\mathrm{Leb}_m(F_{\alpha,y}^{-1}(0)) = 0, \text{ for some fixed } y, \text{ and } \mathrm{Leb}_{n_2}(F_{\alpha,x}^{-1}(0)) = 0, \text{ for some fixed } x.$$

This concludes the second statement by choosing $\eta = \frac{1}{\alpha}$, which is also residual in $(0, \frac{1}{c})$ by Proposition 3.

For the first statement, if -1 is not an eigenvalue of $\nabla_x f(x, y)$, then $I + \nabla_x f(x, y)$ has full rank. Otherwise, since $\alpha$ is dense in any open set that does not contain 1, we can always find a small enough $\epsilon_1$, such that $1 - \epsilon_1$ is inside this set $\mathcal{S}_\alpha$. Then $I + \frac{1}{1-\epsilon_1}\nabla_x f(x, y)$ is has full rank. The first statement follows from choosing $\epsilon_0 = \frac{1}{1-\epsilon_1} - 1$.

$\square$

The following lemmas are not directly used in the proof but reflect properties of this neural network, so we include them here.

**Lemma 12.** *For any $(x_1, \cdots, x_N)$ except for a measure-zero set in $\mathbb{R}^{Nd}$, the Jacobian along the line segment between any two points $u_{\ell,i}, u_{\ell,j}$ has full rank at every layer $\ell = 0, \cdots, L - 1$. Furthermore, for any point $u_\ell$ on the line segment between $x_i$ and $x_j$, the Jacobian at any layer $\ell = 0, \cdots, L - 1$ also has full rank.*

*Proof.* First, by Lemma 10, the critical point set of all $u_\ell$, denoted by $\mathcal{B}$, is a finite union of measure-zero sets in $\mathbb{R}^d$, and thus is measure zero, i.e.,

$$\mathrm{Leb}_d(\mathcal{B}) = \mathrm{Leb}_d\left(\bigcup_{\ell=0}^{L-1}\{x \in \mathbb{R}^d : \nabla_u u_\ell(x) \text{ is not full rank}\}\right)$$

$$\leq \sum_{\ell=0}^{L-1} \mathrm{Leb}_d\left(\{x \in \mathbb{R}^d : \nabla_u u_\ell(x) \text{ is not full rank}\}\right) = 0.$$

Next, we would like to show that the following set is measure zero in $\mathbb{R}^{Nd}$

$$\mathcal{D} = \{(x_1, \cdots, x_n) : \{tu_\ell(x_i) + (1-t)u_\ell(x_j) : t \in [0, 1]; i, j = 1, \cdots, N\} \cap u_\ell(\mathcal{B}) \neq \varnothing \text{ for some } \ell = 0, \cdots, L-1\}$$

Let

$$\bar{U}(\mathcal{B}) = \bigcup_{\ell=0}^{L-1} u_\ell(\mathcal{B}), \text{ and } \bar{\mathcal{B}} = \bigcup_{\ell=0}^{L-1}\{x \in u_\ell^{-1}(\bar{U}(\mathcal{B}))\}.$$

Since $u_\ell(x)$ is analytic, and thus differentiable, then $u_\ell(x)$ is pointwise Lipschitz. By Theorem 9, we have

$$\text{Leb}_d(\bar{U}(\mathcal{B})) \leq \sum_{\ell=0}^{L-1} \text{Leb}_d(u_\ell(\mathcal{B})) = 0.$$

Then by Theorem 8,

$$\text{Leb}_d(\bar{\mathcal{B}}) \leq \sum_{\ell=0}^{L-1} \text{Leb}_d(u_\ell^{-1}(\bar{U}(\mathcal{B}))) = 0.$$

We consider

$$\mathcal{D} \subseteq \left\{(x_1,\cdots,x_n) : \{tu_\ell(x_i) + (1-t)u_\ell(x_j) : t \in [0,1]; i,j = 1,\cdots,N; \ell = 0,\cdots,L-1\} \cap \bar{U}(\mathcal{B}) \neq \varnothing\right\}$$

$$\subseteq \bigcup_{\ell=0}^{L-1} \bigcup_{i,j} \bigcup_{z \in \bar{U}(\mathcal{B})} \left\{(x_1,\cdots,x_N) : z \in \{tu_\ell(x_i) + (1-t)u_\ell(x_j) : t \in [0,1]\}\right\}$$

We denote $\mathcal{D}_{i,j,\ell,z} = \left\{(x_1,\cdots,x_N) : z \in \{tu_\ell(x_i) + (1-t)u_\ell(x_j) : t \in [0,1]\}\right\}$ and consider the measure of $\bigcup_{z \in \bar{U}} \mathcal{D}_{i,j,\ell,z}$, i.e.,

$$\text{Leb}_{Nd}\left(\bigcup_{z \in \bar{U}} \mathcal{D}_{i,j,\ell,z}\right) \leq \int \cdots \int \mathbb{1}_{z \in \{tu_\ell(x_i)+(1-t)u_\ell(x_j):t \in [0,1]\}} dx_1 \cdots dx_N dz$$

By Fubini-Tonelli's Theorem,

$$\text{Leb}_{Nd}\left(\bigcup_{z \in \bar{U}} \mathcal{D}_{i,j,\ell,z}\right) \leq \int \cdots \int \mathbb{1}_{z \in \{tu_\ell(x_i)+(1-t)u_\ell(x_j):t \in [0,1]\}} dx_i dz dx_j \cdots dx_N$$

For any fixed $x_j$ and $z$, the set $\{u_\ell(x_i) : z \in \{tu_\ell(x_i) + (1-t)u_\ell(x_j) : t \in [0,1]\}\}$ is a ray in $\mathbb{R}^d$ and thus is measure zero. Therefore, its preimage is measure-zero in $\mathbb{R}^d$. Also, from the above derivation, $\text{Leb}_d(\bar{U}(\mathcal{B})) = 0$. Then we have

$$\text{Leb}_{Nd}\left(\bigcup_{z \in \bar{U}} \mathcal{D}_{i,j,\ell,z}\right) \leq 0.$$

We then consider $\bigcup_{\ell=0}^{L-1} \bigcup_{i,j} \bigcup_{z \in \bar{U}(\mathcal{B})} \mathcal{D}_{i,j,\ell,z}$. Since there are $\binom{N}{2}$ pair of $(i,j)$ and $L$ such $\ell$, which are both finite, we have

$$\text{Leb}_{Nd}(\mathcal{D}) \leq \text{Leb}_{Nd}\left(\bigcup_{\ell=0}^{L-1} \bigcup_{i,j} \bigcup_{z \in \bar{U}(\mathcal{B})} \mathcal{D}_{i,j,\ell,z}\right) \leq \sum_{\ell=0}^{L-1} \sum_{(i,j)} 0 = 0$$

Next, we would like to show that the following set is measure zero in $\mathbb{R}^{Nd}$

$$\mathcal{D} = \left\{(x_1,\cdots,x_n) : \{u_\ell(tx_i + (1-t)x_j) : t \in [0,1]; i,j = 1,\cdots,N\} \cap u_\ell(\mathcal{B}) \neq \varnothing \text{ for some } \ell = 0,\cdots,L-1\right\}.$$

The proof follows the same idea as the previous statement. The only difference is that for any fixed $x_j$ and $z$, the set $\{x_i : z \in \{u_\ell(tx_i + (1-t)x_j) : t \in [0,1]\}\}$ is a ray in $\mathbb{R}^d$ and thus is measure zero.

$\square$

**Lemma 13.** *Fix $\theta$ and consider the network under some small adjustment of the scale of $\varphi_\ell$ for $\ell = 0,\cdots,L-1$. For any $(x_1,\cdots,x_N)$ except for a measure-zero set in $\mathbb{R}^{Nd}$, each layer satisfies the following lower Lipschitz inequality*

$$\|u_{\ell,i} - u_{\ell,j}\| \geq \mu_{u,\ell,\theta}\|x_j - x_i\|, \ \forall \ell = 0,\cdots,L-1, \ \forall i,j = 1,\cdots,N.$$

*where $\mu_{u,\ell,\theta} > 0$ is some constant depending on $\theta,\ell$.*

*Proof.* First, by Lemma 12, the Jacobians of $u_\ell(x)$ the line segments between each point $u_{\ell,i}, u_{\ell,j}$ are full rank for all $\ell = 0,\cdots,L-1$, except for a measure-zero set of $X = (x_1,\cdots,x_N)$ in $\mathbb{R}^{Nd}$, denoted by $\mathcal{B}_{Nd}$.

Then for fixed $\theta$ and $X = (x_1, \cdots, x_N) \notin \mathcal{B}_{Nd}$, there exists some constant $\mu_{u,\ell,\theta} > 0$, s.t.

$$\mu_{u,\ell,\theta} = \min_{\{x : u_\ell(x) = t u_\ell(x_i) + (1-t) u_\ell(x_j), \, t \in [0,1]\}} \lambda_{\min}(\nabla_x u_\ell(x))$$

where $\lambda_{\min}$ is the minimum singular value.

Then

$$\|u_\ell(x_i) - u_\ell(x_j)\| = \left\| \int_0^1 \nabla u_\ell(x_i + t(x_j - x_i))(x_j - x_i) \, dt \right\|$$

$$= \left\| \int_0^1 \nabla u_\ell(x_i + t(x_j - x_i)) \, dt \cdot (x_j - x_i) \right\|$$

$$\geq \mu_{u,\ell,\theta} \|x_j - x_i\|.$$

$\square$

## F  APPROXIMATION: PROOF OF THEOREM 3

Below is the formal version of Theorem 3, and we will prove it in this section.

**Theorem 11.** *Consider a set of data $\{(x_i, y_i)\}_{i=1}^N$ under Assumption 1. Let $\mathcal{I}_i = \operatorname{conv}\{x_{i_1}, \cdots, x_{i_{d+1}}\}$ with $x_j \notin \mathcal{I}_i$, $\forall j \neq i_1, \cdots, i_{d+1}$, and define $h_{\max_{d+1}} = \max_i \max_{x_q, x_j \in \mathcal{I}_i} h_{qj}$. Suppose there is a family of neural networks that can approximate polynomial of degree $n_d = n_d(N, d)$, i.e., for any $\epsilon_2 > 0$ and $\psi \in P_{n_d}$, there exists some $f$ s.t. $\|f - \psi\|_p \leq \epsilon_2$. Then, for some $C_1 > 0$ and $m > \max\{r, n_d\}$, there exists some $\phi \in W^{m,p}(\Omega)$ satisfying $\|g - \phi\|_{r,p} \leq C_2 h_{\max_{d+1}}^m h_{\min}^{-r} \|\phi\|_{m,p,\mathcal{I}^\circ}$, s.t., with probability 1 over the joint distribution of $x_1, \cdots, x_N$, we have*

$$\|f - g\|_{p,\mathcal{I}^\circ} \leq C_1 \max\{h_{\max_{n+1}}, 1\}^m \min\{h_{\min}, 1\}^{-r} \|\phi\|_{m,p}.$$

*Proof of Theorem 11.* Consider the ground truth function $g \in W^{r,p}(\Omega)$. Since $W^{m,p}(\Omega)$ is dense in $W^{r,p}(\Omega)$ for $m \geq r$, we have that for any $\epsilon_1 > 0$, there exists $\phi \in W^{m,p}(\Omega)$ such that

$$\|g - \phi\|_{r,p} \leq \epsilon_1.$$

Also, by Lemma 14 and Theorem 13, there exists a polynomial $\psi$ of order at most $n_d(N, d)$, s.t.,

$$\|\phi - \psi\|_{r,p,\mathcal{I}^\circ}^p \leq (C \max\{h_{\max_{n+1}}, 1\}^m \min\{h_{\min}, 1\}^{-r} \|\phi\|_{m,p})^p$$

For a family of neural network functions that can approximate polynomial of order $n_d$, i.e., for any $\epsilon_2 > 0$ and $\psi \in P_{n_d}$, there exists some $f$ in this class s.t.

$$\|f - \psi\|_p \leq \epsilon_2$$

Let $\epsilon_1 = \epsilon_2 = \frac{1}{2} C \max\{h_{\max_{n+1}}, 1\}^m \min\{h_{\min}, 1\}^{-r} \|\phi\|_{m,p}$. Then combining all the results above, we have

$$\|f - g\|_{p,\mathcal{I}^\circ} \leq \|f - \psi\|_p + \|\psi - \phi\|_p + \|\phi - g\|_p$$

$$\leq \|f - \psi\|_p + \|\psi - \phi\|_{r,p} + \|\phi - g\|_{r,p}$$

$$\leq 2C \max\{h_{\max_{n+1}}, 1\}^m \min\{h_{\min}, 1\}^{-r} \|\phi\|_{m,p}.$$

$\square$

### F.1  DEFINITIONS AND LEMMAS RELATED TO COMPUTATIONAL GEOMETRY

In this section, we introduce some basic definitions and concepts, mainly based on De Berg (2000).

**Definition 11** (Triangulation). *A triangulation of the polygon is defined as a decomposition of a polygon into triangles by a maximal set of non-intersecting diagonals, or equivalently, no vertex of any triangle lies in the interior of an edge of another triangle.*

**Definition 12** (Delaunay triangulation). *Given a set $\mathcal{P}$ of points in the $n$-dimensional Euclidean space, a Delaunay triangulation is a triangulation DT($\mathcal{P}$) such that if for every $n$-simplex (whose vertices belong to $\mathcal{P}$) in the triangulation, there exists an open $n$-ball that passes through all the vertices of the simplex and that does not contain any other point of $\mathcal{P}$ in its interior.*

**Definition 13** ($n$-simplex). *A $n-$simplex in $n$-dimensional space is the convex hull of $n + 1$ affinely independent points in $\mathbb{R}^n$.*

**Proposition 4.** *Let $\{v_0, v_1, \ldots, V_d\} \subset \mathbb{R}^n$ be the vertices of an $n$-simplex. Let $u_i = v_i - v_0$, for $i = 1, \ldots, n$.*

- *$n + 1$ vertices, $\binom{n+1}{2}$ edges, $\binom{n+1}{k+1}k$-dimensional faces*

- *volumn $V = \frac{1}{n!} |\det([u_1\, u_2 \cdots u_n])|$*

**Definition 14** (General position). *The set of points $\{x_i\}_{i=1}^N$ in $\mathbb{R}^n$ is in general position if any $n + 1$ points are affinely independent.*

**Theorem 12** (Theorem 3.1, and Chapter 9, in De Berg (2000)). *Assume the set $\{x_i\}_{i=1}^N$ is in general position. Then there exists a unique Delaunay triangulation, which consists of $n$-simplices.*

We then prove the following lemma, showing that i.i.d. samples obeying absolutely continuous density are in general position with probability one.

**Lemma 14.** *Suppose $x_1, \cdots, x_N \in \mathbb{R}^n$ are sampled iid from a distribution $P$ and $P \ll$ Leb. Then with probability one over the joint distribution, the set $\{x_i\}_{i=1}^N$ is in general position, i.e., any $n + 1$ points are affinely independent.*

*Proof.* First, a set of $n+1$ points $x_0, \cdots, x_n$ is affinely independent if and only if the following vectors $x_1 - x_0, \cdots, x_n - x_0$ are linearly independent. This is equivalent to

$$\det([x_1 - x_0, \cdots, x_n - x_0]) \neq 0.$$

Let $F(x_0, \cdots, x_n) = \det([x_1 - x_0, \cdots, x_n - x_0])$. Then $F$ is a polynomial. Let

$$\mathcal{N} = \{(x_0, \cdots, x_n) | F(x_0, \cdots, x_n) = 0\}.$$

Then the zero set of the polynomial is a null set in $\mathbb{R}^{(n+1)d}$, i.e., $\mathcal{N}$ has Lebesgue measure zero.

Also, $P$ is absolutely continuous with respect to Lebesgue measure Leb. Thus, $P(\mathcal{N}) = 0$, i.e.,

$$\mathbb{P}((x_0, \cdots, x_n) | F(x_0, \cdots, x_n) = 0) = 0.$$

There are $\binom{N}{n+1}$ such set of $n + 1$ points, and the finite union of measure zero set is still measure zero. Therefore, the set $\{x_i\}_{i=1}^N$ is in general position with probability one. $\square$

### F.2 DATA-DEPENDENT POLYNOMIAL APPROXIMATION IN SOBOLEV SPACE

In this section, we prove the main theorem of our approximation result Theorem 3.

**Theorem 13** (adaptive Bramble-Hilbert). *Let $m > n/p$, $1 \leq p < \infty$. Given $N$ data points $\{(x_i, y_i)\}_{i=1}^N$ in general position, consider the convex hull of all the $x_i$, i.e., $\mathcal{I} = \text{conv}\{x_1, \cdots, x_N\} \subseteq \Omega$. For any $\phi \in W^{m,p}(\mathcal{I}^\circ)$, there exists a polynomial interpolation $\psi$ of order $p(N, n)$, s.t.,*

$$\|\phi - \psi\|_{r,p} \leq C \max\{h_{\max_{n+1}}, 1\}^m \min\{h_{\min}, 1\}^{-r} \|\phi\|_{m,p}$$

*where $h_{\min} = \min_i h_{i,\min}$ and $h_{\max_{n+1}} = \max_{\mathcal{I}_i} h_{i,\max}$ is the maximum $h_{ij}$ over all the $n-$simplex.*

The proof is an adaptation of Watkins (1979).

*Proof.* For any $n-$simplex $\mathcal{I}_i$, define $\Delta v_i = (v_{i1} - v_{i0}, v_{i2} - v_{i0}, \cdots, v_{in} - v_{i0}) \in \mathbb{R}^{n \times n}$, and $h_{ij} = \|v_{ij} - v_{i0}\|$. Also, $\mathcal{I} = \bigcup_i \mathcal{I}_i$.

For $x \in \mathcal{I}_i$, define an affine map $x = v_0 + \Delta v_i \xi$, where $\xi \in \mathcal{D} = \{\xi \in \mathbb{R}^n | \xi \in [0, 1]^n, \sum_{i=1}^n \xi_i \leq 1\}$. Then this map is a one-to-one mapping of $\mathcal{D}$ onto $\mathcal{I}_i$. Let the vertices of $\mathcal{D}$ be $\bar{v}_{ij}$ for $j = 0, \cdots, n$. Then for any function $\phi$ on $\mathcal{I}$, define $\bar{\phi}(\xi) = \phi(x) = \phi(v_0 + \Delta v_i \xi)$. Then

$$\|D^\alpha \phi(x)\|_{0,p,\mathcal{I}_i^\circ} = \prod_{j=1}^n h_{ij}^{-\alpha_i} |J_i|^{1/p} \|D^\alpha \bar{\phi}(\xi)\|_{0,p,\mathcal{D}^\circ}$$

where $|J_i|$ is the determinant of the Jacobian of the affine map. Then, we have

$$
\|\bar{\phi}(\xi)\|_{m,p,\mathcal{D}^\circ} = \left( \sum_{|\alpha| \leq m} \|D^\alpha \bar{\phi}\|_{0,p,\mathcal{D}^\circ}^p \right)^{1/p}
$$

$$
= \left( \sum_{|\alpha| \leq m} \left( \prod_{j=1}^n h_{ij}^{\alpha_j} |J_i|^{-1/p} \right)^p \|D^\alpha \phi(x)\|_{0,p,\mathcal{I}_i^\circ}^p \right)^{1/p}
$$

$$
\leq \max\{h_{i,\max}^m, 1\} |J_i|^{-n/p} \|\phi(x)\|_{m,p,\mathcal{I}_i^\circ} \tag{24}
$$

and similarly,

$$
\|\phi(x)\|_{m,p,\mathcal{I}_i^\circ} \leq \max\{h_{i,\min}^{-m}, 1\} |J_i|^{n/p} \|\bar{\phi}(\xi)\|_{m,p,\mathcal{D}^\circ} \tag{25}
$$

where $h_{i,\min} = \min_j h_{ij}$, and $h_{i,\max} = \max_j h_{ij}$.

For any function $\phi \in C(\mathcal{I})$, we can define a unique polynomial interpolant $\psi = B\phi$, where $\psi \in C(\mathcal{I})$ is a polynomial with degree $\leq p(N,n)$ for some $p(N,n) \geq m-1$ such that $p_{m-1} = Bp_{m-1}$ for any polynomial $p_{m-1}$ of order at most $m-1$, and the operator $B$ is linear (see for example Saniee (2008)). The two functions $\phi, \psi$ are equal at the vertices of the $n$-simplex, i.e., $\phi(v_{ij}) = \psi(v_{ij})$, for all $j = 0, \cdots, n$.

Let $\bar{\phi}(\xi) = \phi(x)$ and $\bar{\psi}(\xi) = \psi(x)$. We can define the linear operator $\bar{B}$, s.t.,

$$
(\phi - B\phi)(x) = (\bar{\phi} - \bar{B}\bar{\phi})(\xi).
$$

Then, we can find a set of barycentric coordinate functions $\bar{\phi}_0(\xi), \cdots, \bar{\phi}_n(\xi)$, which is the basis function of a natural coordinate system for $n$-simplex, s.t.

$$
\bar{\phi}_s(\bar{v}_{ij}) = \delta_{sj} \text{ and } \bar{\psi}(\xi) = \sum_{j=0}^n \bar{\phi}_j(\xi)\bar{\phi}(\bar{v}_{ij}).
$$

Then

$$
\|\bar{B}\bar{\phi}\|_{r,p,\mathcal{D}^\circ} = \|\bar{\psi}\|_{r,p} \leq \max_j \|\bar{\phi}(\bar{v}_{ij})\| \sum_{s=0}^n \|\bar{\phi}_s\|_{r,p,\mathcal{D}^\circ}
$$

$$
\leq \|\bar{\phi}\|_\infty \sum_{s=0}^n \|\bar{\phi}_s\|_{r,p,\mathcal{D}^\circ}
$$

$$
\leq C'\|\bar{\phi}\|_{m,p,\mathcal{D}^\circ} \sum_{s=0}^n \|\bar{\phi}_s\|_{r,p,\mathcal{D}^\circ} = C''\|\bar{\phi}\|_{m,p,\mathcal{D}^\circ}
$$

where the last inequality follows from the Sobolev embedding theorem, and $C', C'' > 0$. This shows that $\bar{B}$ is a bounded operator from $W^{m,p}(\mathcal{D}^\circ)$ to $W^{r,p}(\mathcal{D}^\circ)$, and so is $I - \bar{B}$, i.e.,

$$
\|(I - \bar{B})\bar{\phi}\|_{r,p,\mathcal{D}^\circ} \leq C'''\|\bar{\phi}\|_{m,p,\mathcal{D}^\circ} \tag{26}
$$

for some universal constant $C''' > 0$.

Then

$$
\|\phi - \psi\|_{r,p,\mathcal{I}_i^\circ} = \|\phi - B\phi\|_{r,p,\mathcal{I}_i^\circ} \leq \max\{h_{i,\min}^{-r}, 1\} |J_i|^{n/p} \|\bar{\phi} - \bar{B}\bar{\phi}\|_{r,p,\mathcal{D}^\circ}
$$

$$
= \max\{h_{i,\min}^{-r}, 1\} |J_i|^{n/p} \|(I - \bar{B})\bar{\phi}\|_{r,p,\mathcal{D}^\circ}
$$

$$
\leq C''' \max\{h_{i,\min}^{-r}, 1\} |J_i|^{n/p} \|\bar{\phi}\|_{m,p,\mathcal{D}^\circ}
$$

$$
\leq C''' \max\{h_{i,\max}^m, 1\} \max\{h_{i,\min}^{-r}, 1\} \|\phi\|_{m,p,\mathcal{I}_i^\circ}
$$

where the first inequality follows from (25), the second inequality follows from (26), the last inequality follows from (24).

Then we have for any $\phi \in W^{m,p}(\mathcal{I}^\circ)$,

$$
\|\phi - \psi\|_{r,p} \leq C \max\{h_{\max_{n+1}}^m, 1\} \max\{h_{\min}^{-r}, 1\} \|\phi\|_{m,p}
$$

$$
= C \max\{h_{\max_{n+1}}, 1\}^m \min\{h_{\min}, 1\}^{-r} \|\phi\|_{m,p}
$$

where $h_{\min} = \min_i h_{i,\min}$ and $h_{\max_{n+1}} = \max_{\mathcal{I}_i} h_{i,\max}$ is the maximum $h_{ij}$ over all the $n$-simplex, and $C > 0$ is some universal constant. $\qquad\square$

## G  ADDITIONAL LEMMAS

**Lemma 15.** *Let $\Omega$ be a bounded open set. Then the Sobolev space $W^{m,p}(\Omega)$ is dense in $L^p(\Omega)$.*

*Proof.* By definition, $W^{m,p}(\Omega) \subseteq L^p(\Omega)$. Also, we have $C^\infty(\Omega)$ is dense in $L^p(\Omega)$, and in $W^{m,p}(\Omega)$. Then $W^{m,p}(\Omega)$ is dense in $L^p(\Omega)$. □

**Lemma 16.** *Under Assumption 2, the neural network $f(x) \in W^{r,p}(\Omega)$ for any $1 \le r, p < \infty$.*

*Proof.* By Assumption 2 and Proposition 1, the neural network function $f(x)$ is analytic on some open neighbourhood $\Omega_1$ of $\bar{\Omega}$. Then $f(x) \in C^\infty(\Omega_1)$, and therefore

$$\sup_{x \in \bar{\Omega}} \|D^\alpha f(x)\|_{\ell_p} < \infty$$

for any non-negative $\alpha = (\alpha_1, \cdots, \alpha_d)$, and any $1 \le p < \infty$. By $|\Omega| < \infty$, we have the $L^p(\Omega)$ norm of any derivative

$$\|D^\alpha f(x)\|_p^p \le |\Omega| \sup_{x \in \bar{\Omega}} \|D^\alpha f(x)\|_{\ell_p}^p < \infty.$$

Therefore, $f(x) \in W^{r,p}(\Omega)$ for any $1 \le r, p < \infty$. □

## H  EXTENDED EXPERIMENTS

In our experiments, we first introduce the data uniformity strategy—iteratively selecting data points that maximize the distances among all previously chosen points, and present visualizations comparing the selected uniform data, randomly selected data of the same size, and the full dataset. The purpose of comparing the selected uniform data with randomly selected data of the same size is to isolate the effect of the minimum distance $h_{\min}$ while keeping the dataset size constant. We then conduct supervised fine-tuning with two complementary objectives: $\ell^2$ loss optimized via SGD for theoretical validation, and cross-entropy loss optimized via Adam for practical evaluation. This setting enables us to systematically assess both performance and training efficiency across varying source datasets, optimization strategies, dataset scales, and model sizes, thereby demonstrating the robustness of our method. Specifically, we deploy our approach into state-of-the-art data distillation and data selection baselines, including TeaMs-RL (Gu et al., 2024), WizardLM (Xu et al., 2024), Zcore (Griffin et al., 2024), and LESS (Xia et al., 2024). Comparative experiments show that our method not only improves training efficiency but also achieves superior or comparable performance relative to these strong baselines. The experiments validate our theoretical results by showing that training with the selected uniform data, which implies a larger $h_{\min}$, accelerates convergence (see Corollary 3) while maintaining comparable performance to using the full dataset (see Theorem 3 and the discussions below).

### H.1  DATA SELECTION AND VISUALIZATION

**Data Selection via Distance Maximization.**  To construct compact yet representative instruction-tuning datasets, we apply a greedy uniformity selection strategy based on data distance. Specifically, we represent each instruction–input–output triple as an average Word2Vec embedding trained on the full corpus. Starting from a randomly selected seed, we iteratively add the data point that maximizes the minimum distance to all previously selected points, ensuring maximal coverage of the data space. We apply this strategy to four baselines: TeaMs-RL (Gu et al., 2024), WizardLM (Xu et al., 2024), Zcore (Griffin et al., 2024), and LESS (Xia et al., 2024). The TeaMs-RL and WizardLM datasets are both distilled from ChatGPT-3.5/4, using different strategies to generate diverse instructional data. From the 9k TeaMs-RL dataset, we select a 4.5k maximized distance subset and compare it to a 4.5k random sample, which is composed of half randomly sampled data points and half selected to minimize pairwise distances from the 9k full dataset[3]; from the 20k WizardLM dataset, we select a 10k maximized distance subset comparing it against a 10k random sample, and the

---

[3]This choice of random data is made because the dataset size is relatively small, and we aim to highlight the effect of different minimum distances between data points $h_{\min}$.

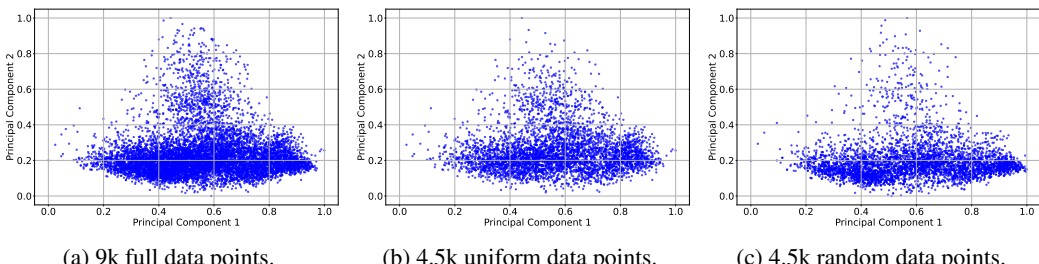

Figure 7: Visualization of data point distributions for datasets selected using different methods from TeaMs-RL (Gu et al., 2024).

random subset consists of the first 10k samples from the original 20k dataset, which is already randomly shuffled. This approach allows us to fairly evaluate the benefits of uniformity-oriented sampling across different dataset scales. Beyond instruction-tuning datasets, we also apply the same strategy to the California Housing dataset (Pace & Barry, 1997) and the LESS dataset (Xia et al., 2024), thereby enabling direct benchmarking against baselines such as Zcore (Griffin et al., 2024) and LESS (Xia et al., 2024).

**Data Visualization via PCA Projection.** To qualitatively assess the distributional coverage of different data selection methods, we project high-dimensional sentence embeddings into two dimensions using PCA (Maćkiewicz & Ratajczak, 1993). For consistency, PCA is fit on the full dataset (either TeaMs-RL 9k or WizardLM 20k) and applied to both the full set and its corresponding uniform and random subsets (the uniform data is selected based on maximizing pairwise distance). The resulting 2D coordinates are normalized using MinMax scaling to [0,1] and plotted for visualization. As shown in Figures 3 and 7, the uniform subsets from TeaMs-RL (Gu et al., 2024) and WizardLM (Xu et al., 2024) (4.5k for TeaMs-RL, 10k for WizardLM) exhibit broader and more dispersed coverage across the embedding space compared to the random subsets and the full dataset, which cluster more tightly and redundantly. These patterns highlight the ability of uniformity-based sampling to retain semantic coverage with reduced sample sizes, supporting its utility for efficient training. A similar strategy is employed in Zcore (Griffin et al., 2024) and LESS (Xia et al., 2024).

## H.2 $\ell^2$-SGD TRAINING EXPERIMENTS

We present training dynamics under $\ell^2$ loss with SGD to validate our theoretical insights. While $\ell^2$-(S)GD provides a useful controlled setting for analysis, it is generally less efficient in practice compared to cross-entropy loss optimized with Adam, which is a widely used approach for training LLMs (Gunel et al., 2021; Mao et al., 2023).

**Training Comparison with Uniform Subsets.** Figure 8 shows the training loss curves of LLaMA-1-7B (Touvron et al., 2023)[4] using $\ell^2$ loss with SGD (Bottou, 2010; Cortes et al., 2012) on the TeaMs-RL dataset under three data configurations: the full dataset (9k), a 4.5k subset selected by maximizing pairwise distance, and a 4.5k randomly sampled subset. The figure includes both unsmoothed (left) and smoothed (right) loss curves, where unsmoothed curves reflect raw stepwise fluctuations in loss, while smoothed curves (e.g., via moving average) highlight overall training trends more clearly. In both plots, we zoom in the early and middle stage of the training for clearer comparison (see full training in Appendix I), and observe that the model trained on the uniform subset consistently converges faster than the one trained on the random subset across training steps. Moreover, despite using only half the number of samples, the uniform subset achieves convergence behavior comparable to or better than the full dataset. This validates our theory (see Section 4) and suggests that uniformity-aware data selection can substantially enhance training efficiency and convergence, even under limited data budgets. More experiments are provided in Appendix I.

---

[4]https://huggingface.co/huggyllama/llama-7b

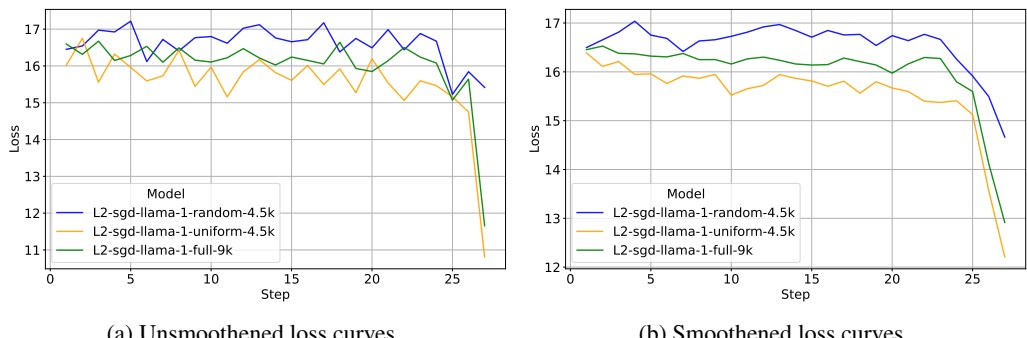

(a) Unsmoothened loss curves.      (b) Smoothened loss curves.

Figure 8: $\ell^2$ SGD Training Experiments: Training loss comparison of TeaMs-RL data point distributions for different dataset sizes: 4.5k uniform (maximized distance) dataset, 4.5k random dataset, 9k full dataset.

### H.3 CROSS ENTROPY LOSS ADAM TRAINING EXPERIMENTS

Beyond the l2-SGD, we conduct the experiments suing cross-entropy loss with Adam, the standard optimization setup for LLMs (Gunel et al., 2021; Mao et al., 2023), we observe consistent improvements. As shown in Figure 6 presents training dynamics of LLaMA-1-7B using cross-entropy loss with AdamW on the LESS dataset (Xia et al., 2024), comparing uniform subsets (ours), random subsets, and the full dataset. Panel (a) shows that the 7k uniform subset reaches the loss threshold more than 2× faster than the 14k full dataset and substantially faster than the 7k random subset, highlighting improved training efficiency. Panel (b) reports MMLU accuracy, where the uniform subset achieves performance comparable to or exceeding both random and full datasets despite using fewer samples. Panel (c) illustrates training loss trajectories across seeds, demonstrating that uniform subsets yield smoother and more stable convergence relative to baselines. Together, these results indicate that uniformity-aware data selection not only reduces training time but also improves robustness and generalization, validating its effectiveness for efficient large-scale model training.

**Cross-Entropy Loss and Adam Optimization with maximized-distance Subsets.** Figures 9 (a) and 9 (b) compare training loss curves under cross-entropy loss (Mao et al., 2023) using the Adam optimizer (Kingma & Ba, 2014) for three TeaMs-RL datasets: the full 9k dataset, a 4.5k maximized-distance subset selected via maximized distance, and a 4.5k random subset. Figure 9 (a) presents raw training loss, while Figure 9 (b) shows smoothed loss for clarity. In both cases, the model trained on the maximized-distance based subset consistently converges faster and achieves significantly lower loss than the one trained on the random subset, despite both using the same number of samples. Notably, the maximized-distance 4.5k subset also outperforms the full 9k dataset in both convergence speed and final loss, highlighting the efficiency gains from data uniformity and well-distributed data. These results reaffirm that maximized-distance data selection not only improves optimization but also reduces training redundancy under standard cross-entropy objectives with modern optimizers. Notably, models trained on uniform data exhibit lower initial loss compared to those trained on random or full datasets. This may be attributed to the higher diversity and broader coverage of the uniform subset, which is more aligned with the base model's existing knowledge, likely due to its pretraining on large-scale corpora exceeding 1T tokens (Touvron et al., 2023).

**Scalability of Uniform Data Benefits with Larger Datasets.** Figures 10 (a) and 10 (b) show the training dynamics of models trained on WizardLM data using cross-entropy loss with the Adam optimizer across three dataset settings: the full 20k dataset, a 10k maximized-distance subset selected by maximizing distance, and a 10k random subset. In both plots, the maximized-distance subset consistently outperforms the random counterpart throughout the training process, achieving lower and more stable loss values. Notably, the maximized-distance 10k subset even surpasses the full 20k dataset regarding convergence, suggesting that data uniformity can compensate for and even outperform brute-force data scaling. This pattern, observed across both TeaMs-RL and WizardLM data, highlights the robustness and scalability of uniform-based sampling strategies, offering

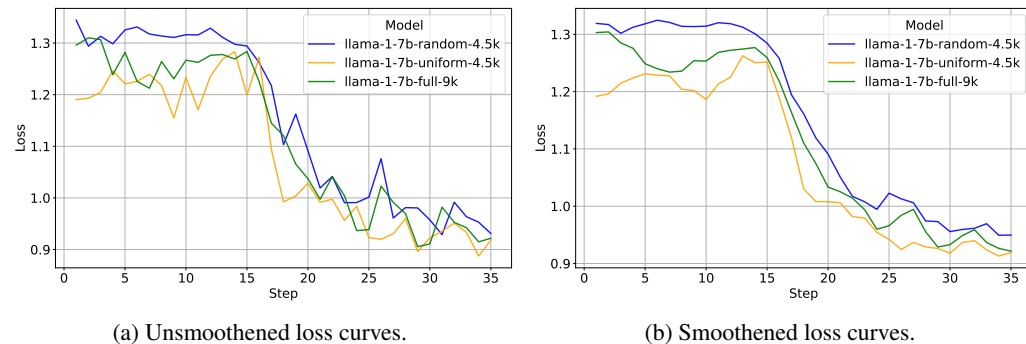

(a) Unsmoothened loss curves.  (b) Smoothened loss curves.

Figure 9: Training loss comparison of TeaMs-RL data point distributions for different dataset sizes using cross entropy loss and Adam: 4.5k uniform (maximized distance) dataset, 4.5k random dataset, 9k full dataset.

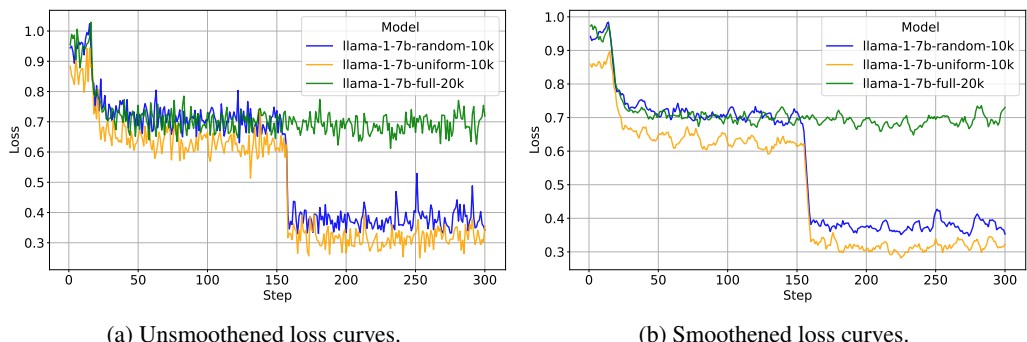

(a) Unsmoothened loss curves.  (b) Smoothened loss curves.

Figure 10: Training loss comparison of WizardLM data point distributions for different dataset sizes using cross entropy and adam: 10k uniform (maximized distance) dataset, 10k random dataset, 20k full dataset.

a compelling approach for reducing computational costs without sacrificing learning quality. The complete training experiments across all steps are provided in Appendix I.4.

**Training on large-scale models.** Figures 11 through 12 present training loss comparisons for the LLaMA-1 13B model (Touvron et al., 2023)[5] using full datasets, random subsets, and uniformity-selected subsets across TeaMs-RL (4.5k/9k) and WizardLM (10k/20k) data. In all settings, the models trained on uniformity-maximized subsets exhibit consistently lower or comparable loss compared to those trained on full datasets, and significantly outperform the randomly sampled subsets. This trend holds both with and without smoothing applied to the loss curves. Particularly in Figures 12 (a) and 12 (b), the 10k uniform subset achieves better final loss than even the full 20k dataset, reinforcing the efficiency of data uniformity in training large-scale models. These results demonstrate that uniformity-aware sampling remains effective even at larger model scales, offering substantial gains in training efficiency without compromising performance. The complete training experiments across all steps are provided in Appendix I.4.

### H.4  PERFORMANCE AND TRAINING TIME COMPARISON

**Evaluation Performance with Diverse Subsets.** Figures 13 and 14 present evaluation results of LLaMA-1-13B on the ARC Challenge (Clark et al., 2018) and TruthfulQA MC benchmarks (Lin et al., 2021), using datasets from TeaMs-RL and WizardLM, respectively. In both settings, models trained on data selected via maximum pairwise distance (orange bars) consistently outperform those trained on random subsets (green) and the original baselines (blue). Notably, the 4.5k and 10k uniform subsets achieve performance comparable to or slightly better than the full datasets

---

[5]https://huggingface.co/huggyllama/llama-13b

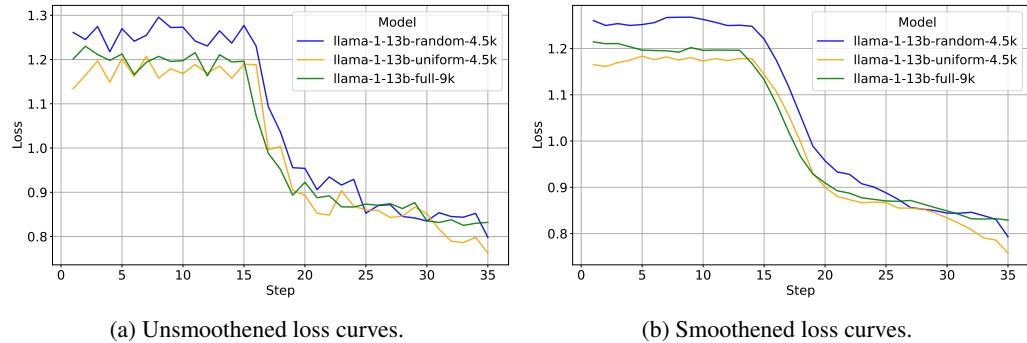

(a) Unsmoothened loss curves.  (b) Smoothened loss curves.

Figure 11: Training loss comparison of TeaMs-RL data point distributions for different dataset sizes using cross entropy loss and Adam on llama-1-13b models: 4.5k uniform (maximized distance) dataset, 4.5k random dataset, 9k full dataset.

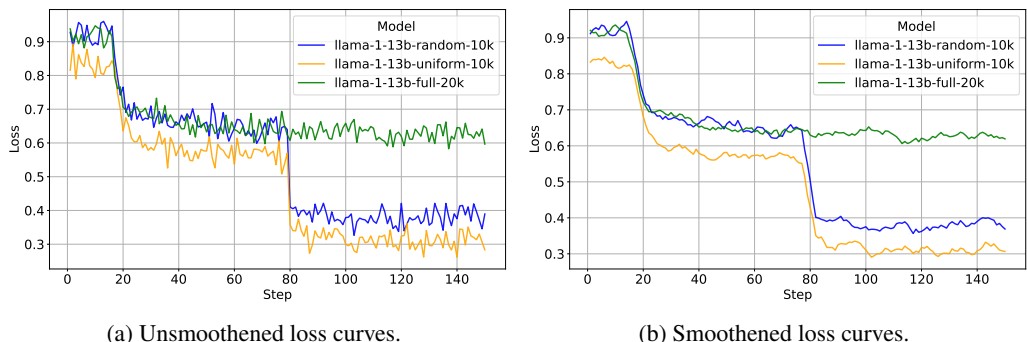

(a) Unsmoothened loss curves.  (b) Smoothened loss curves.

Figure 12: Training loss comparison of WizardLM data point distributions for different dataset sizes using cross entropy and adam on llama-1-13b models: 10k uniform (maximized distance) dataset, 10k random dataset, 20k full dataset.

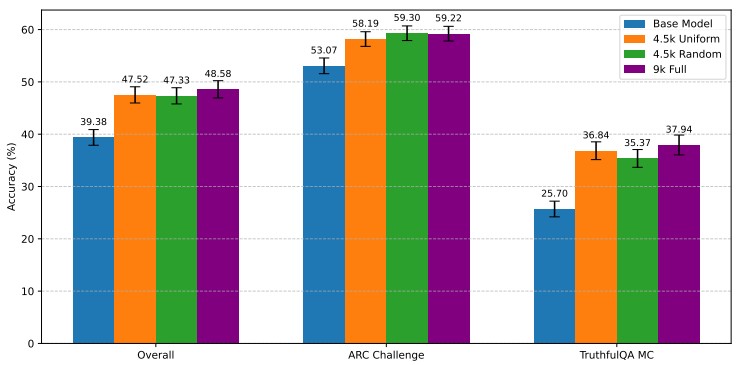

Figure 13: ARC and MC performance on Llama-1-13b using TeaMs-RL diverse datasets.

(red) while using significantly fewer samples. This is particularly evident on the ARC Challenge task (e.g., Figure 15), where the uniform subsets closely match or exceed the accuracy of the full datasets. These results further validate the effectiveness of uniformity-based data selection in improving model generalization and sample efficiency for LLM training. Note that models trained on different datasets may reach different loss levels at the same training steps; therefore, we select checkpoints with approximately equal loss values to ensure a fair performance comparison, e.g., 0.8733 for the TeaMs-RL dataset and 0.7058 for the WizardLM dataset.

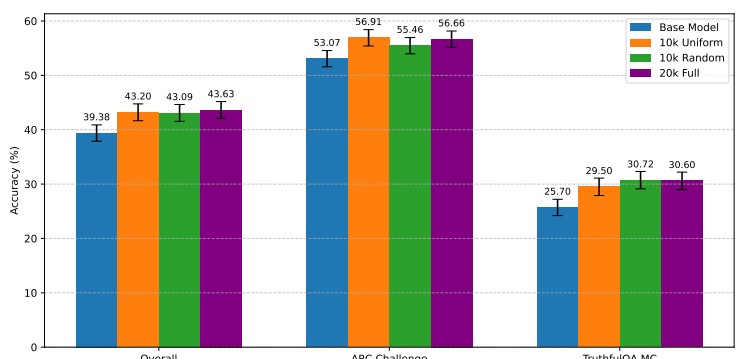

Figure 14: ARC and MC performance on Llama-1-13b using WizardLM diverse datasets.

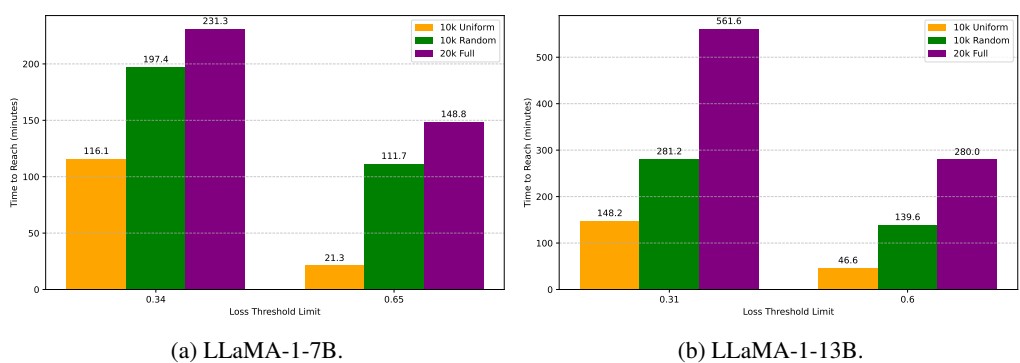

(a) LLaMA-1-7B.

(b) LLaMA-1-13B.

Figure 15: **Training time to reach loss thresholds using a 3-step moving average, measured on a single A100 GPU.** For both (a) LLaMA-1-7B (0.34, 0.65) and (b) LLaMA-1-13B (0.31, 0.6), we compare the convergence speed of Uniform, Random, and Full data selection strategies. Uniform sampling consistently achieves faster convergence, indicating more efficient training.

**Time to Threshold: Faster Convergence with Uniform Data Selection.** As shown in Figures 15 (a) and (b), we compare the convergence efficiency of three data selection strategies—Uniform, Random, and Full, under different loss thresholds using LLaMA-1-7B and LLaMA-1-13B. All times are measured based on wall-clock time using a single A100 GPU. For LLaMA-1-7B, Uniform reaches the 0.34 and 0.65 loss thresholds in 116.1 and 21.3 minutes, respectively, outperforming Random (197.4, 111.7) and Full (231.3, 148.8). Similarly, LLaMA-1-13B trained on the 10k Uniform subset reaches the 0.31 and 0.6 thresholds in 148.2 and 46.6 minutes, significantly faster than Random (281.2, 139.6) and 20k Full (561.6, 280.0). These results highlight that Uniform selection not only reduces the dataset size but also accelerates training, indicating a more efficient optimization path and improved data quality.

# I    SUPPLEMENTARY EXPERIMENTS

## I.1    ADDITIONAL EXPERIMENTS USING $\ell^2$-SGD

## I.2    EFFECTIVENESS OF MAXIMIZED-DISTANCE BASED SAMPLING ON SMALLER SCALE DATASETS.

Figure 16 illustrates the $\ell^2$ SGD training loss curves for LLaMA-1-7B across different data selection strategies on the TeaMs-RL dataset. We compare a 4.5k **uniform subset**, a 4.5k **random subset**, and the **full 9k dataset**. Subfigure (a) shows the raw loss trajectories, while (b) applies smoothing to better visualize convergence patterns. The 4.5k uniform subset achieves convergence comparable to or faster than the full dataset and consistently outperforms the random subset, particularly in the early stages of training. However, since this dataset size is relatively small, the advantage of data

uniformity is not very pronounced in the final stage of training. When the dataset size increases, uniform data selection becomes more effective in the later phase (see Figure 18).

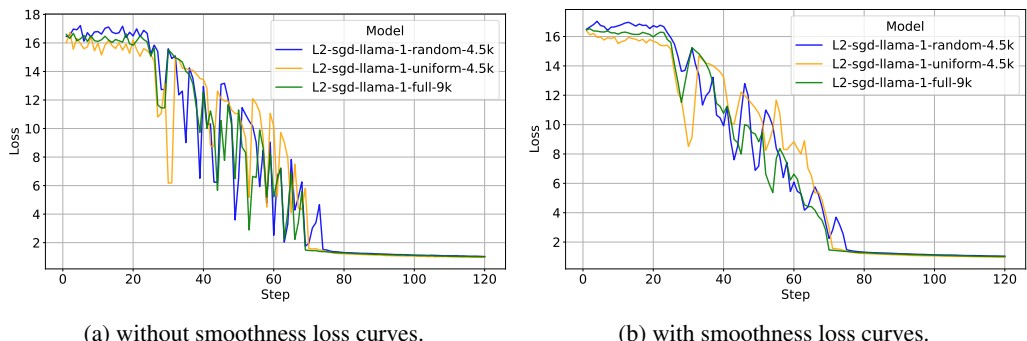

(a) without smoothness loss curves.

(b) with smoothness loss curves.

Figure 16: $\ell^2$ SGD Training Experiments—All Curves: Training loss comparison of TeaMs-RL data point distributions for different dataset sizes: 4.5k uniform (maximized distance) dataset, 4.5k random dataset, 9k full dataset.

## I.3    EFFECTIVENESS OF MAXIMIZED-DISTANCE BASED SAMPLING ON LARGER SCALE DATASETS.

Figures 17 and 18 show the $\ell^2$-SGD training loss curves of llama-1-7b models trained on the WizardLM dataset under three different data selection settings: the full 20k dataset, a 10k maximized-distance subset selected via maximization distance, and a 10k random subset. In both the raw (left) and smoothed (right) plots, the maximized-distance 10k subset outperforms the random 10k subset across training steps, achieving faster convergence and more stable convergence. Notably, despite being half the size, the maximized-distance subset often matches or even surpasses the performance of the full dataset in early and end training regarding the convergence. This result indicates the stability and efficiency of data uniformity sampling strategies, particularly in large-scale training tasks. It highlights the finding that uniform data offers stronger learning signals than merely increasing sample count without regard to data uniformity.

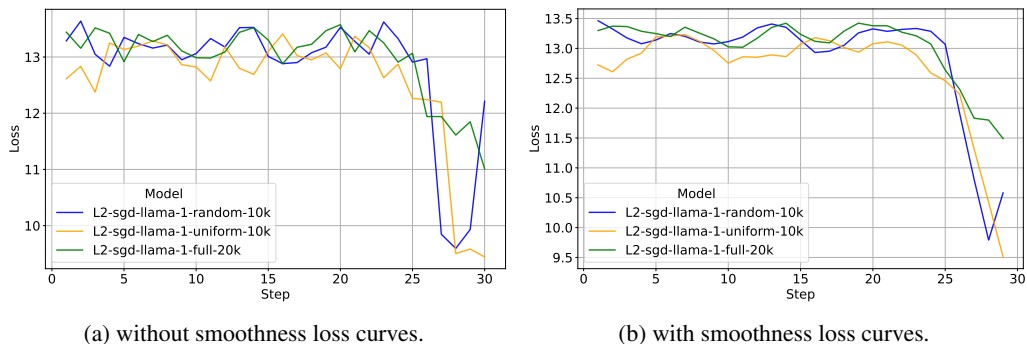

(a) without smoothness loss curves.

(b) with smoothness loss curves.

Figure 17: $\ell^2$ SGD Training Experiments: Training loss comparison of WizardLM data point distributions for different dataset sizes: 10k uniform (maximized distance) dataset, 10k random dataset, 20k full dataset.

## I.4    FULL STEP TRAINING: 7B AND 13B MODELS

As shown in Figure 19 and Figure 20, models generally converge with sufficient training time, but our method achieves faster convergence.

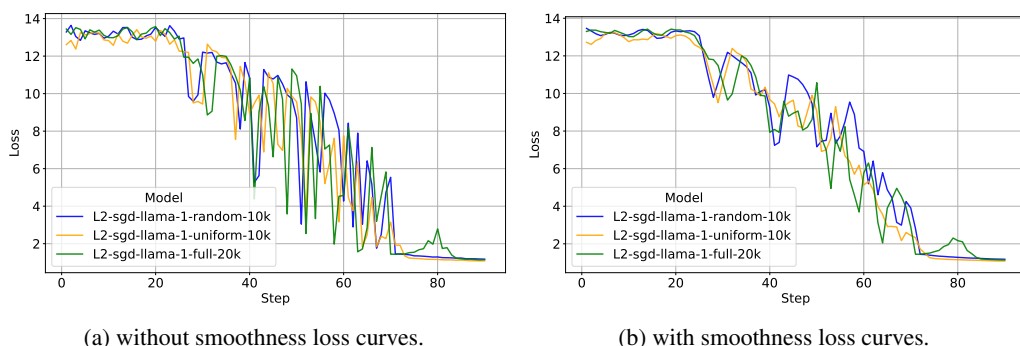

(a) without smoothness loss curves.

(b) with smoothness loss curves.

Figure 18: $\ell^2$ SGD Training Experiments—All Curves: Training loss comparison of WizardLM data point distributions for different dataset sizes: 10k uniform (maximized distance) dataset, 10k random dataset, 20k full dataset.

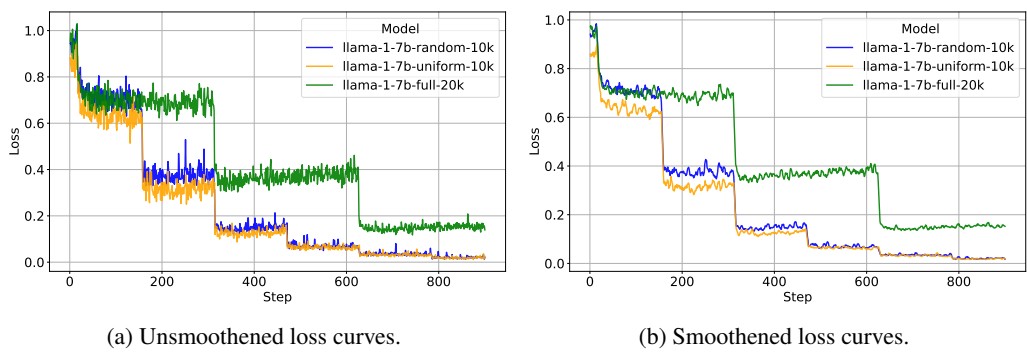

(a) Unsmoothened loss curves.

(b) Smoothened loss curves.

Figure 19: Full Steps: Training loss comparison of WizardLM data point distributions for different dataset sizes using cross entropy and adam: 10k uniform (maximized distance) dataset, 10k random dataset, 20k full dataset.

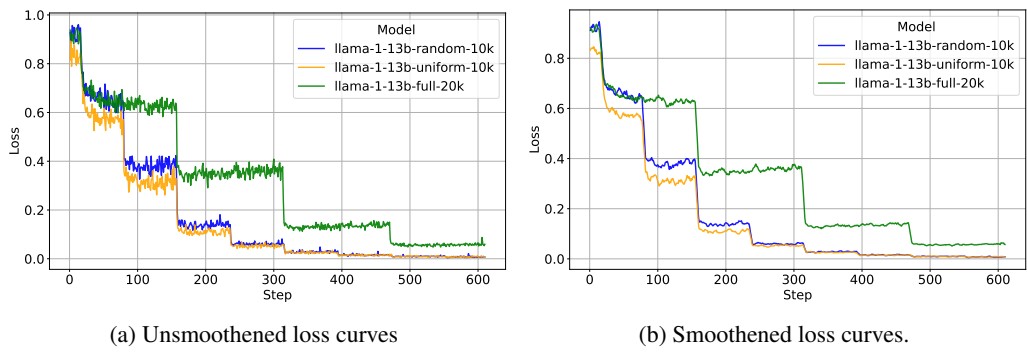

(a) Unsmoothened loss curves

(b) Smoothened loss curves.

Figure 20: Full Steps: Training loss comparison of WizardLM data point distributions for different dataset sizes using cross entropy and adam on llama-1-13b models: 10k uniform (maximized distance) dataset, 10k random dataset, 20k full dataset.

## I.5 ABLATION EXPERIMENT

As shown in Figure 21, it shows that data selection has a clear impact on training efficiency. Across both raw and smoothed curves, the minimize-10k datasets consistently reach lower loss than other datasets, and uniform-10k datasets achieve faster than other datasets, such as random-10k and 20k-full datasets, demonstrating that more principled selection can achieve stronger optimization. This indicates that additional low-quality or redundant data can hinder training efficiency. Overall, these

results highlight that careful data selection can yield both faster convergence and better final model quality, even with substantially less data.

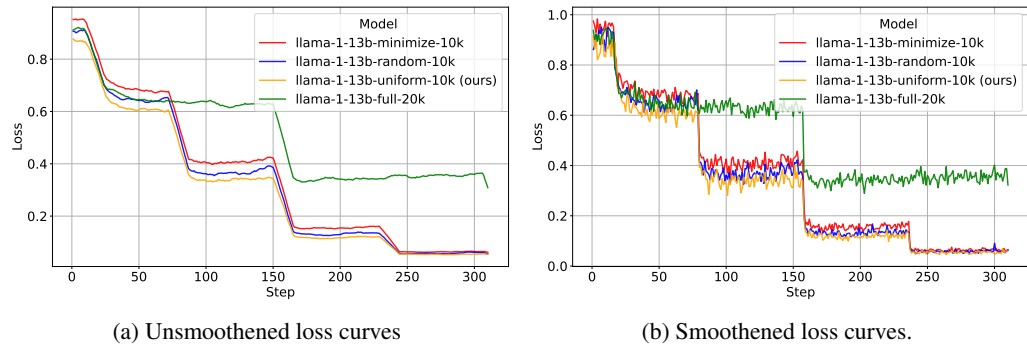

(a) Unsmoothened loss curves          (b) Smoothened loss curves.

Figure 21: Training-efficiency ablation comparing data-selection strategies for Llama-1-13B. Left: raw (unsmoothed) loss curves. Right: smoothed loss curves. We compare minimize-10k, random-10k, uniform-10k (Ours), and full-20k training subsets.

As shown in Figure 22, this ablation examines whether the choice of embedding model affects the effectiveness of our uniform data-selection strategy. The results show that word2vec-based and sentence-BERT–based (s-bert) uniform sampling yield nearly identical training dynamics: both variants reduce loss at a similar rate, maintain stability throughout optimization, and converge to almost the same final value. The close overlap of the curves indicates that the uniformity principle itself, not the specific embedding space, is the primary factor driving performance gains. This suggests that our method is robust, lightweight, and flexible: simple embedding models like word2vec are already sufficient to realize the benefits of uniform selection, while stronger encoders like s-bert do not materially change the outcome.

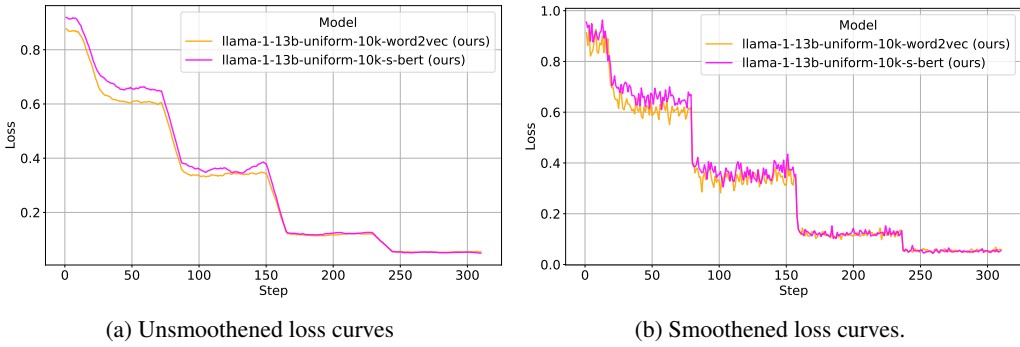

(a) Unsmoothened loss curves          (b) Smoothened loss curves.

Figure 22: Ablation on embedding methods used in our uniform data-selection framework. Left: raw loss curves. Right: smoothed loss curves. We compare uniform-10k-word2vec (ours) and uniform-10k-s-bert (ours) for Llama-1-13B.

As shown in Figure 23, to ensure that the full-dataset and random-subset baselines train without exhibiting the gradient instabilities occasionally observed in default SGD settings (As shown in Figure 4), we fine-tuned their optimization hyperparameters, for example, by substantially reducing the learning rate (e.g., 0.01). While this stabilizes training, it also slows convergence and leads to noticeably less efficient learning. The results clearly show that our 8k Uniform (Ours) subset yields the strongest overall performance among all 8k-sized selections. Uniform sampling achieves faster training-loss reduction than both Random and Z-core, and it even surpasses the 16k Full baseline during the early and mid stages of optimization. On the validation set, the advantage of our method is more substantial: Uniform converges to the lowest MAE, demonstrating that it selects a more in-

formative and better-balanced subset that generalizes more effectively. In contrast, Random suffers from slow convergence, and Z-core plateaus early, while Full is consistently weaker than Uniform despite using twice the data. These results highlight that data quality and coverage are more important than dataset size, and that our uniform selection algorithm consistently provides the most efficient and generalizable training signal.

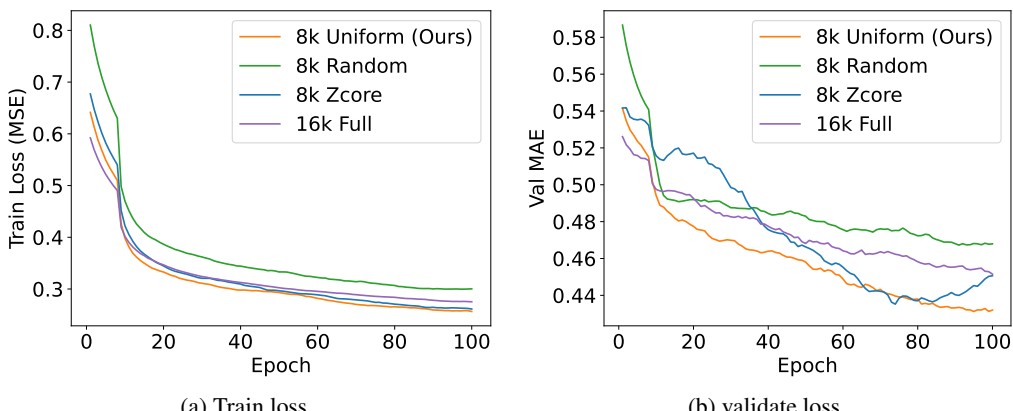

(a) Train loss.           (b) validate loss.

Figure 23: Training-efficiency ablation on the California Housing dataset comparing our uniform selection method with random, z-core, and full-data baselines (Smoothened loss). Left: training MSE loss. Right: validation MAE loss.

# J    EXPERIMENT SETTINGS

We summarize the key configurations [6] used in our experiments to ensure reproducibility and clarity. Table 1 details the training hyperparameters. Based on these and building on Llama-X[7], we train LLaMA-1-7B (one A100 GPU) and 13B (two A100 GPUs) models on datasets of varying sizes: WizardLM[8] (10k uniform datasets, 10k random datasets, and 20k full datasets) and TeaMs-RL[9] (4.5k uniform datasets, 4.5k random datasets, and 9k full datasets). The uniform subsets from TeaMs-RL and WizardLM are constructed using Word2Vec embeddings and a greedy selection strategy that maximizes uniformity. Specifically, one data point is randomly selected to initialize the set, and subsequent points are chosen to maximize the minimum cosine distance to all previously selected points. For TeaMs-RL, the random subset is composed of half randomly sampled data points and half selected to minimize pairwise distances from the 9k full dataset. In the case of WizardLM, the random subset consists of the first 10k samples from the original 20k dataset.

Table 2 shows the DeepSpeed configuration using cross-entropy loss with the Adam optimizer. Table 3 presents the DeepSpeed configuration used for $\ell^2$-regularized SGD training with Zero Stage 3 optimization and CPU offloading. Finally, we leverage lm-evaluation-harness (Gao et al., 2024) for our evaluation. Table 4 outlines the evaluation setup, including tasks, few-shot settings, and batch size settings used in the LM Evaluation Harness. These configurations collectively support our experiments across diverse datasets and model scales.

Table 1: Key experimental settings used in our study, including training with $\ell^2$ loss and cross-entropy loss using SGD and Adam optimizers.

| Parameters | value | Parameters | value |
|---|---|---|---|
| GPUs | [1, 2] | model max length | 512 |
| per device train batch size | 64 | per device eval batch size | 1 |
| gradient accumulation steps | 1 | evaluation strategy | no |
| learning rate | 2e-5 | warmup steps | 2 |
| logging steps | 1 | lr scheduler type | constant |
| gradient checkpointing | True | fp16 | True |

---

[6] https://anonymous.4open.science/r/data-uniformity-1A5C

[7] https://github.com/AetherCortex/Llama-X

[8] https://huggingface.co/datasets/WizardLMTeam/WizardLM_evol_instruct_V2_196k

[9] https://github.com/SafeRL-Lab/TeaMs-RL

Table 2: Key DeepSpeed configuration parameters used in our experiments (Zero Stage 3 with CPU offloading, FP16, AdamW optimizer).

| Parameters | Value | Parameters | Value |
|---|---|---|---|
| zero optimization stage | 3 | overlap communication | True |
| offload optimizer device | cpu | pin memory (optimizer) | True |
| offload param device | cpu | pin memory (param) | True |
| contiguous gradients | True | sub group size | 0 |
| reduce bucket size | auto | prefetch bucket size | auto |
| param persistence threshold | auto | max live parameters | 0 |
| max reuse distance | 0 | gather 16bit weights on save | True |
| fp16 enabled | True | auto cast | False |
| loss scale | 0 | initial scale power | 32 |
| loss scale window | 1000 | hysteresis | 2 |
| optimizer type | AdamW | learning rate | 2e-5 |
| beta1 | 0.9 | beta2 | 0.999 |
| epsilon | 1e-8 | weight decay | 0 |
| train batch size | auto | micro batch size per GPU | auto |
| gradient accumulation steps | auto | wall clock breakdown | False |

Table 3: Key DeepSpeed configuration parameters used in the experiments (Zero Stage 3 with CPU offloading, FP16, SGD optimizer).

| Parameters | Value | Parameters | Value |
|---|---|---|---|
| zero optimization stage | 3 | overlap communication | True |
| offload optimizer device | cpu | pin memory (optimizer) | True |
| offload param device | cpu | pin memory (param) | True |
| contiguous gradients | True | sub group size | 0 |
| reduce bucket size | auto | prefetch bucket size | auto |
| param persistence threshold | auto | max live parameters | 0 |
| max reuse distance | 0 | gather 16bit weights on save | True |
| allow untested optimizer | True | min loss scale | 1 |
| fp16 enabled | True | auto cast | False |
| loss scale | 0 | initial scale power | 32 |
| loss scale window | 1000 | hysteresis | 2 |
| optimizer type | SGD | learning rate | 2e-5 |
| momentum | 0.0 | weight decay | 0 |
| train batch size | auto | micro batch size per GPU | auto |
| gradient accumulation steps | auto | wall clock breakdown | False |

Table 4: Evaluation settings for baseline models using the LM Evaluation Harness.

| Parameter | Value | Parameter | Value |
|---|---|---|---|
| Model type | hf-causal-experimental | Batch size | 2 |
| Few-shot for `arc_challenge` | 25 | Few-shot for `truthfulqa_mc` | 0 |

