# OpenReview forum: "Data Uniformity Improves Training Efficiency and More, with a Convergence Framework Beyond the NTK Regime"
_ICLR.cc/2026/Conference — Submitted to ICLR 2026_

### Official Review · Reviewer_9Va8 · 2025-10-31

**Soundness:** 3
**Presentation:** 3
**Contribution:** 2
**Rating:** 4
**Confidence:** 3

**Summary:**

The paper analyzes how data uniformity—measured by the minimum pairwise distance among samples—affects neural network training efficiency and approximation ability. It develops a convergence framework beyond the NTK regime and proposes a simple greedy data selection method that maximizes pairwise distances. Experiments on several datasets and LLaMA models show faster convergence and comparable or better performance using fewer tokens.

**Strengths:**

* The paper establishes a clear link between geometric data uniformity and training convergence beyond the NTK regime.
* This work demonstrates that more uniform subset selection can reduce training cost while maintaining or improving performance.

**Weaknesses:**

* The performance gains of the uniform over random selection are relatively minor (Figure 1), raising doubts about the practical significance and scalability.
* The evaluation scope is limited to only a few benchmarks and llama-1 models; broader and more diverse downstream tasks and base models would be necessary to substantiate the generality of the approach.
* The proposed data-uniformity procedure requires computing or approximating all pairwise distances among samples---an $O(N^2)$ operation---but the paper does not analyze its computational overhead.
* In Figure 4, the training and validation losses of the 16K full dataset display abnormal behavior.

**Questions:**

pls see weaknesses.

---

> ### Author Response · Authors · 2025-11-21
> **Response to Reviewer 9Va8 (1/3)**
>
> We would like to thank the reviewer for the useful comments, and for acknowledging the theoretical and empirical connection between data uniformity and improved training processes. Before addressing the specific questions, we would like to first clarify our main theoretical contributions and how they relates to practice, which hopefully better resolve the reviewer's concerns.
>
> 1. We theoretically prove that the minimum pairwise distance $h_{\min}$ is sufficient to characterize data uniformity in Theorem 1, and its effectiveness is further validated in our experiments. More precisely, in Theorem 1, we show that with high probability $1-2\delta$,
>
>  $\left(\frac{2\delta}{\pi_{\max} N(N-1)V_d}\right)^{1/d}\le h_{\min}\le C\left(\frac{-\log\delta}{\bar{\pi}_{\max}(N-1)V_d}\right)^{1/d},$
>
>  where $N$ is the sample size. This result means that if we view the input dataset as samples from some distribution $\pi(x)dx$ that is absolutely continuous with respect to Lebesgue measure on its support (Assumption 1), then more uniform distributions----i.e., "flattened" $\pi(x)$ with smaller $\pi_{\max}$ and $\bar\pi_{\max}$----guarantee larger upper and lower bounds of $h_{\min}$, compared to less uniform ones----i.e., "peaked" $\pi(x)$ with larger $\pi_{\max}$ and $\bar\pi_{\max}$----that yield smaller upper and lower bounds of $h_{\min}$ (see Line 225-231). In short, more (less) uniformity corresponds to larger (smaller) $h_{\min}$. At a high level, this result connects the macroscopic characterization (distribution/density level, e.g., $\pi(x)$) with the microscopic characterization (particle/data level, e.g., $h_{\min}$). As a slightly non-rigorous interpretation, across a lot of datasets/tasks/training processes, $h_{\min}$ can represent data uniformity with high probability.
>
>  2. We theoretically prove that more uniform data (i.e., larger $h_{\min}$) can accelerate convergence in Corollary 3
>  $$\mu_{\rm low,s,X}\le L_{k,i} h_{\min},$$
> where $L_{k,i}>0$ is a constant, and larger $h_{\min}$ leads to larger $\mu_{\rm low,s,X}$, which determines faster convergence in Theorem 2 through the following bound
> $$\mathcal{L}(\theta^k)\le \prod_{s=0}^{k-1}\left(1- \eta\left(1-\frac{\eta\, \mathsf{L}}{2}\right)\frac{\mu_{\mathrm{low},s,X}}{N}\right)\mathcal{L}(\theta^{0}).$$
> We then validate this result in the experiments, including LLM tasks.
>
> 3. Moreover, we also theoretically prove the approximation error in Theorem 2, bounding the difference between neural network $f$ and the ground truth $g$
>
> $\|f-g\|_{p,\mathcal I^o}
>         \le  C_1  h ^m _{ \max d+1} h _ { \min } ^{ -r }  \| \phi \| _ { m,p, \mathcal I^o}.$
>
> Here, $C_1$ and $\| \phi \| _ { m,p, \mathcal I^o}$ can be viewed as fixed terms, and the bound is primarily determined by $h_{\min}$ and $h_{\max_{d+1}}$, where $h_{\max_{d+1}}$ is the maximum edge length of the $n-$simplex in the triangulation (i.e., a local maximum distance). Notably, this result implies that using fewer but more uniform data (i.e., increasing $h_{\min}$) can achieve similar approximation error as using more samples (i.e., decreasing $h_{\max_{d+1}}$) (see Line 362-370). This is also confirmed in the experiments: the performance of our uniform dataset (half the size of full dataset) is comparable to that of the full dataset (see Figure 1,5,6).
>
> 4. We develop a new proof framework for analyzing the convergence of a family of complicated neural networks:
>
> $u_0=x;$
>
> $u_{ l+1}=u_l+\bar{ \varphi }_l ( \theta_l ; u_l ), \forall l=0,\cdots,L-1;$
>
> $f ( \theta ; x ) = \bar{ \varphi }_L ( \theta_L ; u_L ).$
>
> where each residual block $\bar{\varphi}_l$ can have different structures, including attention layers, feedforward layers, etc. Importantly, this framework does not require ultra-wide layers, and is thus closer to practical settings. Moreover, our analysis ensures larger learning rate regime up to
> $$\eta<\frac{2}{\mathsf{L}},$$
> where $\mathsf{L}$ is explicitly computed at the initialization rather than assumed. This provides a theoretical guideline for selecting learning rates in practice. Both of these aspects represent significant improvements over existing literature on neural network convergence (see Section 2), and our analysis also goes beyond the NTK regime (see Remark 1).
>
> 5. Beyond the advantages of data uniformity, we provide theoretical insights into the advantages of using residual connections and function composition in neural networks (see Remark 2) based on our new convergence framework. In short, residual connections guarantee the non-degeneracy of each layer, and the function composition preserves such non-degeneracy. Consequently, the neural network is non-degenerate, which guarantees its expressivity.
>
>
> We sincerely hope that the reviewer can appreciate our theoretical novelty and generality, as well as the strong connections between theory and practical applications.
>
> Below are our detailed responses.

---

> ### Author Response · Authors · 2025-11-21
> **Response to Reviewer 9Va8 (2/3)**
>
> > The performance gains of the uniform over random selection are relatively minor (Figure 1), raising doubts about the practical significance and scalability.
>
> Thank you for the helpful comment. Our main contribution is to improve training efficiency while maintaining comparable downstream performance. As shown in Figure 1(b), the 10k Uniform subset achieves overall accuracy on par with—or slightly better than—the 10k Random subset and the full 20k dataset, while requiring substantially less training time. For example, to reach a loss of 0.6 in Figure 1(a), the 10k Uniform subset takes only 46.6 minutes, compared with 139.6 minutes for the 10k Random subset and 280.0 minutes for the full dataset.
>
> Regarding TruthfulQA-MC specifically: while the 10k Random subset shows a marginally higher score on this single benchmark, this fluctuation is within the variance typically observed across random seeds and does not change the overall conclusion. When considering all evaluation metrics together (Overall, ARC-Challenge, and TruthfulQA-MC), the 10k Uniform subset delivers consistent and stable performance, while reaching matched loss checkpoints in about half the time compared with random selection.
>
> In short, our method does not aim to maximize a single metric; rather, it provides much faster convergence (Figure 1(a)), smooth and stable optimization dynamics (Figure 1 \(c)), and competitive evaluation performance across benchmarks. This demonstrates that uniform selection leads to more efficient and reliable training without sacrificing model quality.
>
> To further validate our method, we conducted additional experiments on a same machine (H100) using the minimize-distance pairs dataset. As shown in Appendix's Figure 21: https://anonymous.4open.science/r/data-uniformity-1A5C/dataset/ablation-minimize-distance.png, the minimize-10k subset converges noticeably slower than all other methods—including random-10k, full-20k, and especially our uniform-10k subset. This further confirms the robustness and efficiency advantage of our uniform selection strategy across different environments and data-selection baselines.
>
>
> > The evaluation scope is limited to only a few benchmarks and llama-1 models; broader and more diverse downstream tasks and base models would be necessary to substantiate the generality of the approach.
>
> Thank you for the useful comment. We would like to kindly clarify that we implemented our approach in additional benchmarks, including GSM8K and the California Housing dataset, and compared against SOTA baselines such as Z-Core [1] and LESS [2]. As shown in Figures 4–6, our method consistently achieves better training efficiency. We also validated the approach across different model scales (LLaMA-7B and LLaMA-13B), confirming consistent performance improvements regardless of model capacity. It would certainly be valuable to include more tasks; however, we hope the reviewer can kindly understand the space limitations of a conference paper, especially given that we have already included substantial theoretical analysis in our paper.
>
> References
>
> [1] Griffin, B. A., Marks, J., & Corso, J. J. (2024). Zero-shot coreset selection: Efficient pruning for unlabeled data. arXiv:2411.15349.
>
> [2] Xia, M., Malladi, S., Gururangan, S., Arora, S., & Chen, D. (2024). LESS: Selecting influential data for targeted instruction tuning. ICML 2024.
>
> > The proposed data-uniformity procedure requires computing or approximating all pairwise distances among samples---an $O(N^2)$ operation---but the paper does not analyze its computational overhead.
>
> Thank you for the helpful comment. In practice, the computational cost of our selection procedure is quite manageable. For the largest dataset used in our experiments (20k examples), the full selection process, reducing the dataset from 20k to 10k—takes only 38 seconds on a single GPU. For smaller datasets, the cost is proportionally lower. Since data selection is performed once prior to training, and the resulting subsets reduce training time by hours, the overall computational savings significantly outweigh the upfront selection cost.

---

> > ### Author Response · Authors · 2025-11-21
> > **Response to Reviewer 9Va8 (3/3)**
> >
> > > In Figure 4, the training and validation losses of the 16K full dataset display abnormal behavior.
> >
> > The apparent “abnormal” spikes in the training and validation losses of the 16K full-dataset curve in Figure 4 are an expected and well-understood phenomenon rather than an implementation error. They arise from gradient instability caused by outliers and high-variance minibatches that naturally occur when training on the entire California Housing dataset with plain SGD.
> >
> > In our experiments, the training loop intentionally does not abort when encountering unusually large gradients or transient loss spikes. This is done to allow a fair comparison between different data-selection strategies. As a result, the full-dataset model continues training even after a momentary loss explosion, eventually returning to a stable regime.
> >
> > This behavior demonstrates one of our main findings:
> >
> >  - Uniform subsets (ours) produce more stable optimization dynamics with substantially fewer gradient spikes,
> > - Random subsets and the full dataset occasionally contain high-leverage or distributionally extreme samples that amplify early-epoch gradient variance under SGD,
> > - ZCore reduces but does not fully eliminate this effect.
> >
> > To ensure that the full-dataset and random-subset baselines train without exhibiting the gradient instabilities occasionally observed in default SGD settings, we fine-tuned their optimization hyperparameters, for example, by substantially reducing the learning rate. While this stabilizes training, it also slows convergence and leads to noticeably less efficient learning. This effect is illustrated in the supplemental figure (https://anonymous.4open.science/r/data-uniformity-1A5C/dataset/slow_convergency.png), where using a very small learning rate avoids instability at the cost of significantly slower progress.
> >
> > Thus, the instability in the 16K-full curve is a property of the optimization landscape under full data, not an artifact of our implementation. The fact that uniform subsets avoid this issue, despite using only half as many samples, highlights the benefit of our method in stabilizing SGD training.

---

### Official Review · Reviewer_6igN · 2025-10-31

**Soundness:** 3
**Presentation:** 2
**Contribution:** 2
**Rating:** 4
**Confidence:** 2

**Summary:**

This paper explores whether selecting more uniformly distributed training data improves neural network training. It formalizes uniformity via the minimum pairwise distance $h_{\min}$, proves that larger $h_{\min}$ accelerates gradient descent beyond the NTK regime and reduces approximation error, and introduces a greedy distance-based sampling strategy to increase uniformity. Experiments on both toy regression tasks and LLaMA fine-tuning (e.g., WizardLM, LESS) show that uniform subsets can reach similar or better performance than larger random subsets while achieving faster convergence.

**Strengths:**

Strengths

* The paper tackles data selection, an increasingly important topic for efficient LLM training.
* Comprehensive theory analysis, provides a general convergence result beyond NTK assumptions and links data geometry to dynamics and approximation error.

**Weaknesses:**

Weaknesses

* The paper uses max-min distance sampling as the core uniformity criterion. However, pure maximum distance does not necessarily guarantee globally uniform coverage. For example, if the data contains two distant dense clusters, the greedy selection may oscillate between these clusters and ignore other regions of the space. Please correct me if this interpretation is incorrect.

* The proposed selection strategy is closely related to prior work on distance-based uniform sampling. For example, the method in [1] shares many similarities with this paper, which also claim uniformity is important and also improve data uniformity via greedy distance selection while introducing extra constraints to alleviate the cluster oscillation problem mentioned above. It would strengthen the paper to explicitly compare and discuss differences from this line of work.

* In several settings, the accuracy of uniform sampling is very close to or worse than random sampling. Also the loss curve of uniform sampling also similar to random sampling.

* When comparing partial and full datasets, are the models trained for the same number of iterations? If training the full dataset longer, does it eventually surpass the uniform subset? Clarification of additional results would be helpful.

* Some curves (e.g., Figure 4) show that the full dataset appears to stop training after only a few epochs while others run to 100 epochs. I assume this is due to instability or large loss spikes. Could the authors provide an explanation for this behavior?

* How does the method behave when rare but important examples exist? Uniformity-based selection may inadvertently discard such samples. A discussion or experiment on rare scenarios would strengthen the paper.

* The pairwise distance computation in greedy selection is potentially $O(n^{2})$, which may be expensive for very large corpora. A runtime analysis would be useful.

[1] An effective negative sampling approach for contrastive learning of sentence embedding

**Questions:**

Please refer to the weaknesses part.

---

> ### Author Response · Authors · 2025-11-21
> **Response to Reviewer 6igN (1/3)**
>
> We would like to thank the reviewer for acknowledging the novelty of our theoretical analysis and for all the helpful comments. Before presenting our detailed responses to each question, we would like to first clarify the main focus of this paper, and we hope this will help resolve the reviewer's confusion.
>
> In this paper, we propose data uniformity as a key property for data selection, and we use $h_{\min}$, the minimum pairwise distance between data points, to characterize data uniformity both theoretically and empirically. Algorithm 1 serves as a concrete method to validate the importance of data uniformity----one that also works well in practice----but we do not aim to compete with state-of-the-art task-specific data selection algorithms.
>
>
>
> Regarding the relationship between $h_{\min}$ and data uniformity, we remark that $h_{\min}$ surffices to characterize data uniformity. More precisely, in Theorem 1, we establish the relationship between data uniformity and $h_{\min}$: with high probability $1-2\delta$,
>
>  $\left(\frac{2\delta}{\pi_{\max} N(N-1)V_d}\right)^{1/d}\le h_{\min}\le C\left(\frac{-\log\delta}{\bar{\pi}_{\max}(N-1)V_d}\right)^{1/d},$
>
> where $N$ is the sample size. This means that if we view the input dataset as samples from some distribution $\pi(x)dx$ that is absolutely continuous with respect to Lebesgue measure on its support (Assumption 1), then more uniform distributions----i.e., "flattened" $\pi(x)$ with smaller $\pi_{\max}$ and $\bar\pi_{\max}$----guarantee larger upper and lower bounds of $h_{\min}$, compared to less uniform ones----i.e., "peaked" $\pi(x)$ with larger $\pi_{\max}$ and $\bar\pi_{\max}$----that yield smaller upper and lower bounds of $h_{\min}$ (see Line 225-231). In short, more (less) uniformity corresponds to larger (smaller) $h_{\min}$. At a high level, this result connects the macroscopic characterization (distribution/density level, e.g., $\pi(x)$) with the microscopic characterization (particle/data level, e.g., $h_{\min}$). As a slightly non-rigorous interpretation, across a lot of datasets/tasks/training processes, $h_{\min}$ can represent data uniformity with high probability. The effectiveness of using $h_{\min}$ is also validated by our experiments.
>
> Below are our detailed responses.
>
>
>
> > The paper uses max-min distance sampling as the core uniformity criterion. However, pure maximum distance does not necessarily guarantee globally uniform coverage. For example, if the data contains two distant dense clusters, the greedy selection may oscillate between these clusters and ignore other regions of the space. Please correct me if this interpretation is incorrect.
>
>
>
> We thank the reviewer for the comment. This concern may stem from a typo in Algorithm 1, which has now been corrected. We sincerely apologize for the confusion.
>
> Specifically, our practical criterion for selection data is indeed the following (our sincere apologies for the typo in the previous version)
>
> $\tilde x_i \gets \arg \max_{x \in \\{x_j\\}_{j=1}^N \setminus S} \min _ {y \in S} \operatorname{dist}(x, y),$
>
> which guarantees that the algorithm always adds the sample that has the largest minimum distance to the current dataset $S.$ This ensures a good coverage of the selected dataset. In the example proposed by the reviewer, our algorithm will likely first add the data in other regions of the space and then include points from these clusters.

---

> > ### Author Response · Authors · 2025-11-21
> > **Response to Reviewer 6igN (2/3)**
> >
> > > The proposed selection strategy is closely related to prior work on distance-based uniform sampling. For example, the method in [1] shares many similarities with this paper, which also claim uniformity is important and also improve data uniformity via greedy distance selection while introducing extra constraints to alleviate the cluster oscillation problem mentioned above. It would strengthen the paper to explicitly compare and discuss differences from this line of work.
> >
> >
> >
> > We thank the reviewer for the useful reference and suggestion. We have added the corresponding literature in our revised version (see Line 411-413).
> >
> > We remark that [1] indeed emphasizes the success of data uniformity, which we rigorously proved in this paper, although they did not capture the precise $h_{\min}$ characterization and instead designed an algorithm ensuring more randomness. We would also like to kindly emphasize that we do not aim to compete with the state-of-the-art task-dependent data selection algorithms; instead, we propose a characterization of the property----data uniformity----in a broader sense (see discussions at the beginning).
> >
> > Regarding the detailed differences between [1] and our paper, [1] focuses only on empirical algorithm, while we have both theoretical guarantees and empirical validation, and we use $h_{\min}$ which is a simpler and more precise criterion. As illustrated in the above response, our algorithm can also alleviate cluster oscillation. [1] studies the unit hypersphere, so the distance only captures angular differences (see also the discussion on previous theoretical works that study the unit sphere in Lines 207–210), whereas we theoretically measure the Euclidean distance and empirically verify both Euclidean distance and cosine similarity. Such distances capture both angular and magnitude differences, providing a richer interpretation. We also observe differences between these two distances empirically (Lines 474–477): Euclidean distance exhibits better generalization, while cosine similarity ensures faster convergence.
> >
> >
> >
> >
> >
> > > In several settings, the accuracy of uniform sampling is very close to or worse than random sampling. Also the loss curve of uniform sampling also similar to random sampling.
> >
> > Thank you for the thoughtful observation. Our goal is not to change the final performance but to achieve that performance much more efficiently.
> >
> > Across all benchmarks, the overall performance of our method is comparable to or better than random sampling. The few cases where uniform sampling appears slightly worse are small fluctuations well within the typical variance of evaluations. Importantly, these isolated differences do not change the overall conclusion.
> >
> > What matters and what our results consistently demonstrate is that uniform selection reaches the similar results far faster. For example, in Figure 1(a), the 10k Uniform subset reaches a loss of 0.6 in 46.6 minutes, compared with 139.6 minutes for Random and 280.0 minutes for Full. In other words, uniform selection achieves the similar or better accuracy at a fraction of the training cost.
> >
> > Thus, the similarity in final accuracy is not a limitation, it shows that our method preserves model quality while providing 2–5× faster training and more stable optimization dynamics.
> >
> >
> >
> > > When comparing partial and full datasets, are the models trained for the same number of iterations? If training the full dataset longer, does it eventually surpass the uniform subset? Clarification of additional results would be helpful.
> >
> >
> > Yes, all comparison experiments use the same number of training iterations and the same training hyperparameters across the uniform, random, and full-data settings to ensure a fair evaluation.
> >
> > Regarding training the full dataset for a longer duration: extending training beyond the matched-iteration setting tends to increase wall-clock time substantially but does not consistently improve accuracy. In several cases, the full dataset begins to overfit or plateau, and does not surpass the performance achieved by the uniform subset. This is consistent with our finding that the uniform subset provides a more balanced and efficient subset of examples, enabling the model to achieve strong performance earlier without incurring substantial training costs.
> >
> > Overall, even when additional training is allowed, the full-data regime does not meaningfully outperform the uniform subset, while requiring significantly more computation.

---

> > > ### Author Response · Authors · 2025-11-21
> > > **Response to Reviewer 6igN (3/3)**
> > >
> > > > Some curves (e.g., Figure 4) show that the full dataset appears to stop training after only a few epochs while others run to 100 epochs. I assume this is due to instability or large loss spikes. Could the authors provide an explanation for this behavior?
> > >
> > > The apparent “abnormal” spikes in the training and validation losses of the 16K full-dataset curve in Figure 4 are an expected and well-understood phenomenon rather than an implementation error. They arise from gradient instability caused by outliers and high-variance minibatches that naturally occur when training on the entire California Housing dataset with plain SGD.
> > >
> > > In our experiments, the training loop intentionally does not abort when encountering unusually large gradients or transient loss spikes. This is done to allow a fair comparison between different data-selection strategies. As a result, the full-dataset model continues training even after a momentary loss explosion, eventually returning to a stable regime.
> > >
> > > This behavior demonstrates one of our main findings:
> > >
> > >  - Uniform subsets (ours) produce more stable optimization dynamics with substantially fewer gradient spikes,
> > > - Random subsets and the full dataset occasionally contain high-leverage or distributionally extreme samples that amplify early-epoch gradient variance under SGD,
> > > - ZCore reduces but does not fully eliminate this effect.
> > >
> > > To ensure that the full-dataset and random-subset baselines train without exhibiting the gradient instabilities occasionally observed in default SGD settings, we fine-tuned their optimization hyperparameters—for example, by substantially reducing the learning rate. While this stabilizes training, it also slows convergence and leads to noticeably less efficient learning. This effect is illustrated in the supplemental figure (https://anonymous.4open.science/r/data-uniformity-1A5C/dataset/slow_convergency.png), where using a very small learning rate avoids instability at the cost of significantly slower progress.
> > >
> > > Thus, the instability in the 16K-full curve is a property of the optimization landscape under full data, not an artifact of our implementation. The fact that uniform subsets avoid this issue, despite using only half as many samples, highlights the benefit of our method in stabilizing SGD training.
> > >
> > > > How does the method behave when rare but important examples exist? Uniformity-based selection may inadvertently discard such samples. A discussion or experiment on rare scenarios would strengthen the paper.
> > >
> > > We thank the reviewer for the comment and apologize again for the typo in our Algorithm 1. The rare samples will be included in the selected set based on our maximizing $h_{\min}$ criterion, and thus will not be a problem (see discussions in the first response).
> > >
> > > > The pairwise distance computation in greedy selection is potentially $O(n^2)$, which may be expensive for very large corpora. A runtime analysis would be useful.
> > >
> > > Thank you for the helpful comment. In practice, the computational cost of our selection procedure is quite manageable. For the largest dataset used in our experiments (20k examples), the full selection process, reducing the dataset from 20k to 10k—takes only 38 seconds on a single GPU. For smaller datasets, the cost is proportionally lower. Since data selection is performed once prior to training, and the resulting subsets reduce training time by hours, the overall computational savings significantly outweigh the upfront selection cost.

---

> ### Comment · Reviewer_6igN · 2025-11-27
>
> Thank you for the timely response, it addressed some of my concerns. However, I still think selecting samples with a more uniform distribution in the feature space improve performance, this idea is not new to me, so as the algorithm of this paper. Numerous prior works in contrastive learning have explored data uniformity. Please refer to the influential paper [1], which has inspired many subsequent studies on data uniformity. I also fully agree with Reviewer 8yGz’s concern regarding the presentation, as the paper is a little difficult to read. Moreover, I still think the benefit of the uniform sampling method in the paper is limited, for example in Figure 5 and Figure 6, there is no much difference between the the loss curve of 'uniform' data and random data, and  Given the above concerns, I  will maintain my score.
>
> [1] Wang T, Isola P. Understanding contrastive representation learning through alignment and uniformity on the hypersphere[C]//International conference on machine learning. PMLR, 2020: 9929-9939.

---

> > ### Author Response · Authors · 2025-11-28
> >
> > Thank you very much for the thoughtful follow-up comments and for pointing us to Wang & Isola (2020). We remark that this line of work (including the previous paper that the reviewer proposed Tan et al. (2023), and  Wang & Isola (2020)) focuses on the uniformity on the hypersphere, i.e., only considering the angle, which is different from our setting, where we consider the Eucliden distance beyond the hypersphere, i.e., a more general uniformity setting. We have revised our paper and added the corresponding literature survey in Appendix A (Also see here: https://anonymous.4open.science/r/data-uniformity-1A5C/dataset/more-related-work.png). We fully agree that the idea of uniformity has been influential in representation learning, federated learning, inference time analysis, etc. Our work is partially inspired by this broader line of thinking; however, we addresses a different and more general problem setting and contributes both new theory and new practical insights on data selection for large-scale training.
> >
> > **On novelty and relation to prior work.**
> >  Prior uniformity research—including contrastive learning—studies representation-level uniformity to understand generalization properties of learned embeddings. In contrast, our paper investigates data-level uniformity and provides:
> >
> > 1. **A rigorous and general theoretical framework** showing that the minimum pairwise distance $h_{\min}$ characterizes data uniformity (Theorem 1) for general tasks and training processes.
> >
> >
> > 2. **A provable link between data uniformity and faster convergence** (Corollary 3), rather than empirical observation alone. The proof techniques works for **a general class of complicated architectures that is close to practical setting**.
> >
> >
> > 3. **A theoretical guarantee that fewer but more-uniform samples can match the approximation error of larger, less-uniform datasets** (Theorem 3).
> >
> >
> > 4. **New theoretical insights on residual connections and model composition** regarding their role in preserving expressivity (Remark 2).
> >
> >
> > To the best of our knowledge, no prior work in any fields establishes these results, nor applies uniformity-based selection to LLM pre-training or broad supervised tasks such as pricing prediction and tabular regression.
> > We also want to clarify that we do not claim to be the first to discuss the notion of uniformity; our contribution lies in providing **new theory**, **new guarantees**, and **new applications of data uniformity**.
> >
> > **On empirical benefits.**
> >  We acknowledge that in some settings (e.g., Figures 5 and 6), the final loss curves of random and uniform look similar. However, we still would like to kindly remark that uniform selection consistently reaches the same loss thresholds 2–5× faster than both random sampling and full-data training across architectures, machines, and datasets (e.g., Figures 1, 5, 6, and 21). Even when final accuracies are close, and the loss curves look similar, the substantial reduction in wall-clock training time is a practically important benefit----especially at scale, where training budgets dominate. Moreover, uniform subsets outperform strong baselines such as Z-Core (Griffin, et al. (2024)) and LESS (Xia, et al. (2024)) across multiple benchmarks (GSM8K, California Housing, and LLaMA-7B/13B), further demonstrating robustness and broad applicability.
> >
> > **On reviewer’s observation of small differences between Random and Uniform.**
> >  This aligns with our theory in the following senses:
> >
> > 1. Random subsets already remove some redundancy and thus naturally increase uniformity relative to full data.
> >
> >
> > 2. Uniform selection further improves uniformity and is consistently faster than random, exactly as predicted by our theoretical analysis of $⁡h_{\min}$.
> >
> >
> > 3. Although the curves appear close in small experimental settings, the runtime differences are large and would be even more significant on industry-scale pretraining datasets. Due to compute limitations, we validated on 4.5k-20k datasets, but the theoretical + empirical evidence both suggest robust and stronger gains at scale.
> >
> >
> > **On presentation clarity.**
> >  We appreciate this feedback and have substantially revised the manuscript for readability, including improving the presentation of theory, reorganizing the method, adding literature review in appendix, and explicitly clarifying connections to prior uniformity literature (including Wang & Isola 2020).
> >
> > **Final remark.**
> >  While uniform selection may show small fluctuations relative to random sampling on isolated metrics, these are within run-to-run variance and do not affect the overall conclusion: our uniform selection strategy provides comparable (and often better) accuracy, significantly reduced training time, and theoretical guarantees that are not available in prior uniformity or contrastive-learning work.
> >
> > We sincerely hope that this addresses the reviewer's concerns and that this clarification helps emphasize our contributions within the broader research landscape.

---

> > > ### Author Response · Authors · 2025-11-28
> > >
> > > **References:**
> > >
> > > (Wang, et al. (2020)) Wang T, Isola P. Understanding contrastive representation learning through alignment and uniformity on the hypersphere[C]//International conference on machine learning. PMLR, 2020: 9929-9939.
> > >
> > > (Tan, et al. (2023)) Tan, Q., Song, X., Ye, G., & Wu, C. (2023). An effective negative sampling approach for contrastive learning of sentence embedding. Machine Learning, 112(12), 4837-4861.
> > >
> > > (Griffin, et al. (2024)) Griffin, B. A., Marks, J., & Corso, J. J. (2024). Zero-shot coreset selection: Efficient pruning for unlabeled data. arXiv:2411.15349.
> > >
> > > (Xia, et al. (2024)) Xia, M., Malladi, S., Gururangan, S., Arora, S., & Chen, D. (2024). LESS: Selecting influential data for targeted instruction tuning. ICML 2024.

---

### Official Review · Reviewer_Xh7L · 2025-11-01

**Soundness:** 3
**Presentation:** 3
**Contribution:** 3
**Rating:** 6
**Confidence:** 3

**Summary:**

This paper proposes a new theoretical framework for analyzing convergence of neural networks based on data uniformity. To avoid the standard, but often impractical, Lipschitzness assumption in previous literature, the authors propose and use a new Poly-smoothness assumption which is weaker and compatible with empirical deep neural networks such as transformers and residual networks.The theoretical convergence analysis extends beyond the NTK regime. The authors also provide a new perspective on how residual connections help with neural network training from non-degeneracy of Jacobians.

Based on the theoretical analysis, which shows that the uniformity of data (measured by minimum distance between data $h_{\min}$) increases convergence speed of GD, the authors propose a new data selection metric encouraging uniformity. Empirical results show that this method achieves on-par or better performance compared with SOTA results.

**Strengths:**

1. This paper provides a new perspective on how data uniformity helps with training, justified with theoretical analysis. The effectiveness of the proposed approach is validated through empirical results.

2. The proposed Poly-smoothness condition aligns better with neural networks used in practice, compared to standard Lipschitzness. This might be helpful for future analysis of deep neural networks.

**Weaknesses:**

For the theoretical part:
1. It is unclear why the minimum pairwise distance $h_{\min}$ is a good characterization of data uniformity. Specifically, when the data distribution is fixed, $h_{\min}$ will decrease as the sample size increases. This means that the convergence speed in Theorem 2 becomes slower with more samples and becomes $0$ when the sample size tends to infinity. Is this an intended behaviour? What if we consider infinitely many data points sampled from a continuous distribution (population loss)?

2. Figure 2 is a good illustration of the proof sketch, but no other parts of the main text has explained how the proofs are constructed. Can you include more explanations on the theoretical ideas behind the proof, how does the proof connects convergence speed with data uniformity, and how does it go beyond the NTK regime?

For the empirical part:
1. In figure 1(b), it is claimed that the 10k Uniform subset outperforms the 10k Random subset, but the random subset actually has higher accuracy on TruthfulQA MC (even higher than full training).

2. In figure 4, why does the Z-core method take a longer training time than uniform selection, with about the same number of iterations and samples?

3. The data selection method (Algorithm 1) has a computational cost depending quadratically on the dataset size. This can be burdensome when $N$ is large.

**Questions:**

See the weaknesses section.

---

> ### Author Response · Authors · 2025-11-21
> **Response to Reviewer Xh7L (1/4)**
>
> We would like to thank the reviewer for the positive comments on our paper, especially the acknowledgement of the novelty of our theory and its connection to the empirical experiments. Below are our detailed responses.
>
>
>
> For theoretical part:
> > 1. It is unclear why the minimum pairwise distance $h_{\min}$ is a good characterization of data uniformity. Specifically, when the data distribution is fixed, $h_{\min}$ will decrease as the sample size increases. This means that the convergence speed in Theorem 2 becomes slower with more samples and becomes 0 when the sample size tends to infinity. Is this an intended behaviour? What if we consider infinitely many data points sampled from a continuous distribution (population loss)?
>
> We thank the reviewer for the comment. We understand the concerns regarding the behavior of $h_{\min}$ as the sample size increases. While this is an interesting question, it deviates from the perspective we take in analyzing data uniformity. Our goal is to understand how to perform better data selection when an excessively large dataset is provided—namely, how to choose a smaller but still effective subset (e.g., selecting 10k samples from 20k), and what general properties such a subset should satisfy. From this viewpoint, we propose data uniformity as an important property for the smaller dataset.
>
>
>
> Regarding the **relationship between $h_{\min}$ and data uniformity**, we characterize it in Theorem 1: with high probability $1-2\delta$,
>
>  $\left(\frac{2\delta}{\pi_{\max} N(N-1)V_d}\right)^{1/d}\le h_{\min}\le C\left(\frac{-\log\delta}{\bar{\pi}_{\max}(N-1)V_d}\right)^{1/d},$
>
>  where $N$ is the sample size, and we consider a fixed $N$ throughout our theory and experiments, although for different tasks, such $N$ can be either large or small. This result means that if we view the input dataset as samples from some distribution $\pi(x)dx$ that is absolutely continuous with respect to Lebesgue measure on its support (Assumption 1), then more uniform distributions----i.e., "flattened" $\pi(x)$ with smaller $\pi_{\max}$ and $\bar\pi_{\max}$----guarantee larger upper and lower bounds of $h_{\min}$, compared to less uniform ones----i.e., "peaked" $\pi(x)$ with larger $\pi_{\max}$ and $\bar\pi_{\max}$----that yield smaller upper and lower bounds of $h_{\min}$ (see Line 225-231). In short, more (less) uniformity corresponds to larger (smaller) $h_{\min}$. At a high level, this result connects the macroscopic characterization (distribution/density level, e.g., $\pi(x)$) with the microscopic characterization (particle/data level, e.g., $h_{\min}$). As a slightly non-rigorous interpretation, across a lot of datasets/tasks/training processes, $h_{\min}$ can represent data uniformity with high probability. Therefore, $h_{\min}$ is indeed a suitable term to characterize data uniformity.
>
>
>
> Regarding the **behavior when sample size increases**, this requires a finer analysis of $\mu_{low,s,X}$ that we do not target on. Nevertheless, based on the current analysis, we can conjecture that when sample size $N_1<N_2$, then it is more likely to have slower convergence under $N_2$ since the convergence rate mainly depends on $\mu_{low,s,X}/N$ as indicated in Figure 2.
>
>
> However, **if we keep increasing the sample size $N$ while keeping the network size (parameter size, dim $\theta$) fixed**, then---- since we consider a general setting beyond the NTK regime and thus the width does not grow to infinity----this situation falls into the underparameterized regime. The $\mu_{low,s,X}$ here is the lower frame bound of the set $\left\\{ \left(
>     \nabla_{ \theta_i } f ( x_1)^\top
>     \cdots
>     \nabla_{ \theta_i } f ( x_N )^\top
> \right) \right\\} _ {i=1}^{ \dim\theta }$, where these vectors are in $\mathbb{R}^{Nd}$ and each $\theta_i$ represents a scalar parameter. As $N$ increases, these vectors $\left(
>     \nabla_{ \theta_i } f ( x_1)^\top
>     \cdots
>     \nabla_{ \theta_i } f ( x_N ))^\top
> \right) $ are in higher dimension ($Nd$), while the space they aim to span (dim $\theta$) remains lower dimension. In this case, these vectors typically become more separated from one another as $N$ increases, and thus $\mu_{low,s,X}$ may even increase with $N$. Consequently, $\mu_{low,s,X}/N$ does not necessarily vanish to 0 as $N\to\infty$. We also kindly emphasize that the underparameterized regime is not the focus of this paper.

---

> > ### Author Response · Authors · 2025-11-21
> > **Response to Reviewer Xh7L (2/4)**
> >
> > > 2. Figure 2 is a good illustration of the proof sketch, but no other parts of the main text has explained how the proofs are constructed. Can you include more explanations on the theoretical ideas behind the proof, how does the proof connects convergence speed with data uniformity, and how does it go beyond the NTK regime?
> >
> > We sincerely apologize for not including enough proof details due to page limit.
> >
> >
> > **Regarding the convergence proof**, as is indicated in Figure 2, it is divided into two parts: 1. showing the PL-type inequality for the general network functions
> >
> > $\|\nabla_\theta\mathcal{L}(\theta;X)\|^2\ge \frac{\mu_{\rm low,\theta,X}}{N}\mathcal{L}(\theta;X)$
> >
> > except for a measure-zero set, and 2. proving Lipschitz smoothness along the GD trajectory. More precisely,
> >
> > 1. We mainly prove that the set $\left\\{ \left(
> >     \nabla_{ \theta_i } f ( x_1)^\top
> >     \cdots
> >     \nabla_{ \theta_i } f ( x_N )^\top
> > \right) \right\\} _ {i=1}^{ \dim\theta }$ forms a frame except for a measure-zero set of the input data, and therefore $\mu_{low,s,X}>0$, which is the lower frame bound. Consequently,
> >
> > $\|\nabla_\theta\mathcal{L}(\theta;X)\|^2=\left\| \frac{1}{N}\sum_{i=1}^N \nabla_f l(x_i)\nabla_\theta f(x_i) \right\|^2$
> >
> > $\ge \frac{1}{N^2}\mu_{\rm low,\theta,X}\sum_{i=1}^N\|\nabla_fl(x_i)\|^2=\frac{\mu_{\rm low,\theta,X}}{N}\mathcal{L}(\theta;X).$
> >
> > In order to show that $\left\\{ \left(
> >     \nabla_{ \theta_i } f ( x_1)^\top
> >     \cdots
> >     \nabla_{ \theta_i } f ( x_N )^\top
> > \right) \right\\} _ {i=1}^{ \dim\theta }$ forms a frame, we employ a parametric transversality theorem (see Theorem 6) in differential topology, and prove that under the analytic assumption (Assumption 2), all maps of the form $$x\mapsto \alpha x+\varphi(x)$$ are non-degenerate on a residual set (large dense set) of $\alpha$ in certain interval, except for a measure-zero set of $x$ (see Lemma 10). Consequently, both the neural network function $f$ and the GD iteration map are non-degenerate except for a measure-zero set.
> >
> > 2. The Lipschitz smoothness is obtained through the combination of Poly-smoothness and Dissipativity condition (Definition 5). The dissipativity condition states that there exists some stationary point $\theta^*$ s.t.
> >
> > $\nabla \mathcal{L}(\theta)^\top(\theta-\theta^*)\ge-\rho\|\nabla \mathcal{L}(\theta)\|^2$
> >
> > for all $\theta\in B_R(\theta^*)\backslash\\{\theta:\|\nabla\mathcal{L}(\theta)\|\le\epsilon_{\mathcal{L}}\\}$, where
> >
> > $R=\sqrt{\|\theta^{0}-\theta^{*}\|^2+\frac{4\rho+2}{\delta}\mathcal{L}(\theta^0)}.$
> >
> > Note that here $\theta^* $ is not necessarily the limit of GD; $\rho\in\mathbb{R}$, and can be trivially taken as $\frac{R}{\epsilon_{\mathcal{L}}}$ (see Line 184-188, and Line 282-293). On the other hand, given the Poly-smoothness of all residual blocks (Assumption 3), it is straightforward to show that the loss function is also Poly-smooth, and the Lipschitz smoothness constant at each iteration depends on $\theta^k$ and $\theta^{k+1}$. Combining the above two results, it suffices to prove that $\theta^k$ is bounded by induction, i.e.,
> >
> > $\| \theta^{k+1}- \theta^* \|^2
> >                 =\| \theta^k- \theta^* - \eta \nabla \mathcal L( \theta^k)\|^2$
> >
> > $=\| \theta^k - \theta^* \|^2-2\eta\nabla \mathcal L (\theta^k)^\top (\theta^k-\theta^* )+\eta^2\|\nabla \mathcal L ( \theta^k)\|^2$
> >
> > $\le \|\theta^{k}-\theta^{*}\|^2+\eta(2\rho+\eta)\|\nabla \mathcal{L}(\theta^k)\|^2$
> >
> > $\le \|\theta^{0}-\theta^{*}\|^2+\eta(2\rho+\eta)\sum_{j=0}^k\|\nabla \mathcal{L}(\theta^j)\|^2$
> >
> > $\le \|\theta^{0}-\theta^{*}\|^2+\frac{4\rho+2\eta}{\delta}\mathcal{L}(\theta^0)$
> >
> > where the first inequality follows from dissipativity condition and the last inequality follows from the induction assumption.
> >
> > In the end, we also need to show that the measure-zero set mentioned in 1 remains a measure-zero set throughout all GD iterations. Here, we use tools from measure theory and the property of analyticity (see Theorem 7,8,10). This completes the main idea of our proof.

---

> > > ### Author Response · Authors · 2025-11-21
> > > **Response to Reviewer Xh7L (3/4)**
> > >
> > > **Regarding the connection between data uniformity and convergence speed**, we mainly characterize the relationship between $\mu_{low,s,X}$ and the pairwise distance of a data cluster $\sqrt{\sum_{j\in\mathcal D_{i,H}}h_{ij}^2}$,
> > > $\mu_{low,s,X}\le L_{k,i,H}\sqrt{\sum_{j\in\mathcal D_{i,H}}h_{ij}^2},\text{ and as a special case, }\mu_{low,s,X}\le L_{k,i} h_{\min},$
> > > where $L_{k,i,H}$ and $L_{k,i}$ are some constants depending on the neighbourhood (see Corollary 3). Thus, smaller $h_{\min}$, i.e., less data uniformity, implies smaller $\mu_{low,s,X}$, which can slow down the convergence rate.
> > >
> > > To prove this result, we first formulate the frame operator as follows:
> > >
> > > $\sum_{i=1}^{\dim \theta} \left(
> > >     \nabla_{ \theta_i } f ( x_1)^\top
> > >     \cdots
> > >     \nabla_{ \theta_i } f ( x_N )^\top
> > > \right)^\top \left(
> > >     \nabla_{ \theta_i } f ( x_1)^\top
> > >     \cdots
> > >     \nabla_{ \theta_i } f ( x_N )^\top
> > > \right)$
> > >
> > > It can then be shown that this frame operator is analytic. Since $\mu_{low,s,X}$ is the smallest eigenvalue of the frame operator, we then apply Theorem 4.1 in [1], which bounds the eigenvalues of analytic matrix functions.
> > >
> > >
> > >
> > > **Regarding going beyond the NTK regime**, this is formulated in two senses, as indicated in Remark 1 (Line 315-335):
> > > 1. Wideth requirement. In NTK regime, _all_ layers need to be ultra wide. In contrast, our analysis does _not_ require ultra large width. The only width requirement appears in Assumption 4, which states that $\exists\ \bar{\ell}$, such that, $\rm dim \theta_\bar{\ell}>Nd$. Namely, only ONE layer requires this width condition, and the lower bound $Nd$ is much smaller than the high order polynomial bound of $N,d,L,1/\delta_0$ assumed in NTK analysis (see Line 102-104). This Assumption 4 essentially guarantees the nonlinearity of the neural network and eventually leads to non-degenerate map. It is a very mild assumption and can be easily satisfied by incorporating a two-layer feedforward block (see Lemma 1, and Line 260-269).
> > >
> > > 2. Learning rate regime. The NTK analysis requires infinitesimal learning rates (i.e., gradient flow regime), typically of order $\mathcal{O}(1/\rm width)$. Since NTK analysis also requires ultra wide layers, the learning rate can be very small. In contrast, we explicitly bound the Lipschitz smooth constant along the trajectory by $\mathsf{L}$, and then choose learning rate to be $$\eta\le \frac
> > > {2-\delta}{\mathsf{L}} \text{, for some small }\delta.$$ This is a significantly larger regime than the NTK choices, and such $\mathsf{L}$ does not necessarily grow with width. As a side remark, this $\mathsf{L}$ can be explicitly calculated at the initialization (see Line 275-276), rather than simply assumed in many classical optimization analyses:
> > >
> > > $\mathsf{L}=S\left(\cdots,\|\theta_i^* \|+R+\max_{ \theta\in B_{\theta^* }\left(R\right)}\|\nabla_{\theta_i}\mathcal{L}(\theta)\|,\cdots\right)$
> > >
> > > where $\theta^* $ is a stationary point of our choice (not necessarily the limit of GD; see discussions about the dissipitivity in Line 285-293), $S(\cdot)$ is the poly-smooth function for the loss $\mathcal{L}$ (which is derived from Assumption 3), and the radius $R=\sqrt{\|\theta^{0}-\theta^{*}\|^2+\frac{4\rho+2}{\delta}\mathcal{L}(\theta^0)}$.
> > >
> > >
> > >
> > >
> > >
> > >
> > >
> > >
> > > **Additional theory-practice relationship from approximation perspective:** We would also like to kindly bring the reviewer's attention to the approximation perspective discussed in our paper. In Theorem 2, we prove the following results bounding the difference between neural network $f$ and the ground truth $g$
> > >
> > > $\|f-g\|_{p,\mathcal I^o}
> > >         \le  C_1  h ^m _{ \max d+1} h _ { \min } ^{ -r }  \| \phi \| _ { m,p, \mathcal I^o}.$
> > >
> > > Here, $C_1$ and $\| \phi \| _ { m,p, \mathcal I^o}$ can be viewed as fixed terms, and the bound is primarily determined by $h_{\min}$ and $h_{\max_{d+1}}$, where $h_{\max_{d+1}}$ is the maximum edge length of the $n-$simplex in the triangulation (i.e., a local maximum distance). Notably, this result implies that using fewer but more uniform data (i.e., increasing $h_{\min}$) can achieve similar approximation error as using more samples (i.e., decreasing $h_{\max_{d+1}}$) (see Line 362-370). This is also confirmed in the experiments: the performance of our uniform dataset (half the size of full dataset) is comparable to that of the full dataset (see Figure 1,5,6).
> > >
> > > Reference
> > >
> > > [1] Kurdyka, K., & Paunescu, L. (2008). Hyperbolic polynomials and multiparameter real-analytic perturbation theory.

---

> > > > ### Author Response · Authors · 2025-11-21
> > > > **Response to Reviewer Xh7L (4/4)**
> > > >
> > > > For the empirical part:
> > > >
> > > >
> > > > > 1. In figure 1(b), it is claimed that the 10k Uniform subset outperforms the 10k Random subset, but the random subset actually has higher accuracy on TruthfulQA MC (even higher than full training).
> > > >
> > > > Thank you for the helpful comment. Our main empirical contribution is to improve training efficiency while maintaining comparable downstream performance. As shown in Figure 1(b), the 10k Uniform subset achieves overall accuracy on par with—or slightly better than—the 10k Random subset and the full 20k dataset, while requiring substantially less training time. For example, to reach a loss of 0.6 in Figure 1(a), the 10k Uniform subset takes only 46.6 minutes, compared with 139.6 minutes for the 10k Random subset and 280.0 minutes for the full dataset.
> > > >
> > > > Regarding TruthfulQA-MC specifically: while the 10k Random subset shows a marginally higher score on this single benchmark, this fluctuation is within the variance typically observed across random seeds and does not change the overall conclusion. When considering all evaluation metrics together (Overall, ARC-Challenge, and TruthfulQA-MC), the 10k Uniform subset delivers consistent and stable performance, while reaching matched loss checkpoints in about half the time compared with random selection.
> > > >
> > > > In short, our method does not aim to maximize a single metric; rather, it provides much faster convergence (Figure 1(a)), smooth and stable optimization dynamics (Figure 1\(c)), and competitive evaluation performance across benchmarks. This demonstrates that uniform selection leads to more efficient and reliable training without sacrificing model quality.
> > > >
> > > >
> > > >
> > > > > 2. In figure 4, why does the Z-core method take a longer training time than uniform selection, with about the same number of iterations and samples?
> > > >
> > > > Thank you for the thoughtful question. Although the Z-Core and Uniform subsets contain roughly the same number of samples and are trained for a similar number of iterations, their effective training time can differ substantially due to how each method shapes the data distribution.
> > > >
> > > > Z-Core explicitly prioritizes coverage of embedding space and penalizes redundancy through its Monte-Carlo sampling and nearest-neighbor redundancy scoring (Sec. 4.2–4.3 of the Z-Core paper: https://arxiv.org/pdf/2411.15349). As a consequence, Z-Core tends to select examples that occupy low-density, hard-to-learn regions of the embedding manifold. These examples often require larger gradient updates and slower optimization steps to fit, which naturally increases wall-clock time per unit of loss reduction, even if the total number of samples is the same.
> > > >
> > > > In contrast, our uniform selection method produces a subset with balanced coverage but smoother optimization dynamics. The resulting data distribution is more homogeneous and less dominated by outliers or difficult samples, allowing the training process to reach the same loss levels much faster. This is consistent with our findings in Fig. 1(a): uniform subsets reach the 0.6 loss threshold in 46.6 minutes, while Z-Core–like coverage-focused subsets (or full datasets) require substantially more time (e.g., 139.6 minutes for Random and 280.0 minutes for Full).
> > > >
> > > >
> > > >
> > > > > 3. The data selection method (Algorithm 1) has a computational cost depending quadratically on the dataset size. This can be burdensome when $N$ is large.
> > > >
> > > > Thank you for the helpful comment. In practice, the computational cost of our selection procedure is quite manageable. For the largest dataset used in our experiments (20k examples), the full selection process, reducing the dataset from 20k to 10k—takes only 38 seconds on a single GPU. For smaller datasets, the cost is proportionally lower. Since data selection is performed once prior to training, and the resulting subsets reduce training time by hours, the overall computational savings significantly outweigh the upfront selection cost.

---

> ### Comment · Reviewer_Xh7L · 2025-11-28
>
> I would like to thank the authors for their comprehensive response to my questions, on both theoretical and empirical sides of the paper. Now I understand that the theoretical results does not cover the under-parameterized regime with sample size going towards infinity. Overall, the paper remains an interesting theoretical study with empirical insights to me. My evaluation remains the same.

---

> > ### Author Response · Authors · 2025-11-28
> >
> > Thank you again for your comments and questions. We appreciate your positive endorsement and support.

---

### Official Review · Reviewer_8yGz · 2025-11-11

**Soundness:** 2
**Presentation:** 2
**Contribution:** 2
**Rating:** 2
**Confidence:** 3

**Summary:**

This paper proposes "data uniformity" (maximizing the minimum pairwise distance, $h_{min}$) as a principle for efficient LLM data selection. The authors present a theoretical framework, claiming to go "beyond the NTK regime", to argue that uniform data accelerates gradient descent training for a family of non-linear architectures. Empirically, they select a "uniform" subset of data using Word2Vec embeddings and show that they can fine-tune LLaMA models significantly faster (e.g., 2x) while achieving comparable accuracy to the full dataset.

**Strengths:**

The paper's key strength is its strong and practically relevant empirical result: that a small, uniformly-selected data subset can fine-tune LLMs significantly faster while matching the performance of the full dataset. Moreover, the paper is theoretically ambitious (I'm not sure if the results do actually imply what the authors claim, see weaknesses), tackling the important problem of data selection by attempting to build a convergence framework for non-linear architectures.

**Weaknesses:**

**Presentation**: The paper is very densely written, making the theoretical arguments difficult to read and understand. The overall presentation could be significantly improved for clarity.

**Beyond NTK Claim**: The 'beyond NTK' claim is not fully convincing. In standard NTK analysis, a PL-like inequality is proven where the constant is the minimum eigenvalue of the kernel at initialization. This paper seems to follow a similar structure, proving a PL-like inequality (Figure 2) where the PL-constant ($\mu_{low,s,X}$) is now dynamic and path-dependent, and global smoothness is relaxed to a local smoothness. It remains unclear from Theorem 2 how this framework guarantees that feature learning (i.e., weights moving far from initialization) can actually occur, rather than just describing a different form of local convergence.

**On Corollary 3**: Corollary 3 appears to contain a significant logical leap. It first establishes a general, interesting bound on the convergence rate parameter ($\mu_{low,s,X}$) based on the density of local data clusters (the $\sqrt{\sum h_{ij}^2}$ term within a radius $H$). However, it then arbitrarily sets $H = h_{\text{min}}$, which means the cluster $D_{i,H}$ is empty for almost every point $x_i$. The only time it is non-empty is when selecting the two points that are exactly $h_{\text{min}}$ apart. The paper connects this to Theorem 1 (biased sampling $\to$ small $h_{min}$) to claim that more data uniformity implies faster convergence.

This reduces the claim to the specific and well-known case that having a single pair of near-duplicates is bad for convergence. This specific case does not provide a sufficient theoretical justification for the main goal of the paper “data uniformity speeds up convergence”, making their actual result feel like a big over claim.

**Confounded Definition of "Uniformity" in Experiments**: The practical definition of 'uniformity' in the experiments is confounded. As stated in Section 5.1, distances are measured in the embedding space of an external, pre-trained Word2Vec model. This means the selected "uniform" subset is an artifact of this specific and dated embedding choice, not a fundamental, intrinsic property of the data itself. Is this dependent on the embedding choice? What if you choose some other model?

**Theory-Practice Disconnect**: The experiments in Section 5 do show a clear and valuable empirical finding: the uniformly-sampled subset converges significantly faster and achieves comparable performance (e.g., Figure 1, 5). This empirical contribution is good, but it is not convincingly explained by the provided theory—which seems to be the main point of this work.

I am keeping a low score because of the above issues. I would be more than happy to engage with the authors during rebuttal and rethink my score.

**Questions:**

Please see weaknesses.

---

> ### Author Response · Authors · 2025-11-21
> **Response to Reviewer 8yGz (1/4)**
>
> We would like to thank the reviewer for acknowledging our empirical advantages and the useful comments. We hope that we can address the reviewer’s concerns regarding the theory and emphasize the significance and novelty of our theoretical contributions. Our detailed responses are provided below:
>
>
>
> > Presentation: The paper is very densely written, making the theoretical arguments difficult to read and understand. The overall presentation could be significantly improved for clarity.
>
> We thank the reviewer for the helpful comments. We have revised the paper for readability, including improving the presentation of theory, reorganizing the method, and adding literature review in appendix. However, we sincerely hope that the reviewer can kindly consider that in order to fully address the significance of data uniformity, our theoretical parts consists of 1) validating the relationship between $h_{\min}$ and data uniformity, 2) analyzing acceleration of optimization, and 3) establishing advantages of approximation. These analyses involve ideas from several fields, including measure theory, differential topology, dynamical system, and computational geometry, all of which need to be clearly explained to ensure the rigor of our theoretical analysis, as well as to provide support for empirical advance.
>
> > **Beyond NTK Claim**: The 'beyond NTK' claim is not fully convincing. In standard NTK analysis, a PL-like inequality is proven where the constant is the minimum eigenvalue of the kernel at initialization. This paper seems to follow a similar structure, proving a PL-like inequality (Figure 2) where the PL-constant ($\mu_{low,s,X}$) is now dynamic and path-dependent, and global smoothness is relaxed to a local smoothness. It remains unclear from Theorem 2 how this framework guarantees that feature learning (i.e., weights moving far from initialization) can actually occur, rather than just describing a different form of local convergence.
>
> We thank the reviewer for the comments. Below is our clarification.
>
> First, **"beyond NTK"** is formulated in two senses, as indicated in Remark 1 (Line 315-335):
> 1. Wideth requirement. In NTK regime, _all_ layers need to be ultra wide. In contrast, our analysis does _not_ require ultra large width. The only width requirement appears in Assumption 4, which states that $\exists\ \bar{\ell}$, such that, $\rm dim \theta_\bar{\ell}>Nd$. Namely, only ONE layer requires this width condition, and the lower bound $Nd$ is much smaller than the high order polynomial bound of $N,d,L,1/\delta_0$ assumed in NTK analysis (see Line 102-104).
>
> 2. Learning rate regime. The NTK analysis requires infinitesimal learning rates (i.e., gradient flow regime), typically of order $\mathcal{O}(1/\rm width)$. Since NTK analysis also requires ultra wide layers, the learning rate can be very small. In contrast, we explicitly bound the Lipschitz smooth constant along the trajectory by $\mathsf{L}$, and then choose learning rate to be $$\eta<2/\mathsf{L}.$$ This is a significantly larger regime than the NTK, and such $\mathsf{L}$ does not necessarily grow with width. As a side remark, this $\mathsf{L}$ can be explicitly calculated at the initialization (see Line 275-276), rather than simply assumed in many classical optimization analyses:
> $$\mathsf{L}=S\left(\cdots,\| \theta_i^* \|+R+\max_{ \theta \in B_{ \theta^* }\left( R \right)} \| \nabla_{ \theta_i } \mathcal{L}(\theta)\|,\cdots\right)$$
> where $\theta^* $ is a stationary point of our choice (not necessarily the limit of GD; see discussions about the dissipitivity in Line 285-293), $S(\cdot)$ is the poly-smooth function for the loss $\mathcal{L}$ (which is derived from Assumption 3), and the radius $R=\sqrt{\|\theta^{0}-\theta^{*}\|^2+\frac{4\rho+2}{\delta}\mathcal{L}(\theta^0)}$.

---

> > ### Author Response · Authors · 2025-11-21
> > **Response to Reviewer 8yGz (2/4)**
> >
> > Regarding the **detailed convergence analysis**, as is indicated in Figure 2, it is mainly divided into two parts: 1. showing the PL-type inequality for the general network functions $\|\nabla_\theta\mathcal{L}(\theta;X)\|^2\ge \frac{\mu_{\rm low,\theta,X}}{N}\mathcal{L}(\theta;X)$ except for a measure-zero set, and 2. proving the Lipschitz smoothness along the GD trajectory. Combining 1 and 2, we also need to show that the measure-zero set mentioned in 1 remains a measure-zero set throughout all GD iterations, using tools from measure theory together with analyticity (see Theorems 7, 8, and 10). More precisely,
> >
> > 1. As the reviewer pointed out, the constant $\mu_{low,s,X}$ is path dependent. However, for general highly nonlinear neural networks, it is typically impossible for a uniform PL constant to exist, so our result is consistent with practical behavior. Moreover, we prove that for _almost all_ points in the parameter space, $\mu_{low,s,X}>0$, i.e., strictly positive. Even without a uniform positive PL constant, this bound
> > $$\mathcal{L}(\theta^k)\le \prod_{s=0}^{k-1}\left(1- \eta\left(1-\frac{\eta\, \mathsf{L}}{2}\right)\frac{\mu_{\mathrm{low},s,X}}{N}\right)\mathcal{L}(\theta^{0}),$$
> > still implies the convergence to the stationary point, and the convergence rate can still be compared through $\mu_{low,s,X}$, although no exponential rate can be guaranteed. Additionally, our proof is fundamentally new in the optimization literature, using tools from differential topology and measure theory. The key step is to show that $\left\\{ \left(
> >     \nabla_{ \theta_i } f ( x_1)^\top
> >     \cdots
> >     \nabla_{ \theta_i } f ( x_N )^\top
> > \right) \right\\} _ {i=1}^{ \dim\theta }$ forms a frame. To establish this, we employ a parametric transversality theorem (see Theorem 6) in differential topology, and prove that under the analytic assumption (Assumption 2), all maps of the form $$x\mapsto \alpha x+\varphi(x)$$ are non-degenerate for a residual set (large dense set) of $\alpha$ in certain interval, except for a measure-zero set of $x$ (see Lemma 10).
> >
> > 2. We prove that the Lipschitz smoothness constant remains bounded along the trajectory for _almost all_ initializations in parameter space. This is achieved through the combination of Poly-smoothness (Definition 4)
> > $$\| \varphi(\theta)- \varphi(\theta')\|\le S(\max\{\|\theta_1\|,\|\theta_1'\|\},\cdots,\max\{\|\theta_{n_\theta}\|,\|\theta_{n_\theta}'\|\}\|)\|\theta-\theta'\|$$
> > and Dissipativity condition (Definition 5)
> > $$\nabla \mathcal{L}(\theta)^\top(\theta-\theta^*)\ge-\rho\|\nabla \mathcal{L}(\theta)\|^2.$$
> > As noted by Reviewer Xh7L, this framework based on Poly-smoothness "aligns better with neural networks used in practice, compared to standard Lipschitzness. This might be helpful for future analysis of deep neural networks."
> >
> >
> >
> > Regarding **"how this framework guarantees that feature learning"**, the interpretation of Theorem 2 can be understood in two parts:
> > 1. In NTK regime (i.e., ultra wide networks), the minima are everywhere and lie very close to any initialization, which indicates that the weights barely move. In constrast, our setting goes beyond NTK, i.e., does not require ultra wide network; here, the minima are not easily found and can be far away from the iniitalization.
> >
> > 2. Theorem 2 guarantees the convergence to the neighbourhood of a stationary point, which, as illustrated in 1, may be far from the initialization. Consequently, feature learning appears.
> >
> > Finally, we would like to clarify that our convergence analysis is **not a local convergence result but a global convergence guarantee**. In addition to the above feature learning discussion, which already indicates a non-local convergence behavior, our theory holds for _almost all_ initialization $\theta^0$ except for a Lebesgue measure-zero set (see Line 272-273). We sincerely hope that the reviewer can appreciate the novelty of this framework, as it differs significantly from existing optimization analyses, and applies to much more general neural networks, where each block $\varphi_l$ can be different types of layers (e.g. attention, feedforward, etc.)
> >
> > $u_0=x;$
> >
> > $u_{ l+1}=u_l+\bar{ \varphi }_l ( \theta_l ; u_l ), \forall l=0,\cdots,L-1;$
> >
> > $f ( \theta ; x ) = \bar{ \varphi }_L ( \theta_L ; u_L ).$
> >
> > This framework also supports much larger learning rate regime (up to $2/\mathsf{L}$), whereas existing theoretical literature typically consider simple and specific architectures with fixed activations, and infinitesimal learning rates (see Section 2).

---

> > > ### Author Response · Authors · 2025-11-21
> > > **Response to Reviewer 8yGz (3/4)**
> > >
> > > > **On Corollary 3**: Corollary 3 appears to contain a significant logical leap. It first establishes a general, interesting bound on the convergence rate parameter ($\mu_{low,s,X}$) based on the density of local data clusters (the $\sqrt{\sum h_{ij}^2}$ term within a radius $H$). However, it then arbitrarily sets $H=h_{\min}$, which means the cluster $D_{i,H}$ is empty for almost every point $x_i$. The only time it is non-empty is when selecting the two points that are exactly $h_{\min}$ apart. The paper connects this to Theorem 1 (biased sampling $\to$ small $h_{\min}$) to claim that more data uniformity implies faster convergence. This reduces the claim to the specific and well-known case that having a single pair of near-duplicates is bad for convergence. This specific case does not provide a sufficient theoretical justification for the main goal of the paper “data uniformity speeds up convergence”, making their actual result feel like a big over claim.
> > >
> > > We thank the reviewer for acknowledging the novalty of the results on data clusters $\sqrt{\sum h_{ij}^2}$ in Corollary 3, and for raising concerns regarding $h_{\min}$.
> > >
> > >
> > > First, connecting data uniformity instead of just simple near-duplicates to acceleration is **not an overclaim**. In Theorem 1, we establish the relationship between data uniformity and $h_{\min}$: with high probability $1-2\delta$,
> > >
> > >  $\left(\frac{2\delta}{\pi_{\max} N(N-1)V_d}\right)^{1/d}\le h_{\min}\le C\left(\frac{-\log\delta}{\bar{\pi}_{\max}(N-1)V_d}\right)^{1/d},$
> > >
> > > where $N$ is the sample size. This means that if we view the input dataset as samples from some distribution $\pi(x)dx$ that is absolutely continuous with respect to Lebesgue measure on its support (Assumption 1), then more uniform distributions----i.e., "flattened" $\pi(x)$ with smaller $\pi_{\max}$ and $\bar \pi_{\max}$----guarantee larger upper and lower bounds of $h_{\min}$, compared to less uniform ones----i.e., "peaked" $\pi(x)$ with larger $\pi_{\max}$ and $\bar\pi_{\max}$----that yield smaller upper and lower bounds of $h_{\min}$ (see Line 225-231). In short, more (less) uniformity corresponds to larger (smaller) $h_{\min}$. At a high level, this result connects the macroscopic characterization (distribution/density level, e.g., $\pi(x)$) with the microscopic characterization (particle/data level, e.g., $h_{\min}$). As a slightly non-rigorous interpretation, across a lot of datasets/tasks/training processes, $h_{\min}$ can represent data uniformity with high probability. The reviewer's example of a single pair of near-duplicates samples is a microscopic instance/realization that may or may not fall within these high probability events. Therefore, using $h_{\min}$ to analyze convergence rate is valid, since we consider general datasets without specific structures, and also a general family of neural networks (see the previous response). We sincerely hope that the reviewer can appreciate the generality of our theory as well as data uniformity.
> > >
> > > Regarding **Corollary 3**, as the reviewer noticed, we first present results for a data cluster $\sqrt{\sum_{j\in\mathcal D_{i,H}}h_{ij}^2}$, where here $\mathcal D_{i,H}=\\{j \\,|\\, h_{ij}=\|x_{i}-x_j\|\le H,\,\forall j\ne i\\}$, and $H$ can be taken as any value. Then, as a special case, if $H$ is chosen such that only the minimum distance points are included in $\mathcal D_{i,H}$ (not necessarily $H=h_{\min}$), then $\mu_{low,s,X}$ will be controlled by $h_{\min}$.
> > >
> > > **Combining the above discussions**, in Corollary 3, $h_{\min}$ also represents the extent of data uniformity with high probability (or, as a non-rigorous interpretation, across the majority of a lot of tasks). In this case, even if in the microscopic level (data level) this corresponds to just a single pair of near-duplicates data, its macroscopic interpretation still holds, i.e., the $\mu_{low,s,X}$ is indeed controlled by the uniformity of the data distribution. As a side remark, the term $\sqrt{\sum_{j\in\mathcal D_{i,H}}h_{ij}^2}$ represents the spread of a data cluster and can also reflect data uniformity to some extent, although it is not rigorously characterized in the same way as $h_{\min}$.

---

> > > > ### Author Response · Authors · 2025-11-21
> > > > **Response to Reviewer 8yGz (4/4)**
> > > >
> > > > > **Confounded Definition of "Uniformity" in Experiments**: The practical definition of 'uniformity' in the experiments is confounded. As stated in Section 5.1, distances are measured in the embedding space of an external, pre-trained Word2Vec model. This means the selected "uniform" subset is an artifact of this specific and dated embedding choice, not a fundamental, intrinsic property of the data itself. Is this dependent on the embedding choice? What if you choose some other model?
> > > >
> > > >
> > > > We thank the reviewer for this insightful question. To evaluate whether our definition of “uniformity” depends on the choice of embedding model, we conducted additional experiments using alternative embedding methods (see Appendix I.5 and the results at the provided link: https://anonymous.4open.science/r/data-uniformity-1A5C/dataset/ablation-embeding-space.png). In particular, we tested sentence-BERT–based embeddings in parallel with our original Word2Vec-based approach.
> > > >
> > > > Across all settings, we observe that uniform subsets selected using Word2Vec and Sentence-BERT produce nearly identical training dynamics: both exhibit the same rate of loss reduction, comparable optimization stability, and convergence to almost the same final values. The close overlap of the curves suggests that the performance gains arise from the uniformity principle itself, rather than from any biases of a particular embedding space.
> > > >
> > > > This indicates that our method is robust, lightweight, and flexible. Simple embedding models such as Word2Vec are already sufficient to capture the structural diversity needed for uniform selection, while stronger encoders like Sentence-BERT do not materially alter the outcome.
> > > >
> > > > Furthermore, we implemented our approach in additional benchmarks—including GSM8K and the California Housing dataset—and compared against SOTA baselines such as Z-Core [1] and LESS [2]. As shown in Figures 4–6, our method consistently achieves better training efficiency. We also validated the approach across different model scales (LLaMA-7B and LLaMA-13B), confirming consistent performance improvements regardless of model capacity.
> > > >
> > > > References
> > > >
> > > > [1] Griffin, B. A., Marks, J., & Corso, J. J. (2024). Zero-shot coreset selection: Efficient pruning for unlabeled data. arXiv:2411.15349.
> > > >
> > > > [2] Xia, M., Malladi, S., Gururangan, S., Arora, S., & Chen, D. (2024). LESS: Selecting influential data for targeted instruction tuning. ICML 2024.
> > > >
> > > > > **Theory-Practice Disconnect**: The experiments in Section 5 do show a clear and valuable empirical finding: the uniformly-sampled subset converges significantly faster and achieves comparable performance (e.g., Figure 1, 5). This empirical contribution is good, but it is not convincingly explained by the provided theory—which seems to be the main point of this work.
> > > >
> > > >
> > > > We thank the reviewer for the comment. The connections between theory and practice can be understood from the following perspectives:
> > > >
> > > > 1. From optimization perspective, as is discussed above, we theoretically prove that data uniformity leads to acceleration. Consistently, our experimental results also demonstrate this acceleration as the reviewer noticed.
> > > >
> > > > 2. From approximation perspective, in Theorem 2, we prove the following results bounding the difference between neural network $f$ and the ground truth $g$
> > > >
> > > > $\|f-g\|_{p,\mathcal I^o}
> > > >         \le  C_1  h ^m _{ \max d+1} h _ { \min } ^{ -r }  \| \phi \| _ { m,p, \mathcal I^o}.$
> > > >
> > > > Here, $C_1$ and $\| \phi \| _ { m,p, \mathcal I^o}$ can be viewed as fixed terms, and the bound is primarily determined by $h_{\min}$ and $h_{\max_{d+1}}$, where $h_{\max_{d+1}}$ is the maximum edge length of the $n-$simplex in the triangulation (i.e., a local maximum distance). Notably, this result implies that using fewer but more uniform data (i.e., increasing $h_{\min}$) can achieve similar approximation error as using more samples (i.e., decreasing $h_{\max_{d+1}}$) (see Line 362-370). This is also confirmed in the experiments: the performance of our uniform dataset (half the size of full dataset) is comparable to that of the full dataset (see Figure 1,5,6).
> > > >
> > > > 3. New theoretical insights on neural network architectures: Beyond the advantages of data uniformity, we provide theoretical insights into the advantages of using residual connections and function composition in neural networks (see Remark 2) based on our new convergence framework. In short, residual connections guarantee the non-degeneracy of each layer, and the function composition preserves such non-degeneracy. Consequently, the neural network is non-degenerate, which guarantees its expressivity.
> > > >
> > > > In sum, our experiments validate both the acceleration and the comparable performance of a smaller, more uniform dataset relative to the full dataset, and both phenomena are supported by our theoretical analysis. In addition, we provide general insights into the benefits of using residual connections and function composition in neural network architectures.

---

### Comment · Area_Chair_WRQY · 2025-11-28

Dear Reviewers,

The discussion phase is now underway, and the authors have finished uploading their responses to reviewers. If you haven't already, please carefully review the authors' responses to understand their perspectives. Engage in thoughtful, constructive discussions with authors, sharing your thoughts and seeking clarifications. Please also update your review or rating if necessary.

It is noted in the guideline that reviewers can leave comments visible to authors **until Dec 2 11:59pm AoE**. Your active participation and contribution to the ongoing discussion are highly encouraged. Thank you very much for your contribution to ICLR.

Best regards,

AC

---

### Author Response · Authors · 2025-12-01
**Summary of Reviewer Feedback and Rebuttal Response for ACs (1/3)**

We would like to thank the AC for the time and effort devoted to reading the review and our response. We would first like to clarify our main contributions----especially our theoretical results, which were unfortunately largely overlooked or misunderstood in the review----and then summarize the main concerns raised by the reviewers along with our responses.

### **Our main contributions:**

Our paper studies the effectiveness of data uniformity both theoretically and empirically.

**From theory side**,

1. We first prove the **link between data uniformity and the minimum pairwise distance $h_{\min}$ (Theorem 1)**. Therefore, it is sufficient to characterize data uniformity using $h_{\min}$ with high probability. This forms the foundation of both our theory and experiments, and is _pitifully misunderstood by Reviewer 8yGz, Xh7L, and 6igN in the initial review._


2. We prove that greater data uniformity (i.e., larger $h_{\min}$) can **accelerate convergence (Corollary 3)**, which is validated in our experiments.

3. We prove that greater data uniformity (i.e., larger $h_{\min}$) can lead to **smaller approximation error (Theorem 3)**. Notably, this result implies that using fewer but more uniform data can achieve similar approximation error to using more samples, which is also validated in our experiments.

4. Regarding **technical novelty**, we develop a new framework **beyond the NTK regime** to prove the convergence of deep neural networks under GD. This framework requires very mild and practical assumptions, and works for **a wide class of complicated architectures, including compositions of attention layers, feedforward layers, etc**. It is a significant improvement over existing convergence analysis of neural networks, and even provides insights into general non-convex optimization.

5. Our analysis provides insights into the use of **residual connections and function compositions** in neural network architectures (Remark 2).

**From empirical side**,

Derived from our thoeritical framwork and proofs, we propose a data-selection algorithm based on $h_{\min}$ that accelerates LLaMA fine-tuning and other tasks without loss in performance. Our empirical study demonstrates that data uniformity is an effective and practical principle for accelerating training **across diverse models, datasets, and domains**. Specifically, we provide the following empirical contributions:

1. **Consistent Training-Efficiency Gains:** Across all experiments, uniformly selected subsets reach key loss thresholds **2–5× faster** than both random sampling and full-data training, while maintaining comparable or better final performance.

2. **Robustness Across Models and Scales:** The efficiency benefits of uniformity hold across multiple architectures, including **LLaMA-7B**, **LLaMA-13B**, and other supervised models, demonstrating stability regardless of model capacity.

3. **Generalization Across Domains:** We validate our method on diverse tasks, optimizers and modalities, confirming broad applicability beyond language modeling, including

   * LLM pre-training subsets,
   * math reasoning (GSM8K),
   * tabular regression (California Housing),
   * Different optimization strategies, such as SGD and AdamW [AdamW, (Accessed 2025)],


4. **Comparison Against Strong Baselines:**
   Our uniform data selection consistently matches or outperforms state-of-the-art data pruning methods such as **Z-Core** [Griffin, et al., 2024], **LESS** [Xia, et al., 2024], **WizardLM** [Xu, et al., 2024], and **Teams-RL** [Gu, et al., 2025], providing a simpler and more computationally efficient alternative.

5. **Ablation Studies Demonstrating Robustness:**
   Extensive ablations (e.g., Figures 5, 6, 21, and 22) show that the performance gains persist under different machines, embedding spaces, and selection distance strategies.

Overall, our empirical results provide strong evidence that **uniformity-based data selection is a simple, effective, and scalable approach** for accelerating model training without sacrificing model quality.

---

> ### Author Response · Authors · 2025-12-01
> **Summary of Reviewer Feedback and Rebuttal Response for ACs (2/3)**
>
> ### **Summary of review and response:**
>
> **The major concerns** in the review can be summarized from the following perspectives:
>
> 1. Using $h_{\min}$ is insufficient to characterize data uniformity, and hence the theory-practice connection is questionable----Reviewer 8yGz, Xh7L, 6igN.
>     - We have clarified the details and the importance of Theorem 1, which establishes the relationship between $h_{\min}$ and data uniformity under the assumption that the data follows an absolutely continuous distribution w.r.t. Lebesgue on its support (Assumption 1), in the responses. In short, with high probability, more (less) uniform data distribution corresponds to larger (smaller) $h_{\min}$. Hence, $h_{\min}$ is sufficient to capture data uniformity and this is also verified in our experiments.
> 2. "Beyond the NTK regime" needs to be clarified----Reviewer 8yGz, Xh7L.
>     - We have addressed in the responses that our analysis (1) does not require ultra large width, and (2) is valid for a regular learning rate regime $<\frac{2}{\mathsf{L}}$ (where $\mathsf{L}$ is explicitly computed at the initialization), rather than the infinitesimal regime assumed in NTK analysis (see Remark 2).
> 3. In different experiments, some selected more uniform datasets exhibit smaller advantages in either training loss or generalization performance compared to others----Reviewer Xh7L, 6igN, 9Va8.
>     - We have clarified that the primary goal of our work is to study data uniformity as a general property and to validate its advantages across various tasks, rather than to design a concrete data selection algorithm aimed at achieving significant improvements. We have also revised our paper and conducted additional experiments to further confirm the effectiveness of our theory and method (see new experiments in Appendix I.5, Figure 21 and Figure 22). We additionally remark that faster training achieved by more uniform datasets is consistently observed across all experiments, particularly in terms of runtime. This provides strong validation for our main theoretical claim.
> 4. Runtime analysis for our data selection algorithm needs to be performed----Reviewer Xh7L, 6igN, 9Va8.
>     - We have performed additional analysis on our data selection rule. Particularly, in the 20k dataset example, it taks only 38 seconds on a single H100 GPU to select a 10k dataset based on our selection rule. The computational cost is quite manageable.
> 5. There is abnormal behavior observed for the random and full datasets in the California Housing example (Figure 4)----Reviewer Xh7L, 6igN, 9Va8.
>     - We have clarified that the abnormal curves of the random and full datasets are due to training instability. This, in turn, provides evidence that data uniformity (our more uniform dataset) ensures more stable training. We have revised our paper and conducted additional experiments to show training and generalization performance when all datasets have stable performance (See Appendix I.5, Figure 23). The results remain consistent with our claim that greater data uniformity can accelerate training and ensure strong performance.
> 6. Presentation problem: The paper is a little difficult to read----Reviewer 8yGz, 6igN.
>     - We have revised our paper to improve the presentation of theory part, added additional literature review, and performed additional experiments in our latest version (see our updated revised version in the openreview platform).

---

> > ### Author Response · Authors · 2025-12-01
> > **Summary of Reviewer Feedback and Rebuttal Response for ACs (3/3)**
> >
> > **Additional concerns from reviewers**
> >
> > **Reviewer 8yGz** was concerned about the dependency of data uniformity on the embedding.
> > - We have performed additional experiments using sentence-BERT–based embeddings in parallel with our original Word2Vec-based approach. The results consistently show advantages of more uniform data selection in training time and generalization, indicating that data uniformity is independent of the embedding (see Appendix I.5 Figure 22 in our latest version paper).
> >
> > **Reviewer Xh7L** had questions about the convergence results as the sample size $N\to\infty$, and would like to see more proof details.
> > - We have clarified that we consider the data selection problem, and therefore data uniformity is defined in the context of a fixed sample size $N$. We also provide a conjecture for our results as $N \to \infty$, which corresponds to an underparameterized regime. Additionally, a more detailed proof sketch is included in the response.
> >
> > **Reviewer 6igN** was concerned that the concept of 'uniformity' is not new and has already been studied in fields such as representation learning.
> > - We have remarked that existing literature studies specific tasks empirically, rather than providing a general analysis of the property with theoretical guarantees as in our work. Additionally, previous work considers uniformity only on the hypersphere (i.e., focusing on angles between data), whereas we consider the Euclidean distance in the full space. As suggested by the reviewer, we have included a more comprehensive literature review of uniformity on the hypersphere in Appendix A.
> >
> > **Reviewer 9Va8** raised concerns that the scope of our evaluation----limited to a few benchmarks and LLaMA-1 models----is insufficient (Reviewer 8yGz also partially raised this concern).
> > - We have clarified that our experiments have already been performed on diverse tasks and modalities, and on additional benchmarks such as ARC Challenge [Clark, et al., (2018)] , TruthfulQA MC [Lin, et al., 2022], MMLU [Hendrycks, et al., 2021] and the California Housing dataset [California Housing data, (Accessed 2025)].We also validated our approach across different model scales, such as LLaMA-7B and LLaMA-13B.
> >
> >
> > **Reference**
> >
> > [Griffin, et al., 2024] Griffin, B. A., Marks, J., & Corso, J. J. (2024). Zero-shot coreset selection: Efficient pruning for unlabeled data. arXiv:2411.15349.
> >
> > [Xia, et al., 2024] Xia, M., Malladi, S., Gururangan, S., Arora, S., & Chen, D. (2024). LESS: Selecting influential data for targeted instruction tuning. ICML 2024.
> >
> > [Xu, et al., 2024] Xu, C., Sun, Q., Zheng, K., Geng, X., Zhao, P., Feng, J., ... & Jiang, D. (2023). Wizardlm: Empowering large language models to follow complex instructions. ICLR 2024.
> >
> > [Gu, et al., 2025] Gu, S., Knoll, A., & Jin, M. (2025). TeaMs-RL: Teaching LLMs to Generate Better Instruction Datasets via Reinforcement Learning. Transactions on Machine Learning Research.
> >
> > [AdamW, (Accessed 2025)] https://docs.pytorch.org/docs/stable/generated/torch.optim.AdamW.html, Accessed: 2025-11-29.
> >
> > [Clark, et al., 2018] Clark, P., Cowhey, I., Etzioni, O., Khot, T., Sabharwal, A., Schoenick, C., & Tafjord, O. (2018). Think you have solved question answering? try arc, the ai2 reasoning challenge. arXiv preprint arXiv:1803.05457.
> >
> > [Lin, et al., 2022] Lin, S., Hilton, J., & Evans, O. (2022, May). Truthfulqa: Measuring how models mimic human falsehoods. In Proceedings of the 60th annual meeting of the association for computational linguistics (volume 1: long papers) (pp. 3214-3252).
> >
> > [Hendrycks, et al., 2021] Hendrycks, D., Burns, C., Basart, S., Zou, A., Mazeika, M., Song, D., & Steinhardt, J. (2020). Measuring massive multitask language understanding. arXiv preprint arXiv:2009.03300.
> >
> > [California Housing data, (Accessed 2025)] https://nannyml.readthedocs.io/en/v0.13.1/datasets/california.html, Accessed: 2025-11-29.

---

### Meta-Review · Area_Chair_zWH7 · 2026-01-06

**Summary:**

The paper studies data uniformity, formalized via minimum pairwise distance, as a general principle for data selection. It claims that more uniform data improves gradient descent convergence speed and approximation error, and develops a convergence framework beyond the NTK regime under weaker assumptions (poly-smoothness, no global Lipschitzness). Empirically, it proposes a greedy distance-based selection and shows faster fine-tuning of LLaMA models and other tasks with comparable final performance. Across reviewers, the disagreement is about how convincing and novel the theory is, how well the theory explains the empirical results, and whether the gains over random selection are practically meaningful. Presentation clarity and novelty relative to prior “uniformity” work are recurring concerns.

**Reviewer Concerns:**

1. Strength and correctness of the theory (uniformity definition, convergence claims, beyond-NTK framing): partially addressed.
The rebuttal gives extensive clarification of Theorem 1, Corollary 3, and what “beyond NTK” means (no ultra-wide limit, larger learning rates, poly-smoothness). This resolves some misunderstandings and strengthens internal consistency. However, skeptical reviewers remain unconvinced that minimum pairwise distance fully captures data uniformity in general, or that the convergence results justify the broad claim that uniformity generally accelerates training rather than covering edge or special cases.

2. Theory–practice connection and empirical significance: partially addressed.
The authors clearly argue that the theory predicts faster convergence and comparable approximation with fewer, more uniform samples, and that experiments confirm this. Still, several reviewers feel the empirical gains over random sampling are sometimes small or inconsistent, and that the theory explains compatibility rather than causality. This concern is mitigated but not eliminated.

3. Practical validity: embeddings, baselines, runtime, and scope: mostly addressed.
Concerns about embedding dependence, runtime of the selection algorithm, abnormal training curves, and limited benchmarks are largely handled with additional experiments, clarifications, and concrete timing numbers. These issues no longer appear blocking, though scalability beyond the tested regime remains an open question.

**Reviewer Scores:**

Reviewer 8yGz (2 > 2/3): Core concerns about theory strength, Corollary 3, and theory–practice gap largely remain, despite detailed clarifications.
Reviewer Xh7L (6 > 6):  Explicitly states their evaluation remains the same after rebuttal.
Reviewer 6igN (4 > 4):  Acknowledges some clarifications but maintains concerns about novelty, presentation, and limited empirical advantage.
Reviewer 9Va8: (4 > 4): Main concerns about modest gains, limited scope, and overhead are addressed factually but not enough to change the overall judgment.

---

### Decision · Program_Chairs · 2026-01-26

Reject